# Solving Inverse Problems via Diffusion-Based Priors: An Approximation-Free Ensemble Sampling Approach

**Haoxuan Chen**                                                                 *haoxuanc@stanford.edu*
*Institute for Computational and Mathematical Engineering (ICME)*
*Stanford University*
*Stanford, CA 94305*

**Yinuo Ren**                                                                    *yinuoren@stanford.edu*
*Institute for Computational and Mathematical Engineering (ICME)*
*Stanford University*
*Stanford, CA 94305*

**Martin Renqiang Min**                                                          *renqiang@nec-labs.com*
*Machine Learning Department*
*NEC Labs America*
*Princeton, NJ 08540*

**Lexing Ying**                                                                  *lexing@stanford.edu*
*Department of Mathematics*
*Institute for Computational and Mathematical Engineering (ICME)*
*Stanford University*
*Stanford, CA 94305*

**Zachary Izzo**                                                                 *zach@nec-labs.com*
*Machine Learning Department*
*NEC Labs America*
*Princeton, NJ 08540*

**Reviewed on OpenReview:** *https://openreview.net/forum?id=qN8ASsfjKs*

## Abstract

Diffusion models (DMs) have proven to be effective in modeling high-dimensional distributions, leading to their widespread adoption for representing complex priors in Bayesian inverse problems (BIPs). However, current DM-based posterior sampling methods proposed for solving common BIPs rely on heuristic approximations to the generative process. To exploit the generative capability of DMs and avoid the usage of such approximations, we propose an ensemble-based algorithm that performs posterior sampling without the use of heuristic approximations. Our algorithm is motivated by existing work that combines DM-based methods with the sequential Monte Carlo (SMC) method. By examining how the prior evolves through the diffusion process encoded by the pre-trained score function, we derive a modified partial differential equation (PDE) governing the evolution of the corresponding posterior distribution. This PDE includes a modified diffusion term and a reweighting term, which can be simulated via stochastic weighted particle methods. Theoretically, we prove that the error between the true posterior and the empirical distribution of the generated samples can be bounded in terms of the training error of the pre-trained score function and the number of particles in the ensemble. Empirically, we validate our algorithm on several inverse problems in imaging to show that our method gives more accurate reconstructions compared to existing DM-based methods. Our code is available at the following Github repository https://github.com/HaoxuanSteveCO0/AFDPS-TMLR.

# 1 Introduction

Inverse problems are fundamentally challenging tasks that span multiple scientific and engineering fields like fluid dynamics (Cotter et al., 2009; Sellier, 2016), geophysics (Richter, 2021), medical imaging (Lustig et al., 2007), microscopy (Choi et al., 2007; Bertero et al., 2021), etc. These problems basically involve reconstructing an unknown parameter $x$ from incomplete and noise-corrupted measurements $y$. Due to the inherent limitations in measurements, there is often substantial uncertainty in determining the true parameter $x$. Instead of pursuing a single point estimate, a more principled approach involves adopting a Bayesian framework, where we specify a prior distribution on $x$ and characterize the uncertainty through posterior sampling of $p(x|y)$. However, in many practical inverse problems, the prior distribution is already high-dimensional and may contain multiple well-separated modes. Coupled with an ill-posed forward model and noisy observations, such complex priors often induce posteriors that are likewise high-dimensional and strongly multimodal. Consequently, traditional Markov Chain Monte Carlo (MCMC) methods (Neal et al., 2011; Welling & Teh, 2011; Cui et al., 2016) often struggle with sampling from these posterior distributions primarily due to metastability, *i.e.*, the difficulty in transitioning between distinct high-probability modes that are separated by regions of low probability.

To address these limitations, prior work has leveraged generative models like normalizing flows (NFs) (Asim et al., 2020; Hou et al., 2019; Zhang et al., 2021; Whang et al., 2021b;a; Hagemann et al., 2022) and generative adversarial networks (GANs) (Patel & Oberai, 2019; Bora et al., 2017) to model and sample from those high-dimensional and multimodal posterior distributions. Recently, Diffusion models (DMs) and probability flow-based models (Albergo et al., 2023b; Albergo & Vanden-Eijnden, 2022; Lipman et al., 2022; Liu et al., 2022b; Sohl-Dickstein et al., 2015; Ho et al., 2020; Song et al., 2020a; 2021a; Song & Ermon, 2019; Song et al., 2020b; Zhang et al., 2018a) have emerged as leading methods in modern generative modeling. These models generate samples from a high-dimensional target distribution $p_0$ by inverting a diffusion process that transforms the target distribution $\boldsymbol{x}_0 \sim p_0$ into a simple distribution $\boldsymbol{x}_T \sim p_T$ (typically Gaussian). The effectiveness of DMs has led to their adoption as prior distributions in inverse problems, spawning various DM-based posterior sampling methods (Chung et al., 2022; Song et al., 2023b; Wu et al., 2023; Cardoso et al., 2023; Dou & Song, 2024; Sun et al., 2024; Xu & Chi, 2024; Wu et al., 2024c; Bruna & Han, 2024). For a comprehensive review, we refer the readers to either Appendix A.1 or (Daras et al., 2024a). These methods can be categorized into two main approaches:

1. Methods that leverage Bayes' formula to construct a conditional diffusion model using a pre-trained score function associated with the prior distribution: Specifically, for any time $t \in [0, T]$, applying Bayes' formula $p_t(\boldsymbol{x}_t|\boldsymbol{y}) \propto p_t(\boldsymbol{x}_t)p_t(\boldsymbol{y}|\boldsymbol{x}_t)$ yields

$$\nabla_{\boldsymbol{x}_t} \log p_t(\boldsymbol{x}_t|\boldsymbol{y}) = \nabla_{\boldsymbol{x}_t} \log p_t(\boldsymbol{x}_t) + \nabla_{\boldsymbol{x}_t} \log p_t(\boldsymbol{y}|\boldsymbol{x}_t). \tag{1.1}$$

To implement this approach, one needs to evaluate the left-hand side of (1.1), which is known as the conditional score function and defines a reverse-time diffusion process from $p_T(\boldsymbol{x}_T|\boldsymbol{y})$ to $p_0(\boldsymbol{x}_0|\boldsymbol{y})$. The first term on the right-hand side is the score function from the pre-trained DM modeling the prior distribution. The second term requires evaluating an integral $p_t(\boldsymbol{y}|\boldsymbol{x}_t) = \int p(\boldsymbol{y}|\boldsymbol{x}_0)p_{0|t}(\boldsymbol{x}_0|\boldsymbol{x}_t)\mathrm{d}\boldsymbol{x}_0$ over all possible $\boldsymbol{x}_0$'s that could lead to $\boldsymbol{x}_t$ through the pre-trained DM, to address which methods in this category employ various approximations for $\nabla_{\boldsymbol{x}_t} \log p_t(\boldsymbol{y}|\boldsymbol{x}_t)$.

Among different methods belonging to this approach, one group of methods (Song et al., 2020b; Choi et al., 2021; Song et al., 2021b; Chung et al., 2022; Song et al., 2023b; Boys et al., 2023; Wu et al., 2023) makes simplifying assumptions, while others (Choi et al., 2021; Wang et al., 2022; Kawar et al., 2022; Rout et al., 2023) use empirically constructed updates without structured assumptions. These heuristic, problem-specific approximations might be inaccurate in certain scenarios. In particular, for the special case of linear inverse problems, we assume that it is modeled by $y = \boldsymbol{y}_0 = \boldsymbol{A}\boldsymbol{x}_0 + \boldsymbol{n}$ with $\boldsymbol{y}_0 \in \mathbb{R}^m$, $\boldsymbol{x}_0 \in \mathbb{R}^n$, $\boldsymbol{A} \in \mathbb{R}^{m \times n}$ and $\boldsymbol{n} \sim \mathcal{N}(\boldsymbol{0}, \kappa^2 \boldsymbol{I}_m)$. Consider a standard and widely used case of DMs, whose associated forward diffusion process is given by $\boldsymbol{x}_t = \boldsymbol{x}_0 + \sigma(t)\boldsymbol{w}$ with injected noise $\boldsymbol{w} \sim \mathcal{N}(0, \boldsymbol{I}_n)$. The corresponding forward diffusion process for the measurement is further denoted by $\boldsymbol{y}_t = \boldsymbol{y}_0 + \sigma(t)\boldsymbol{\eta}$, where $\boldsymbol{\eta} = \boldsymbol{A}\boldsymbol{w}$ is a transformed multivariate Gaussian distribution in $\mathbb{R}^m$. Then we have the following

examples of approximations to the term $\nabla_x \log p_t(\boldsymbol{y}|\boldsymbol{x}_t)$ used in existing work like Iterative Latent Variable Refinement (ILVR) (Choi et al., 2021) and Diffusion Posterior Sampling (DPS) (Chung et al., 2022):

$$\nabla_{\boldsymbol{x}_t} \log p_t(\boldsymbol{y}|\boldsymbol{x}_t) \approx -\frac{1}{\kappa^2} \left(\boldsymbol{A}^\intercal \boldsymbol{A}\right)^{-1} \boldsymbol{A}^\intercal \left(\boldsymbol{y}_t - \boldsymbol{A}\boldsymbol{x}_t\right), \tag{ILVR}$$

$$\nabla_{\boldsymbol{x}_t} \log p_t(\boldsymbol{y}|\boldsymbol{x}_t) \approx \frac{1}{\kappa^2}(\boldsymbol{I}_n + \sigma(t)^2 \nabla_{\boldsymbol{x}_t}^2 \log p_t(\boldsymbol{x}_t))^T A^T (\boldsymbol{y} - A\mathbb{E}[\boldsymbol{x}_0|\boldsymbol{x}_t]). \tag{DPS}$$

For a detailed explanation of the intuitions behind the approximations above, we refer the readers to Appendix A.3.1.

2. Approximation-free methods that integrate DMs with traditional posterior sampling methods: Examples include split Gibbs sampler (SGS) + DM methods (Xu & Chi, 2024; Wu et al., 2024c; Coeurdoux et al., 2024; Zheng et al., 2025), which are built upon the split Gibbs sampler for Bayesian inference (Vono et al., 2019; Pereyra et al., 2023), and sequential Monte Carlo (SMC) + DM methods (Wu et al., 2023; Cardoso et al., 2023; Dou & Song, 2024; Kelvinius et al., 2025; Skreta et al., 2025; Lee et al., 2025; Holderrieth et al., 2025; Achituve et al., 2025), which combine DMs with SMC (Liu, 2001; Chopin, 2002; Del Moral et al., 2006; Doucet et al., 2009; Del Moral, 2013; Moral, 2004) to obtain asymptotically consistent posterior samples.

We advance the second approach by introducing a novel ensemble-based *Approximation-Free Diffusion Posterior Sampler (AFDPS)*. Our method enhances the synergy between DMs and SMC methods, which use weighted particle ensembles and strategic resampling to approximate the posterior distribution. The key innovation stems from our principled utilization of pre-trained DMs for prior evolution and our derivation of the exact partial differential equation (PDE) governing the corresponding posterior evolution, which reveals fundamentally distinct dynamics compared to existing approaches. Leveraging the flexibility of our framework, we propose two different approaches based on SDE and ODE+Corrector, respectively. Through careful analysis of the discrepancy between the derived PDE dynamics and the time-reversal of the true diffusion process, we establish error bounds for our posterior sampling algorithm based on stochastic weighted particle method. In practice, our algorithm demonstrates versatile compatibility with various pre-trained diffusion models. Extensive experiments on various imaging inverse problems are provided to demonstrate the effectiveness of our method.

**Our Contributions.** We summarize our main contributions as follows:

- We propose a novel ensemble-based posterior sampling method that integrates sequential Monte Carlo with diffusion models to achieve **exact posterior sampling without heuristic approximations**, founded on rigorously derived and previously unexplored PDE dynamics.
- We provide comprehensive theoretical guarantees demonstrating that our ensemble-based algorithm, implemented via stochastic weighted particle methods, **converges asymptotically to the derived PDE dynamics**. We additionally derive **precise error bounds** relating posterior sampling accuracy to the quality of the pre-trained score function.
- We empirically evaluate our method on multiple imaging inverse problems using large-scale datasets like FFHQ-256 (Karras et al., 2019) and ImageNet-256 (Deng et al., 2009), showing that our method **achieves better reconstruction quality** over existing methods.

## 2 Preliminaries

In this section, we provide a quick overview of problem setup, basic concepts, and existing work related to solving Bayesian inverse problems (BIPs) with diffusion models.

### 2.1 Basics of Inverse Problems

In BIPs, we aim to recover a ground truth parameter $\boldsymbol{x}$ from measurements $\boldsymbol{y}$. The relationship between $\boldsymbol{x}$ and $\boldsymbol{y}$ is described by:

$$\boldsymbol{y} = \mathcal{A}(\boldsymbol{x}) + \boldsymbol{n}, \tag{2.1}$$

where $\boldsymbol{x} \in \mathbb{R}^n$, $\boldsymbol{y} \in \mathbb{R}^m$, $\mathcal{A} : \mathbb{R}^n \to \mathbb{R}^m$ is a differentiable forward operator (linear or nonlinear), and $\boldsymbol{n} \in \mathbb{R}^m$ represents measurement noise. Under the Bayesian framework, the posterior distribution we seek to sample from is:

$$p(\boldsymbol{x}|\boldsymbol{y}) \propto p_0(\boldsymbol{x})p(\boldsymbol{y}|\boldsymbol{x}) = p_0(\boldsymbol{x})\exp(-\mu_{\boldsymbol{y}}(\boldsymbol{x})), \tag{2.2}$$

where $p_0(\boldsymbol{x})$ denotes the prior distribution and $\mu_{\boldsymbol{y}}(\boldsymbol{x}) = -\log p(\boldsymbol{y}|\boldsymbol{x})$ is the negative log-likelihood function for a fixed observation $\boldsymbol{y}$.

Many practical inverse problems are ill-posed due to measurement noise and non-injective forward models, making unique solutions impossible to obtain. Traditional optimization-based methods often fail to capture the complex solution landscape, motivating the use of Bayesian formulations where posterior sampling methods can systematically account for uncertainty and explore multiple plausible solutions. For a comprehensive treatment of BIPs, we refer readers to (Stuart, 2010).

Deep generative models have emerged as powerful prior distributions that can capture complex solution spaces while remaining computationally tractable. Unlike traditional priors that rely on structural assumptions, these models effectively represent high-dimensional and multi-modal distributions given sufficient training data. In this work, we focus on diffusion models (DMs), which represent the current state-of-the-art in generative modeling with successful applications across physics (Cotler & Rezchikov, 2023; Habibi et al., 2024; Zhu et al., 2024c), chemistry (Xu et al., 2022; Alakhdar et al., 2024; Riesel et al., 2024), biology (Alamdari et al., 2023; Watson et al., 2023), computer vision (Rombach et al., 2022; Chan et al., 2024), and natural language processing (Li et al., 2022b).

## 2.2 Diffusion Models: the EDM Framework

We adopt the "Elucidating the design space of Diffusion Models (EDM)" framework from (Karras et al., 2022) to model prior distributions. The EDM framework provides a unified approach for the design of diffusion models by systematically analyzing noise schedules, sampling algorithms, and training objectives.

Building on the continuous formulation of diffusion models (Song et al., 2020b), the framework starts off with a forward diffusion process governed by the stochastic differential equation (SDE):

$$\mathrm{d}\boldsymbol{x}_s = F(s)\boldsymbol{x}_s\mathrm{d}s + G(s)\mathrm{d}\boldsymbol{w}_s. \tag{2.3}$$

where $(\boldsymbol{w}_s)_{s \geq 0}$ is a standard Brownian motion and $p_s$ denotes the distribution of $\boldsymbol{x}_s$, with $p_0$ being the prior distribution from (2.2). Following (Anderson, 1982), the corresponding reverse-time SDE is:

$$\mathrm{d}\breve{\boldsymbol{x}}_t = \left[-F(t)\breve{\boldsymbol{x}}_t + \tfrac{G(t)^2+V(t)^2}{2}\nabla_{\boldsymbol{x}}\log\breve{p}_t(\breve{\boldsymbol{x}}_t)\right]\mathrm{d}t + V(t)\mathrm{d}\boldsymbol{w}_t, \tag{2.4}$$

where $\breve{p}_0 = p_T$, $\breve{p}_T = p_0$, $\breve{*}_t$ denotes $*_{T-t}$, and $V : \mathbb{R} \to \mathbb{R}$ is a scalar-valued function. The score function $\nabla \log \breve{p}_t(\boldsymbol{x})$ is typically approximated by a neural network $\boldsymbol{\phi}_\theta(\boldsymbol{x}, t)$ trained via score matching (Hyvärinen & Dayan, 2005; Vincent, 2011). We use $\widehat{\breve{\boldsymbol{x}}}_t$ and $\widehat{\breve{p}}_t$ to denote the particle trajectory and its distribution when using the approximated score function $\boldsymbol{\phi}_\theta(\boldsymbol{x}, t)$, with $\widehat{\breve{p}}_0$ being an approximation of the distribution $p_T$ and $\widehat{\breve{p}}_T$ approximating the target distribution $p_0$.

The EDM framework reparameterizes the drift coefficient $F(t)$ and diffusion coefficient $G(t)$ using

$$s(t) := \exp\left(\int_0^t F(\xi)\mathrm{d}\xi\right) \quad \text{and} \quad \sigma(t) := \sqrt{\int_0^t \frac{G(\xi)^2}{s(\xi)^2}\mathrm{d}\xi},$$

yielding $F(t) = \frac{\dot{s}(t)}{s(t)}$ and $G(t) = s(t)\sqrt{2\dot{\sigma}(t)\sigma(t)}$. This reparameterization enables more accurate score estimation under appropriate choices of $s$ and $\sigma$, as demonstrated empirically in (Karras et al., 2022) and theoretically in (Wang et al., 2024). Also, the framework allows for different implementations based on the choice of diffusion coefficient $V$. Setting $V(t) = G(t) = s(t)\sqrt{2\dot{\sigma}(t)\sigma(t)}$ yields the SDE implementation:

$$\mathrm{d}\widehat{\breve{\boldsymbol{x}}}_t = \left[-\tfrac{\dot{s}(t)}{s(t)}\widehat{\breve{\boldsymbol{x}}}_t + 2s(t)^2\dot{\sigma}(t)\sigma(t)\boldsymbol{\phi}_\theta(\widehat{\breve{\boldsymbol{x}}}_t, t)\right]\mathrm{d}t + s(t)\sqrt{2\dot{\sigma}(t)\sigma(t)}\mathrm{d}\boldsymbol{w}_t. \tag{2.5}$$

Alternatively, setting $V(t) = 0$ yields the probability-flow ODE (PF-ODE) implementation:

$$\mathrm{d}\widehat{\widehat{\boldsymbol{x}}}_t = \left[ -\frac{\dot{s}(t)}{s(t)}\widehat{\widehat{\boldsymbol{x}}}_t + s(t)^2 \dot{\sigma}(t)\sigma(t)\boldsymbol{\phi}_\theta(\widehat{\widehat{\boldsymbol{x}}}_t, t) \right] \mathrm{d}t. \qquad (2.6)$$

In practice, it is common to focus on the cases when $p_T$ converges to some Gaussian distribution under the EDM framework. For instance, when $s(t) = 1$, we have that $p_T$ is given by the convolved distribution $p_0 * \mathcal{N}(0, \sigma^2(t)\boldsymbol{I}_n)$. We note that this is also the setting adopted in both the theoretical analysis and the experiments of this paper.

## 3 Methodology

In this section, we present the key derivation underlying our posterior sampling algorithm. Our approach can be interpreted as solving a high-dimensional PDE that governs posterior distribution evolution using either the (stochastic) weighted particle method (Degond & Mas-Gallic, 1989; Degond & Mustieles, 1990; Rjasanow & Wagner, 1996; Bossy & Talay, 1997; Talay & Vaillant, 2003; Raviart, 2006; Chertock, 2017) or the SMC method (Chopin, 2002; Del Moral et al., 2006; Doucet et al., 2009; Del Moral, 2013; Moral, 2004). Throughout the derivation, we assume the log-likelihood function $\mu_{\boldsymbol{y}}(\boldsymbol{x})$ is at least twice differentiable w.r.t. $\boldsymbol{x}$ for fixed $\boldsymbol{y}$. Details of both algorithmic variants are given in the pseudocode in subsection 3.2.

### 3.1 Algorithm Outline

Following the setting in Section 2, we assume the prior distribution $p(\boldsymbol{x})$ is represented by a DM under the EDM framework. Specifically, $p_0(\boldsymbol{x})$ is approximated by $\widehat{\widehat{p}}_T(\boldsymbol{x})$, obtained by simulating (2.5) or (2.6) from a Gaussian $\widehat{\widehat{p}}_0$. Fix some parametrized curve $\alpha_t = \alpha(t) : [0, T] \to [0, 1]$ with $\alpha_0 > 0$. We then define the associated time-dependent posterior distribution as:

$$\widehat{q}_{\alpha,\boldsymbol{y}}(\boldsymbol{x}, t) := \frac{\widehat{\widehat{p}}_t(\boldsymbol{x})e^{-\alpha_t \mu_{\boldsymbol{y}}(\boldsymbol{x})}}{\int_{\mathbb{R}^n} \widehat{\widehat{p}}_t(\boldsymbol{x})e^{-\alpha_t \mu_{\boldsymbol{y}}(\boldsymbol{x})}\mathrm{d}\boldsymbol{x}} := \frac{\widehat{Q}_{\alpha,\boldsymbol{y}}(\boldsymbol{x}, t)}{\widehat{Z}_{\alpha,\boldsymbol{y}}(t)}, \qquad (3.1)$$

where $\widehat{Q}_{\alpha,\boldsymbol{y}}(\boldsymbol{x}, t) = \widehat{\widehat{p}}_t(\boldsymbol{x})e^{-\alpha_t \mu_{\boldsymbol{y}}(\boldsymbol{x})}$ denotes the unnormalized posterior and $\widehat{Z}_{\alpha,\boldsymbol{y}}(t) = \int_{\mathbb{R}^n} \widehat{Q}_{\alpha,\boldsymbol{y}}(\boldsymbol{x}, t)\mathrm{d}\boldsymbol{x}$ represents the normalizing constant.

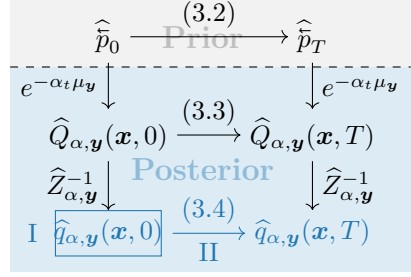

Figure 1: A roadmap for our posterior sampling method. I, II refers to the two stages of the proposed algorithm.

Our algorithm consists of the following two stages:

**Stage I: Sample from the initial distribution $\widehat{q}_{\alpha,\boldsymbol{y}}(\boldsymbol{x}, 0)$.** We first need to sample from the initial distribution $\widehat{q}_{\alpha,\boldsymbol{y}}(\boldsymbol{x}, 0) \propto \widehat{\widehat{p}}_0(\boldsymbol{x})e^{-\alpha_0 \mu_{\boldsymbol{y}}(\boldsymbol{x})}$, which is analogous to the likelihood step in (Wu et al., 2024c). This step is highly related to our choice of the parametrized curve $\alpha$, which in turn induces different distribution paths. In general, when the likelihood function $\mu_{\boldsymbol{y}}(\boldsymbol{x})$ is differentiable, we note that $\widehat{q}_{\alpha,\boldsymbol{y}}(\boldsymbol{x}, 0)$ can be approximately sampled by well-known gradient-based samplers like Metropolis Adjusted Langevin Algorithm (MALA) (Roberts & Stramer, 2002), Annealed Importance Sampling (AIS) (Neal, 2001), or more advanced methods (Lu et al., 2019b; Tan & Lu, 2023; Chen & Ying, 2024a; Lindsey et al., 2022). However, for certain special choices of the curve $\alpha$, we note that the initial distribution can also be sampled in an exact way. For instance, when $\alpha_t$ is an annealing-type curve with $\alpha_0 = 0$ and $\alpha_1 = 1$ (e.g., the linear schedule $\alpha_t = \frac{t}{T}$), the initial distribution $\widehat{q}_{\alpha,\boldsymbol{y}}(\boldsymbol{x}, 0) = \widehat{\widehat{p}}_0$ is Gaussian and can be sampled directly. Moreover, for linear BIPs with Gaussian noise, where $\mathcal{A} := \boldsymbol{A} \in \mathbb{R}^{m \times n}$ and $\boldsymbol{n} \sim \mathcal{N}(\boldsymbol{0}, \boldsymbol{\Sigma})$, the initial distribution is also normally distributed. Specifically, assuming $\widehat{\widehat{p}}_0 = \mathcal{N}(\boldsymbol{0}, \rho^2 \boldsymbol{I}_n)$, we have that $\widehat{q}_{\alpha,\boldsymbol{y}}(\boldsymbol{x}, 0)$ simplifies to:

$$\widehat{q}_{\alpha,\boldsymbol{y}}(\boldsymbol{x}, 0) \propto \exp\left( -\frac{\alpha_0}{2}(\boldsymbol{y} - \boldsymbol{A}\boldsymbol{x})^\mathsf{T}\boldsymbol{\Sigma}^{-1}(\boldsymbol{y} - \boldsymbol{A}\boldsymbol{x}) - \frac{1}{2\rho^2}\|\boldsymbol{x}\|_2^2 \right) = \mathcal{N}(\boldsymbol{\gamma}, \boldsymbol{\Lambda}^{-1}),$$

where $\boldsymbol{\Lambda} = \alpha_0 \boldsymbol{A}^\mathsf{T}\boldsymbol{\Sigma}^{-1}\boldsymbol{A} + \frac{1}{\rho^2}\boldsymbol{I}_n$ and $\boldsymbol{\gamma} = \alpha_0 \boldsymbol{\Lambda}^{-1}\boldsymbol{A}^\mathsf{T}\boldsymbol{\Sigma}^{-1}\boldsymbol{y}$.

**Stage II: Solve the PDE dynamics governing the posterior evolution.** Below we first derive the PDE dynamics $\left(\widehat{Q}_{\alpha,\boldsymbol{y}}(\boldsymbol{x},t)\right)_{t\in[0,T]}$ from $\left(\widehat{\overline{p}}_t\right)_{t\in[0,T]}$ based on the diffusion process (2.4), which consists of the following two steps and corresponds to the first phase in Figure 1 above.

1. The Fokker-Planck equation evolving from $\widehat{\overline{p}}_0$ to $\widehat{\overline{p}}_T$ is given by

$$\frac{\partial}{\partial t}\widehat{\overline{p}}_t = -\nabla_{\boldsymbol{x}}\cdot\left(\left(-F(t)\boldsymbol{x}+\frac{G(t)^2+V(t)^2}{2}\boldsymbol{\phi}_\theta(\boldsymbol{x},t)\right)\widehat{\overline{p}}_t\right)+\tfrac{1}{2}V(t)^2\Delta_{\boldsymbol{x}}\widehat{\overline{p}}_t. \tag{3.2}$$

2. Substituting $\widehat{\overline{p}}_t(\boldsymbol{x}) = \widehat{Q}_{\alpha,\boldsymbol{y}}(\boldsymbol{x},t)\exp(\alpha_t\mu_{\boldsymbol{y}})$ into (3.2) yields:

$$\begin{aligned}\frac{\partial}{\partial t}\widehat{Q}_{\alpha,\boldsymbol{y}} = &-\nabla_{\boldsymbol{x}}\cdot\left(\left(\widehat{\boldsymbol{H}}(\boldsymbol{x},t)-\alpha_t V(t)^2\nabla_{\boldsymbol{x}}\mu_{\boldsymbol{y}}\right)\widehat{Q}_{\alpha,\boldsymbol{y}}\right)+\tfrac{1}{2}V(t)^2\Delta_x\widehat{Q}_{\alpha,\boldsymbol{y}}\\ &+\left(U_{\alpha,\boldsymbol{y}}(\boldsymbol{x},t)-\alpha_t\widehat{\boldsymbol{H}}(\boldsymbol{x},t)^\intercal\nabla_{\boldsymbol{x}}\mu_{\boldsymbol{y}}-\alpha_t'\mu_{\boldsymbol{y}}\right)\widehat{Q}_{\alpha,\boldsymbol{y}},\end{aligned} \tag{3.3}$$

where $\widehat{\boldsymbol{H}}(\boldsymbol{x},t):=-F(t)\boldsymbol{x}+\frac{G(t)^2+V(t)^2}{2}\boldsymbol{\phi}_\theta(\boldsymbol{x},t)$ and $U_{\alpha,\boldsymbol{y}}(\boldsymbol{x},t):=\tfrac{1}{2}V(t)^2\left(\alpha_t^2\|\nabla_{\boldsymbol{x}}\mu_{\boldsymbol{y}}\|_2^2-\alpha_t\Delta_{\boldsymbol{x}}\mu_{\boldsymbol{y}}\right)$. We remark that the PDE above consists of three main components: the first term corresponds to the drift, the second to the diffusion, and the third to the reweighting term. On the one hand, the term $\widehat{\boldsymbol{H}}(x,t)$ above is exactly the original drift used in the Fokker Planck equation (3.2) that $(p_t)_{t\in[0,T]}$ satisfies. On the other hand, both the additional term $V(t)^2\nabla_{\boldsymbol{x}}\mu_{\boldsymbol{y}}$ in the drift and the function $U(\boldsymbol{x},t)$ in the reweighting term are only dependent on $V(t)$ and $\mu_{\boldsymbol{y}}$ from our derivation, which originates from the tilted term $e^{\mu_{\boldsymbol{y}}}$. For a complete derivation of (3.3), the readers may refer to Lemma B.1 in Appendix B.

3. By averaging the linear term on the RHS of (3.3) to "normalize" the PDE dynamics, which corresponds to the second phase in Figure 1, we obtain the PDE dynamics governing the evolution of $\widehat{q}_{\alpha,\boldsymbol{y}}(\boldsymbol{x},t)$ as follows:

---

**PDE Dynamics for Posterior Evolution**

$$\begin{aligned}\frac{\partial}{\partial t}\widehat{q}_{\alpha,\boldsymbol{y}} = &-\nabla_{\boldsymbol{x}}\cdot\left(\left(\widehat{\boldsymbol{H}}(\boldsymbol{x},t)-\alpha_t V(t)^2\nabla_{\boldsymbol{x}}\mu_{\boldsymbol{y}}\right)\widehat{q}_{\alpha,\boldsymbol{y}}\right)+\tfrac{1}{2}V(t)^2\Delta_x\widehat{q}_{\alpha,\boldsymbol{y}}\\ &+\left(W_{\alpha,\boldsymbol{y}}(\boldsymbol{x},t)-\int_{\mathbb{R}^n}W_{\alpha,\boldsymbol{y}}(\boldsymbol{z},t)\widehat{q}_{\alpha,\boldsymbol{y}}(\boldsymbol{z})\mathrm{d}\boldsymbol{z}\right)\widehat{q}_{\alpha,\boldsymbol{y}}.\end{aligned} \tag{3.4}$$

---

where $W_{\alpha,\boldsymbol{y}}(\boldsymbol{x},t):=U_{\alpha,\boldsymbol{y}}(\boldsymbol{x},t)-\alpha_t\widehat{\boldsymbol{H}}(\boldsymbol{x},t)^\intercal\nabla_{\boldsymbol{x}}\mu_{\boldsymbol{y}}-\alpha_t'\mu_{\boldsymbol{y}}(\boldsymbol{x})$ in the reweighting part above. We note that the normalization technique have also been used in recent works like (Skreta et al., 2025; Lee et al., 2025; Holderrieth et al., 2025). For a complete proof one may refer to Lemma B.2 in Appendix B. Moreover, one of our method's key novelties is that the derived PDE (3.4) includes a linear term proportional to $\widehat{q}_{\alpha,\boldsymbol{y}}$, which needs to be simulated via keeping track of an ensemble of weighted samples. Intuitively, particles with larger weights are more likely to lie in high-probability regions of the posterior distribution and thus correspond to reconstructions of higher quality. In fact, such procedure also admits an interpretation based on debiasing, as maintaining an ensemble of weighted particles essentially helps reduce simulation bias.

## 3.2 Posterior Sampling via Weighted Particles

We now present two ensemble-based posterior samplers within the SMC framework, which can also be interpreted as solving the PDE (3.4) numerically via (stochastic) weighted particles. Below we use $(\boldsymbol{x}_t,\beta_t)$ to denote the time-dependent position and weight associated with a single particle. The joint probability distribution of $(\boldsymbol{x}_t,\beta_t)$ is further denoted by $\gamma_t=\gamma_t(\boldsymbol{x},\beta)$.

**(Stochastic) Weighted Particle / Sequential Monte Carlo Methods.** As shown in Lemma B.4 of Appendix B, the posterior evolution (3.4) can be simulated via the following dynamics of a single weighted

particle:

$$\begin{cases} \mathrm{d}\boldsymbol{x}_t &= \left( \widehat{\boldsymbol{H}}(\boldsymbol{x}_t, t) - \alpha_t V(t)^2 \nabla_{\boldsymbol{x}} \mu_{\boldsymbol{y}}(\boldsymbol{x}_t) \right) \mathrm{d}t + V(t)\mathrm{d}\boldsymbol{w}_t, \\ \mathrm{d}\beta_t &= \left( U_{\alpha,\boldsymbol{y}}(\boldsymbol{x}_t, t) - \alpha_t \widehat{\boldsymbol{H}}(\boldsymbol{x}_t, t)^\mathsf{T} \nabla_{\boldsymbol{x}} \mu_{\boldsymbol{y}}(\boldsymbol{x}_t) - \alpha'_t \mu_{\boldsymbol{y}}(\boldsymbol{x}_t) \right) \beta_t \mathrm{d}t \\ &\quad - \left( \int_{\mathbb{R}^n} \left( U_{\alpha,\boldsymbol{y}}(\boldsymbol{x}, t) - \alpha_t \widehat{\boldsymbol{H}}(\boldsymbol{x}, t)^\mathsf{T} \nabla_{\boldsymbol{x}} \mu_{\boldsymbol{y}}(\boldsymbol{x}) - \alpha'_t \mu_{\boldsymbol{y}}(\boldsymbol{x}) \right) (P_\beta \gamma_t)(\boldsymbol{x})\mathrm{d}\boldsymbol{x} \right) \beta_t \mathrm{d}t, \end{cases} \quad (3.5)$$

where $P_\beta \gamma_t(\boldsymbol{x}) := \int_{\mathbb{R}} \beta \gamma_t(\boldsymbol{x}, \beta)\mathrm{d}\beta$ above denotes the weighted projection of $\gamma_t$ onto $\boldsymbol{x}$. To effectively approximate the integral in $P_\beta \gamma_t$, we then use the empirical measure $\gamma_t(\boldsymbol{x}, \beta) \approx \frac{1}{N} \sum_{i=1}^N \delta_{(\boldsymbol{x}_t^{(i)}, \beta_t^{(i)})}$ formed by $N$ weighted particles to approximate $\gamma_t(\boldsymbol{x}, \beta)$. This leads to the following joint dynamics for $\{(\boldsymbol{x}_t^{(i)}, \beta_t^{(i)})\}_{i=1}^N$:

---

**Weighted Particle Dynamics for Posterior Evolution**

$$\begin{cases} \mathrm{d}\boldsymbol{x}_t^{(i)} &= \left( \widehat{\boldsymbol{H}}(\boldsymbol{x}_t^{(i)}, t) - \alpha_t V(t)^2 \nabla_{\boldsymbol{x}} \mu_{\boldsymbol{y}}(\boldsymbol{x}_t^{(i)}) \right) \mathrm{d}t + V(t)\mathrm{d}\boldsymbol{w}_t^{(i)}, \\ \mathrm{d}\beta_t^{(i)} &= \left( U_{\alpha,\boldsymbol{y}}(\boldsymbol{x}_t^{(i)}, t) - \alpha_t \widehat{\boldsymbol{H}}(\boldsymbol{x}_t^{(i)}, t)^\mathsf{T} \nabla_{\boldsymbol{x}} \mu_{\boldsymbol{y}}(\boldsymbol{x}_t^{(i)}) - \alpha'_t \mu_{\boldsymbol{y}}(\boldsymbol{x}_t^{(i)}) \right) \beta_t^{(i)} \mathrm{d}t \\ &\quad - \left( \frac{1}{N} \sum_{j=1}^N \left( U_{\alpha,\boldsymbol{y}}(\boldsymbol{x}_t^{(j)}, t) - \alpha_t \widehat{\boldsymbol{H}}(\boldsymbol{x}_t^{(j)}, t)^\mathsf{T} \nabla_{\boldsymbol{x}} \mu_{\boldsymbol{y}}(\boldsymbol{x}_t^{(j)}) - \alpha'_t \mu_{\boldsymbol{y}}(\boldsymbol{x}_t^{(j)}) \right) \beta_t^{(j)} \right) \beta_t^{(i)} \mathrm{d}t, \end{cases} \quad (3.6)$$

---

with initial conditions $\boldsymbol{x}_0^{(i)} \sim \widehat{q}_{\alpha,\boldsymbol{y}}(\cdot, 0)$ and $\beta_0^{(i)} = 1$ for $i \in [N]$. The weighted projection equals $\frac{1}{N} \beta_t^{(j)}$ when $\boldsymbol{x} = \boldsymbol{x}_t^{(j)}$ for some $j$, and zero otherwise. While numerical discretization of (3.6) yields a prototypical sampling algorithm, the particle weights $\beta_t^{(i)}$ may diverge during simulation, reducing the ensemble's Effective Sample Size (ESS). To address this, we employ a resampling strategy commonly used in the SMC methods (Liu, 2001; Chopin, 2002; Del Moral et al., 2006; Doucet et al., 2009; Del Moral, 2013; Moral, 2004), whose detailed description is provided in Algorithm 1 below. Such resampling sub-routine essentially performs global moves by eliminating low-weight particles and duplicating high-weight ones, similar to the birth-death process used in (Moral, 2004; Lu et al., 2019b; Tan & Lu, 2023; Chen & Ying, 2024a; Lindsey et al., 2022). However, the resampling approach is computationally more efficient as the weight dynamics (3.6) can be parallelized.

---

**Algorithm 1:** Resampling Step

**Input:** Threshold $c \in (0, 1)$, weighted particles $\{(\boldsymbol{x}^{(j)}, \beta^{(j)})\}_{j=1}^N$

**Output:** Updated particles $\{(\widehat{\boldsymbol{x}}^{(j)}, \widehat{\beta}^{(j)})\}_{j=1}^N$

  **1 if** $ESS = \dfrac{\left( N^{-1} \sum_{j=1}^N \beta^{(j)} \right)^2}{N^{-1} \sum_{j=1}^N (\beta^{(j)})^2} < c$ **then**

  **2** $\quad$ Sample $\{\widehat{\boldsymbol{x}}^{(j)}\}_{j=1}^N$ with replacement from $\{\boldsymbol{x}^{(j)}\}_{j=1}^N$ with probability $\left\{ \dfrac{\beta^{(j)}}{\sum_{i=1}^N \beta^{(i)}} \right\}_{j=1}^N$;

  **3** $\quad$ $\widehat{\beta}^{(j)} \leftarrow 1$, for $j \in [N]$;

  **4 else**

  **5** $\quad$ $\{(\widehat{\boldsymbol{x}}^{(j)}, \widehat{\beta}^{(j)})\}_{j=1}^N \leftarrow \{(\boldsymbol{x}^{(j)}, \beta^{(j)})\}_{j=1}^N$;

  **6 end**

---

**SDE Approach (AFDPS-SDE).** We first consider the SDE implementation (2.5) of the diffusion model, where $V(t) = G(t) = s(t)\sqrt{2\dot{\sigma}(t)\sigma(t)}$. We directly discretize (3.6) with an Euler-Maruyama scheme and add Algorithm 1 as an adjustment step at the end of each iteration, which leads to Algorithm 2. We have omitted the averaging term, *i.e.*, the last line of (3.6) in Algorithm 2, in practical implementation since the update is the same for all particles and therefore cancels out when we normalize the weights. Such cancellation property also holds for the ODE approach presented below. For high-dimensional problems, we can further reduce the computational cost of both the SDE and the ODE approach via practical techniques like using a smaller ensemble, omitting the resampling step, and simply returning the particle with the highest weight as the best estimator, as discussed in subsection 5.1 and Appendix D.

**ODE+Corrector Approach (AFDPS-ODE).** Next, we consider an alternative implementation based on the probability flow ODE (2.6) by setting $V(t) = 0$. While this leads to the ODE dynamics (3.6), relying solely on deterministic evolution may not sufficiently explore the target distribution. To enhance exploration, we incorporate a stochastic corrector step inspired by predictor-corrector schemes in diffusion models (Song et al., 2020b; Chen et al., 2024c; Bradley & Nakkiran, 2024). The corrector uses the Unadjusted Langevin Algorithm (ULA, Algorithm 3) to draw samples from the intermediate posterior distribution $\widehat{q}_{\alpha,\boldsymbol{y}}(\boldsymbol{x}, t) \propto \widehat{\overline{p}}_t(\boldsymbol{x}) \exp(-\alpha_t \mu_{\boldsymbol{y}}(\boldsymbol{x}))$ at each timestep. The complete ODE+Corrector algorithm (Algorithm 4) is thus obtained by discretizing the probability flow ODE (3.6), and applying both resampling (Algorithm 1) and ULA correction (Algorithm 3) steps for adjustments.

---

**Algorithm 2:** Approximation-Free Diffusion Posterior Sampler via SDE (AFDPS-SDE)

> **Input:** Noisy observation $\boldsymbol{y}$, log-likelihood $\mu_{\boldsymbol{y}}(\cdot)$, functions $s(t)$ and $\sigma(t)$, parametrized curve $\alpha_t$, time grid $\{t_i\}_{i=0}^K$ with $t_0 = 0$ and $t_K = T$, thresholds $\{c_l\}_{l=1}^K$, score function $\boldsymbol{\phi}_\theta(\cdot, t)$, ensemble size $N$, initial weights $\beta_0^{(j)} = 1$ for $j \in [N]$.
>
> **Output:** Posterior approximation $\sum_{j=1}^N \beta_T^{(j)} \delta_{\boldsymbol{x}_T^{(j)}} / \sum_{j=1}^N \beta_T^{(j)}$.

**1** Draw $\{\boldsymbol{x}_0^{(i)}\}_{i=1}^N$ i.i.d. from $\widehat{q}_{\alpha,\boldsymbol{y}}(\cdot, 0)$ via Stage I samplers in Section 3.1;

**2** **for** $k = 0$ **to** $K - 1$ **do**

**3**     Draw $\{\xi_k^{(j)}\}_{j=1}^N$ i.i.d. from $\mathcal{N}(\boldsymbol{0}, \boldsymbol{I}_n)$;

**4**     **for** $j = 1$ **to** $N$ **do**

**5**
$$\widehat{\boldsymbol{x}}_{t_{k+1}}^{(j)} \leftarrow \left(1 - (t_{k+1} - t_k)\frac{\dot{s}(t_k)}{s(t_k)}\right) \boldsymbol{x}_{t_k}^{(j)} + s(t_k)\sqrt{2\dot{\sigma}(t_k)\sigma(t_k)(t_{k+1} - t_k)}\xi_k^{(j)}$$
$$+ 2(t_{k+1} - t_k)s(t_k)^2\dot{\sigma}(t_k)\sigma(t_k)\left(\boldsymbol{\phi}_\theta(\boldsymbol{x}_{t_k}^{(j)}, t_k) - \alpha_{t_k}\nabla_{\boldsymbol{x}}\mu_{\boldsymbol{y}}(\boldsymbol{x}_{t_k}^{(j)})\right);$$
$$\log \widehat{\beta}_{t_{k+1}}^{(j)} \leftarrow \log \beta_{t_{k+1}}^{(j)} - \alpha'_{t_k}(t_{k+1} - t_k)\mu_{\boldsymbol{y}}(\boldsymbol{x}_{t_k}^{(j)}) + (t_{k+1} - t_k)\alpha_{t_k}\frac{\dot{s}(t_k)}{s(t_k)}\nabla_{\boldsymbol{x}}\mu_{\boldsymbol{y}}(\boldsymbol{x}_{t_k}^{(j)})^\intercal\boldsymbol{x}_{t_k}^{(j)}$$
$$- 2(t_{k+1} - t_k)\alpha_{t_k}s(t_k)^2\dot{\sigma}(t_k)\sigma(t_k)\nabla_{\boldsymbol{x}}\mu_{\boldsymbol{y}}(\boldsymbol{x}_{t_k}^{(j)})^\intercal\boldsymbol{\phi}_\theta(\boldsymbol{x}_{t_k}^{(j)}, t_k)$$
$$+ (t_{k+1} - t_k)s(t_k)^2\dot{\sigma}(t_k)\sigma(t_k)\left(\alpha_{t_k}^2\|\nabla_{\boldsymbol{x}}\mu_{\boldsymbol{y}}(\boldsymbol{x}_{t_k}^{(j)})\|_2^2 - \alpha_{t_k}\Delta_{\boldsymbol{x}}\mu_{\boldsymbol{y}}(\boldsymbol{x}_{t_k}^{(j)})\right);$$

**6**     **end**

**7**     $\{(\boldsymbol{x}_{t_{k+1}}^{(j)}, \beta_{t_{k+1}}^{(j)})\}_{j=1}^N \leftarrow$ Algorithm 1$\left(c_{k+1}, \{(\widehat{\boldsymbol{x}}_{t_{k+1}}^{(j)}, \widehat{\beta}_{t_{k+1}}^{(j)})\}_{j=1}^N\right)$;

**8 end**

---

**Remark 3.1** (A brief comparison between the two proposed methods AFDPS-SDE and AFDPS-ODE)**.** *On the one hand, we remark that the ODE-based method typically requires tuning of the hyperparameters associated with the corrector, such as the step size and number of steps, whereas the SDE-based method does not. On the other hand, the ODE-based method is often more amenable to the design of higher-order numerical solvers with improved accuracy and faster sampling speed. More broadly, we note that the relative advantages of SDE versus ODE formulations remain an active research topic, and the readers may refer to (Song et al., 2020b; Cao et al., 2023; Chen et al., 2024c) for related studies.*

**Remark 3.2** (Connection with Feynman-Kac corrector (Skreta et al., 2025) and Guidance (Dhariwal & Nichol, 2021; Bradley & Nakkiran, 2024))**.** *Here we will discuss the main novelty of our method compared to a concurrent work (Skreta et al., 2025), which also proposed an ensemble-based sampler within the SMC framework. For a more detailed comparison with other SMC-based methods, the readers may refer to Appendix A.3.2. Specifically, the dynamics derived in our setting differ from those in Proposition D.5 of (Skreta et al., 2025), which is essentially the ODE case without correctors in our method. The key difference is the presence of a gradient term, $\nabla_{\boldsymbol{x}}\mu_{\boldsymbol{y}}$, in the dynamics of $\boldsymbol{x}_t$ (3.5), which is absent in their formulation. Such component, previously used in SGS + DM methods (Xu & Chi, 2024; Wu et al., 2024c) and in optimization-based denoising algorithms such as ADMM (Gabay & Mercier, 1976; Wang et al., 2008; Boyd et al., 2011; Sun et al., 2016; Chan et al., 2016; Ryu et al., 2019) and FISTA (Beck & Teboulle, 2009; Zhang & Ghanem, 2018; Xiang et al., 2021), is incorporated into our DM-based framework in a systematic way. The derivation illustrated in Figure 1 is shown to be essential for the method's empirical performance (*cf. *Section 5). A detailed comparison is given in Remark B.8 of Appendix B.*

*In contrast to prior work on guided diffusion sampling (Dhariwal & Nichol, 2021; Bradley & Nakkiran, 2024; Wu et al., 2024a; Chidambaram et al., 2024) and its extensions (Ho & Salimans, 2022; Bansal et al., 2023; Song et al., 2023c; He et al., 2023; Guo et al., 2024; Lu & Wang, 2024; Zheng et al., 2024; Ye et al., 2024), which augment single-particle dynamics with a gradient term such as $\nabla_{\boldsymbol{x}} \log p_t(\boldsymbol{y}|\boldsymbol{x}_t)$ or $\nabla_{\boldsymbol{x}} \log p_t(\boldsymbol{x}_t|\boldsymbol{y})$, our PDE-based derivation naturally yields the gradient term $\nabla_{\boldsymbol{x}} \mu_{\boldsymbol{y}}$ within a principled framework. Additionally, our formulation introduces a linear term that must be simulated via an ensemble of weighted particles, rather than from a single trajectory. Such ensemble-based structure allows us to integrate gradient-based guidance and diffusion sampling under the SMC framework in a unified way, resulting in improved empirical performance.*

---

**Algorithm 3:** Corrector Step

---

**Input:** Initialization $\widehat{x}_0$, time $t$, iterations $L$, stepsize $h$, log-likelihood $\mu_{\boldsymbol{y}}(\cdot)$, score function $\boldsymbol{\phi}_\theta(\cdot)$, parametrized curve $\alpha_t$.

**Output:** Sample $\widehat{x}_L \sim \widehat{q}_{\alpha,\boldsymbol{y}}(\boldsymbol{x}, t) \propto \widehat{\bar{p}}_t(\boldsymbol{x}) \exp(-\alpha_t \mu_{\boldsymbol{y}}(\boldsymbol{x}))$.

**1** Draw $\{\xi_l\}_{l=1}^L$ i.i.d. from $\mathcal{N}(\boldsymbol{0}, \boldsymbol{I}_n)$;

**2** **for** $l = 0$ **to** $L - 1$ **do**

**3**  $\quad \widehat{\boldsymbol{x}}_{l+1} \leftarrow \widehat{\boldsymbol{x}}_l + h\left(\boldsymbol{\phi}_\theta(\widehat{\boldsymbol{x}}_l, t) - \alpha_t \nabla_{\boldsymbol{x}} \mu_{\boldsymbol{y}}(\widehat{\boldsymbol{x}}_l)\right) + \sqrt{2h} \xi_{l+1}$;

**4** **end**

---

**Algorithm 4:** Approx.-Free Diffusion Posterior Sampler via ODE+Corrector (AFDPS-ODE)

---

**Input:** Noisy observation $\boldsymbol{y}$, log-likelihood $\mu_{\boldsymbol{y}}(\cdot)$, functions $s(t)$ and $\sigma(t)$, parametrized curve $\alpha_t$, time grid $\{t_i\}_{i=0}^K$ with $t_0 = 0$ and $t_K = T$, thresholds $\{c_l\}_{l=1}^K$, score function $\boldsymbol{\phi}_\theta(\cdot, t)$, corrector iterations $n_c$, stepsize $h_c$, ensemble size $N$, initial weights $\beta_0^{(j)} = 1$ for $j \in [N]$.

**Output:** Posterior approximation $\sum_{j=1}^N \beta_T^{(j)} \delta_{\boldsymbol{x}_T^{(j)}} / \sum_{j=1}^N \beta_T^{(j)}$.

**1** Draw $\{\boldsymbol{x}_0^{(i)}\}_{i=1}^N$ i.i.d. from $\widehat{q}_{\alpha,\boldsymbol{y}}(\cdot, 0)$ via Stage I samplers in Section 3.1;

**2** **for** $k = 0$ **to** $K - 1$ **do**

**3** $\quad$ **for** $j = 1$ **to** $N$ **do**

**4** $\qquad \widetilde{\boldsymbol{x}}_{t_{k+1}}^{(j)} \leftarrow \left(1 - (t_{k+1} - t_k)\frac{\dot{s}(t_k)}{s(t_k)}\right)\boldsymbol{x}_{t_k}^{(j)} + (t_{k+1} - t_k)s(t_k)^2 \dot{\sigma}(t_k)\sigma(t_k)\boldsymbol{\phi}_\theta(\boldsymbol{x}_{t_k}^{(j)}, t_k)$;

**5** $\qquad \widehat{\boldsymbol{x}}_{t_{k+1}}^{(j)} \leftarrow$ Algorithm 3 $\left(\widetilde{\boldsymbol{x}}_{t_{k+1}}^{(j)}, t_{k+1}, n_c, h_c, \mu_{\boldsymbol{y}}(\cdot), \boldsymbol{\phi}_\theta(\cdot, t)\right)$;

**6** $\qquad \log \widehat{\beta}_{t_{k+1}}^{(j)} \leftarrow \log \beta_{t_k}^{(j)} - \alpha'_{t_k}(t_{k+1} - t_k)\mu_{\boldsymbol{y}}(\boldsymbol{x}_{t_k}^{(j)}) + (t_{k+1} - t_k)\alpha_{t_k}\frac{\dot{s}(t_k)}{s(t_k)}\nabla_{\boldsymbol{x}} \mu_{\boldsymbol{y}}\left(\boldsymbol{x}_{t_k}^{(j)}\right)^\intercal \boldsymbol{x}_{t_k}^{(j)}$

$\qquad\qquad\qquad - (t_{k+1} - t_k)\alpha_{t_k}s(t_k)^2 \dot{\sigma}(t_k)\sigma(t_k)\nabla_{\boldsymbol{x}} \mu_{\boldsymbol{y}}\left(\boldsymbol{x}_{t_k}^{(j)}\right)^\intercal \boldsymbol{\phi}_\theta\left(\boldsymbol{x}_{t_k}^{(j)}, t_k\right)$;

**7** $\quad$ **end**

**8** $\quad \{(\boldsymbol{x}_{t_{k+1}}^{(j)}, \beta_{t_{k+1}}^{(j)})\}_{j=1}^N \leftarrow$ Algorithm 1 $\left(c_{k+1}, \{(\widehat{\boldsymbol{x}}_{t_{k+1}}^{(j)}, \widehat{\beta}_{t_{k+1}}^{(j)})\}_{j=1}^N\right)$;

**9** **end**

---

## 4  Theoretical Analysis

In this section, we present our theoretical results of the ensemble-based posterior samplers introduced in Section 3. Our analysis is conducted in continuous time, based on the weighted particle dynamics (3.4) and (3.6). The impact of numerical discretization, as implemented in Algorithm 2 and Algorithm 4, is not considered here and is left for future work. Without loss of generality, we focus on the backward SDE setting (2.5), specifically using $s(t) = 1$ and $\sigma(t) = t$. We begin by introducing several technical assumptions.

**Assumption 4.1** (Regularity of the log-likelihood)**.** *The log-likelihood function $\mu_{\boldsymbol{y}}$ is twice differentiable and lower bounded by some constant $C_{\boldsymbol{y}}^{(1)}$ depending only on the observation $\boldsymbol{y}$.*

**Assumption 4.2** (Bounded second moment)**.** *The prior distribution $p_0$ satisfies a second-moment bound: $\mathbb{E}_{p_0}\left[\|\boldsymbol{x}\|_2^2\right] \leq m_2^2$.*

**Assumption 4.3** (Score matching error). *The neural network estimator $\phi_\theta(\boldsymbol{x}, t)$ approximates the score function $\nabla_{\boldsymbol{x}} \log \breve{p}_t(\boldsymbol{x})$ with uniformly bounded error across $t \in [0, T]$:*

$$\int_{\mathbb{R}^n} \|\phi_\theta(\boldsymbol{x}, t) - \nabla_{\boldsymbol{x}} \log \breve{p}_t(\boldsymbol{x})\|_2^2 \, \breve{p}_t(\boldsymbol{x}) \mathrm{d}\boldsymbol{x} \leq \epsilon_{\boldsymbol{s}}^2. \tag{4.1}$$

On the one hand, Assumption 4.1 ensures the absence of singularities in the log-likelihood $\mu_{\boldsymbol{y}}$, which is a condition adopted in existing work on BIPs (Stuart, 2010) and satisfied by common noise models such as Gaussian and Poisson (with $C_{\boldsymbol{y}}^{(1)} = 0$). We note that this assumption also ensures the existence of the time-dependent constant $\widehat{Z}_{\alpha, \boldsymbol{y}}(t)$ introduced at the beginning of subsection 3.1. Specifically, for any time $t \in [0, T]$, a direct computation yields $\widehat{Z}_{\alpha, \boldsymbol{y}}(t) = \int \widehat{p}_t(\boldsymbol{x}) e^{-\alpha_t \mu_{\boldsymbol{y}}(\boldsymbol{x})} \mathrm{d}\boldsymbol{x} \leq \int \widehat{p}_t(\boldsymbol{x}) e^{-\alpha_t C_{\boldsymbol{y}}^{(1)}} \mathrm{d}\boldsymbol{x} \leq \max\{1, e^{-C_{\boldsymbol{y}}^{(1)}}\}$. On the other hand, Assumptions 4.2 and 4.3 are aligned with recent theoretical frameworks for diffusion models (Wang et al., 2024; Chen et al., 2022; 2023a; Benton et al., 2023; Chen et al., 2024c). In particular, Assumption 4.3 quantifies the approximation error due to neural network training and reflects the quality of the pre-trained score function. We now present our first main result, which quantifies the discrepancy between the true posterior $q_{\boldsymbol{y}, 0}$ and the distribution $\widehat{q}_{\alpha, \boldsymbol{y}, T}$ obtained by evolving our derived PDE dynamics (3.4) for time $T$. Below we use $\epsilon_I$ to denote an upper bound on the error incurred when sampling from the initial distribution $\widehat{q}_{\alpha, \boldsymbol{y}}(\cdot, 0)$, *i.e.*, $\mathrm{TV}(\widehat{q}_{\alpha, \boldsymbol{y}, 0}, \widehat{q}_{\alpha, \boldsymbol{y}}(\cdot, 0)) \leq \epsilon_I$.

> **Theorem 4.1** (Error Bound for Posterior Estimation within the Continuous-Time Dynamics). *When $s(t) = 1$, $\sigma(t) = t$ and Assumptions 4.1–4.3 all hold, the total variation (TV) distance between the approximate and true posterior distributions satisfies:*
>
> $$\mathrm{TV}(\widehat{q}_{\alpha, \boldsymbol{y}, T}, q_{\boldsymbol{y}, 0}) \leq C_{\boldsymbol{y}}^{(2)} \sqrt{\frac{m_2^2}{T^2} + T^2 \epsilon_{\boldsymbol{s}}^2} + C_{\boldsymbol{y}, T}^{(3)} \epsilon_I, \tag{4.2}$$
>
> *where $q_{\boldsymbol{y}, 0}(\boldsymbol{x}) \propto p_0(\boldsymbol{x}) \exp(-\mu_{\boldsymbol{y}}(\boldsymbol{x}))$ is the exact posterior, and $\widehat{q}_{\alpha, \boldsymbol{y}, T}$ is the solution to the posterior evolution dynamics (3.4) with initial distribution $\widehat{q}_{\alpha, \boldsymbol{y}, 0}$. The constant $C_{\boldsymbol{y}}^{(2)}$ depends only on the observation $\boldsymbol{y}$, while $C_{\boldsymbol{y}, T}^{(3)}$ on both $\boldsymbol{y}$ and the time $T$. In particular, when the initial error $\epsilon_I = 0$, i.e., $\widehat{q}_{\alpha, \boldsymbol{y}}(\cdot, 0)$ can be sampled exactly, optimizing the right-hand side yields the asymptotic bound $\mathrm{TV}(\widehat{q}_{\boldsymbol{y}, T}, q_{\boldsymbol{y}, 0}) \lesssim \sqrt{\epsilon_{\boldsymbol{s}}}$ when $T \asymp \sqrt{\epsilon_{\boldsymbol{s}}^{-1}}$.*

A detailed proof of Theorem 4.1 can be found in Section C.1. It essentially combines techniques from the theory of diffusion models (Chen et al., 2023a; Wang et al., 2024) and the well-posedness theory of Bayesian inverse problems, which is closely related to (Purohit et al., 2024, Theorem 4.1). The result of Theorem 4.1 reveals that under Assumptions 4.1–4.3, the discrepancy between the posterior distributions can be upper-bounded by a term proportional to the discrepancy between the prior distributions when the initial distribution can be sampled exactly. The upper bound further captures a tradeoff between the error of the forward process and the score matching error, which is controlled by the time horizon $T$. Next, we study the particle approximation to the PDE solution (3.4). In particular, we examine the convergence of the dynamics of the weighted particle ensemble (3.6) in the many-particle limit.

**Assumption 4.4** (Boundedness and Lipschitz continuity of $I$). *Define the function*

$$I(\boldsymbol{x}, t) := \alpha_t^2 \|\nabla_{\boldsymbol{x}} \mu_{\boldsymbol{y}}(\boldsymbol{x})\|_2^2 - \alpha_t \Delta_{\boldsymbol{x}} \mu_{\boldsymbol{y}}(\boldsymbol{x}) - 2\alpha_t \phi_\theta(\boldsymbol{x}, t)^{\mathsf{T}} \nabla_{\boldsymbol{x}} \mu_{\boldsymbol{y}}(\boldsymbol{x}) - \alpha_t' \mu_{\boldsymbol{y}}(\boldsymbol{x}).$$

*We assume that $I(\boldsymbol{x}, t)$ is uniformly bounded and Lipschitz continuous over $\mathbb{R}^n \times [0, T]$: $\max\{\|I\|_{L^\infty(\mathbb{R}^n \times [0, T])}, \mathrm{Lip}(I)\} \leq B_{\boldsymbol{y}}$, for some constant $B_{\boldsymbol{y}}$ depending only on $\boldsymbol{y}$.*

We note that function $I(\boldsymbol{x}, t)$ in Assumption 4.4 is highly related to the function $U_{\alpha, \boldsymbol{y}} - \alpha_t \widehat{\boldsymbol{H}}^{\mathsf{T}} \nabla_{\boldsymbol{x}} \mu_{\boldsymbol{y}} - \alpha_t' \mu_{\boldsymbol{y}}$ in (3.5) above, which depicts the evolution of the particle weights. Specifically, for the special case when $s(t) = 1$ and $\sigma(t) = t$, a direct computation yields

$$U_{\alpha, \boldsymbol{y}} - \alpha_t \widehat{\boldsymbol{H}}^{\mathsf{T}} \nabla_{\boldsymbol{x}} \mu_{\boldsymbol{y}} - \alpha_{\boldsymbol{y}}' \mu_{\boldsymbol{y}} = \frac{2t}{2} \left( \alpha_t^2 \|\nabla_{\boldsymbol{x}} \mu_{\boldsymbol{y}}\|_2^2 - \alpha_t \Delta_{\boldsymbol{x}} \mu_{\boldsymbol{y}} \right) - \frac{4t}{2} \alpha_t \phi_\theta^{\mathsf{T}} \nabla_{\boldsymbol{x}} \mu_{\boldsymbol{y}} = tI(\boldsymbol{x}, t).$$

Hence, we have that Assumption 4.4 essentially controls the particle weights during evolution and thereby mitigates weight degeneracy, which is analogous to Assumption 1 in (Domingo-Enrich et al., 2020). In practice, however, the particle weights are often controlled via the resampling step and ESS threshold in Algorithm 1. While we adopt this assumption for analytical tractability, relaxing it and developing a rigorous theory of the resampling step remain important future directions, which might link to existing theoretical studies (Lu et al., 2023; Chen et al., 2023b; Yan et al., 2024) on sampling algorithms that use birth-death dynamics or Fisher–Rao gradient flow.

---

**Theorem 4.2** (Convergence in the Many-Particle Limit). *When $s(t) = 1$, $\sigma(t) = t$ and Assumptions 4.1–4.4 all hold, the empirical distribution of the particle system converges to the solution of the posterior PDE. Specifically, for all $t \in [0, T]$,*

$$\lim_{N \to \infty} \mathbb{E}[\mathcal{W}_2^2(\gamma_t^N, \gamma_t)] = 0,$$

*where $\gamma_t$ is the law of a single weighted particle pair $(\boldsymbol{x}_t, \beta_t)$ governed by (3.5), such that the marginal $\widehat{q}_{\alpha, \boldsymbol{y}}(\cdot, t)$ is recovered via $P_\beta \gamma_t(\cdot) = \int_{\mathbb{R}} \beta \gamma_t(\cdot, \beta) \mathrm{d}\beta = \widehat{q}_{\alpha, \boldsymbol{y}}(\cdot, t)$, and $\gamma_t^N$ is the empirical measure of the N-particle system $\left\{ \left( \boldsymbol{x}_t^{(i)}, \beta_t^{(i)} \right) \right\}_{i=1}^N$ governed by (3.6).*

---

Theorem 4.2 establishes the mean-field consistency of the weighted ensemble approximation in the 2-Wasserstein sense. Its proof, which is presented in Appendix C.2, is based on results from propagation of chaos (Sznitman, 1991; Lacker, 2018). Together with Theorem 4.1, our theoretical results provide both rigorous guarantee for the accuracy of the continuum posterior approximation and justification of the proposed ensemble-based implementation.

Overall, we would like to emphasize that our theoretical analysis in this section provides a principled continuous-time framework for bounding the distributional discrepancy between the true posterior and the distribution induced by the generated samples. In particular, the bound consists of two components, where Theorem 4.1 quantifies the contribution from score-matching error and Theorem 4.2 captures the approximation error induced by using a finite number of particles. More broadly, these results also open the door to future work on consistency and stability with respect to time discretization. To the best of our knowledge, pursuing this direction will require more advanced tools from the numerical analysis of stochastic weighted particle/SMC methods. For a list of related works, we refer the readers to the end of section 6.

## 5 Experiments

In this section, we evaluate the empirical performance of our method on several BIPs in imaging. We emphasize that, although we develop a posterior-sampling algorithm here, our empirical evaluations primarily focus on the reconstruction quality under standard metrics, which is most relevant for practical imaging applications. This also aligns with the selection criterion adopted in our algorithm, which picks the best-performing sample among a collection of candidates. Due to space constraints, here we focus on the case of linear inverse problems in the main text. For implementation details and complete experimental results in the linear setting, as well as supplementary experiments, we refer the readers to Appendix D, E and F, respectively.

### 5.1 Linear Inverse Problems

**Problem Setting.** We consider the following four canonical linear inverse problems: Gaussian Deblurring (GD), Motion Deblurring (MD), Super Resolution (SR), and Box Inpaint (BI). In all these tasks, we assume that the observational noise is isotropic Gaussian with variance 0.2, *i.e.*, $\boldsymbol{n} \sim \mathcal{N}(\boldsymbol{0}, 0.2\boldsymbol{I}_m)$ in (2.1), a more challenging setting compared to the commonly used low-noise scenario with variance $2.5 \times 10^{-3}$ (Dou & Song, 2024; Wu et al., 2024c). Experiments are conducted on FFHQ-256 (Karras et al., 2019) and ImageNet-256 (Deng et al., 2009), two widely used datasets in imaging and vision.

**Baselines.** We compare our proposed algorithms with several state-of-the-art diffusion model-based posterior sampling methods:

- *DPS (Chung et al., 2022)*: a sampler that guides the pretrained DM with approximations of manifold-constrained gradients derived from the measurement likelihood.
- *DCDP (Li et al., 2024b)*: a framework alternating between optimization steps that ensure data consistency and pretrained DMs for posterior sampling.
- *SGS-EDM (Wu et al., 2024c)*: a split Gibbs sampler coupled with a DM for efficient posterior inference.
- *FK-Corrector (Skreta et al., 2025)*: an SMC-based sampler using Feynman-Kac formula to correct trajectories.
- *PF-SMC-DM (Dou & Song, 2024)*: a particle filtering framework combining SMC with diffusion models.

**Experimental Settings.** To ensure a fair comparison, we use the same checkpoints for the two pre-trained score functions provided in (Chung et al., 2022) and fix the number of function evaluations (NFE) to $2 \times 10^4$ across all methods. For ensemble-based approaches, the number of particles is set to $N = 10$. In the case of AFDPS-ODE (Algorithm 4), we reduce the number of particles to $N = 5$ to offset the additional computational cost from the corrector step, while maintaining the total NFE consistent with AFDPS-SDE (Algorithm 2). Within each ensemble, the particle with the largest weight is returned as the final estimator of the recovered image, which is analogous to the best-of-$N$ strategy. Here we choose the parametrized curve $\alpha_t$ to be constant, *i.e.*, $\alpha_t \equiv 1$. We evaluate reconstruction quality using two metrics: PSNR (Peak Signal-to-Noise Ratio), which quantifies pixel-level accuracy, and LPIPS (Learned Perceptual Image Patch Similarity) (Zhang et al., 2018b), which measures perceptual similarity. Both metrics are computed between the selected reconstruction sample and the ground truth image over a set of 100 validation images.

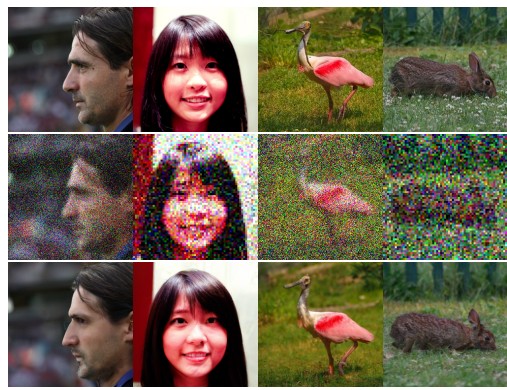

Figure 2: Visualization of posterior samples by AFDPS on linear inverse problems. **Upper:** Original; **Middle:** Noisy (Measurement); **Lower:** Reconstructed.

Table 1: Results on 4 linear inverse problems for 100 validation images from FFHQ-256.

| Method | Gaussian Deblurring | | Motion Deblurring | | Super Resolution | | Box Inpainting | |
|---|---|---|---|---|---|---|---|---|
| | PSNR (↑) | LPIPS (↓) | PSNR (↑) | LPIPS (↓) | PSNR (↑) | LPIPS (↓) | PSNR (↑) | LPIPS (↓) |
| DPS (Chung et al., 2022) | 22.57 | 0.2976 | 21.00 | 0.3280 | 19.09 | 0.5627 | 21.57 | 0.3245 |
| DCDP (Li et al., 2024b) | 24.77 | 0.2868 | 21.57 | 0.3487 | 21.23 | 0.5139 | 22.05 | 0.4525 |
| SGS-EDM (Wu et al., 2024c) | 24.78 | 0.2776 | 23.45 | 0.3009 | 22.41 | 0.3225 | 23.69 | 0.2301 |
| FK-Corrector (Skreta et al., 2025) | 21.22 | 0.4023 | 20.51 | 0.4275 | 20.67 | 0.4133 | 16.97 | 0.5490 |
| PF-SMC-DM (Dou & Song, 2024) | 23.00 | 0.3940 | **26.59** | 0.3435 | 18.92 | 0.5049 | 25.54 | 0.3391 |
| **AFDPS-SDE (Alg. 2)** | 24.83 | 0.2580 | 23.58 | **0.2869** | **22.96** | **0.3063** | 25.45 | 0.2084 |
| **AFDPS-ODE (Alg. 4)** | **24.98** | **0.2560** | 23.52 | 0.2905 | 21.47 | 0.3345 | **25.73** | **0.1969** |

Table 2: Results on 4 linear inverse problems for 100 validation images from ImageNet-256.

| Method | Gaussian Deblurring | | Motion Deblurring | | Super Resolution | | Box Inpainting | |
|---|---|---|---|---|---|---|---|---|
| | PSNR (↑) | LPIPS (↓) | PSNR (↑) | LPIPS (↓) | PSNR (↑) | LPIPS (↓) | PSNR (↑) | LPIPS (↓) |
| DPS (Chung et al., 2022) | 20.60 | 0.4351 | 20.46 | 0.5328 | 19.17 | 0.4940 | 22.70 | 0.3765 |
| DCDP (Li et al., 2024b) | 22.34 | 0.4821 | 20.59 | 0.5338 | 20.26 | 0.5597 | 21.67 | 0.4344 |
| SGS-EDM (Wu et al., 2024c) | 19.31 | 0.4807 | 20.54 | 0.4653 | 19.61 | 0.4986 | 21.42 | 0.4643 |
| FK-Corrector (Skreta et al., 2025) | 18.39 | 0.5973 | 18.34 | 0.6022 | 18.57 | 0.5887 | 16.28 | 0.7132 |
| PF-SMC-DM (Dou & Song, 2024) | 20.06 | 0.5927 | **23.91** | **0.4195** | 18.42 | 0.6462 | 21.34 | 0.4195 |
| **AFDPS-SDE (Alg. 2)** | 22.38 | **0.3925** | 19.46 | 0.4936 | **20.97** | **0.4643** | **23.15** | 0.3051 |
| **AFDPS-ODE (Alg. 4)** | **22.42** | 0.4633 | 21.54 | 0.4944 | 19.60 | 0.5634 | 22.76 | **0.2716** |

**Results.** The quantitative performance of our proposed methods - AFDPS-SDE (Algorithm 2) and AFDPS-ODE (Algorithm 4) - is presented in Table 1 for the FFHQ-256 dataset and Table 2 for the ImageNet-256

dataset. On FFHQ-256, both methods consistently demonstrate strong or highly competitive results across all evaluated inverse problems, frequently outperforming existing baselines in terms of both PSNR and LPIPS. The two variants show complementary strengths across different tasks, underscoring the benefit of incorporating both formulations. Similar trends are observed on the more diverse ImageNet-256 dataset, where both AFDPS methods continue to achieve robust and often superior performance. Qualitative examples are provided in Figure 2 and more in Appendix E, illustrating the visual quality of reconstructions across tasks with comparisons to baselines.

# 6  Discussion and Conclusion

In this paper, we introduced a new method for solving Bayesian inverse problems using diffusion models as the prior. Our method derives a novel PDE that exactly characterizes the exact posterior dynamics under an evolving diffusion prior, avoiding the heuristic approximations employed by previous methods and leading to better SMC-type algorithms in practice. Theoretically, we provide the error bounds of the posterior sampling algorithm in terms of the score function error, and justify the convergence of the ensemble method in the many-particle limit. Empirically, our method outperforms state-of-the-art diffusion-based solvers across a range of computational imaging tasks.

This work opens several promising directions for future research. One straight forward extension is to apply our proposed methods to other inverse problems arising in various fields with twice-differentiable log-likelihoods, including optics, medical imaging, video analytics, geoscience, astronomy, fluid dynamics, chemistry and biology (Sun et al., 2024; Wu et al., 2024c; Zheng et al., 2025; Jaganathan et al., 2016; Fienup, 1982; Candes et al., 2015b;a; Kantas et al., 2014; Daras et al., 2024b; Zhang et al., 2025b; Jing et al., 2024; Maddipatla et al., 2025; Sridharan et al., 2022; Hu et al., 2024).

Methodologically, our framework could also be extended to settings such as multi-marginal sampling (Albergo et al., 2023a; Lindsey, 2025), conditional sampling (Zhu et al., 2023), reward-guided sampling (Uehara et al., 2025), and other variants of DMs, such as latent diffusion models (LDMs) (Rombach et al., 2022; Song et al., 2023a), discrete diffusion models (Murata et al., 2024; Luan et al., 2025; Chu et al., 2025; Austin et al., 2021; Hoogeboom et al., 2021a;b; Meng et al., 2022; Sun et al., 2022; Richemond et al., 2022; Lou et al., 2023; Floto et al., 2023; Santos et al., 2023; Chen & Ying, 2024b; Ren et al., 2024a), flow matching (Zhang et al., 2024c), or to the general framework of denoising Markov model with variants like generator matching (Benton et al., 2024; Holderrieth et al., 2024; Ren et al., 2025c). Essentially speaking, Let $\bar{p}_T$ denote some high-dimensional prior parameterized by a diffusion model (e.g., a discrete or latent DM), with PDE dynamics $\mathcal{P}$ that maps a tractable base distribution $\bar{p}_0$ to $\bar{p}_T$. An analogous task is to sample from the tilted/posterior distribution $q_T \propto \bar{p}_T e^{\mu}$, where $\mu$ is some log-likelihood term or reward function. By defining $q_t \propto \bar{p}_t e^{\mu}$ for $t \in [0, T]$ and deriving the corresponding PDE $\mathcal{Q}$ governing $(q_t)_{t \in [0, T]}$ based on $\mathcal{P}$, an algorithm of similar type can be developed by simulating $\mathcal{Q}$ with a weighted-particle (SMC-type) method to obtain samples from $q_T$.

Theoretically, further work could explore numerical analysis of our method (Chen et al., 2022; 2023a; 2024c) or incorporate it with faster inference methods like parallel sampling (Shih et al., 2024; Tang et al., 2024; Cao et al., 2024; Selvam et al., 2024; Chen et al., 2024a; Gupta et al., 2024), high-order solvers (Karras et al., 2022; Lu et al., 2022b; Liu et al., 2022a; Lu et al., 2022a; Zheng et al., 2023; Li et al., 2024a; Wu et al., 2024b; Ren et al., 2025a) and their variants. In addition, it will also be of interest to investigate how one may theoretically refine the dependency of the asymptotic bound on the score matching error $\epsilon_s$ in Theorem 4.1.

# 7  Acknowledgments and Disclosure of Funding

Part of this research was conducted during Haoxuan Chen's internship at NEC Labs America. Haoxuan Chen, Yinuo Ren and Lexing Ying also acknowledge support of the National Science Foundation (NSF) under Award No. DMS-2208163. Haoxuan Chen would also like to thank Prof. Yifan Chen, Prof. Yiping Lu, Dr. Chenguang Duan and Qizheng Zhang for their generous help and valuable feedback.

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

## A  Further Discussion on Related Work and Notations

In this section, we provide additional discussion and context around our work through a comprehensive literature review and clarification of notations used throughout the paper.

### A.1  Related Work

In this subsection, we provide a more comprehensive overview of related work.

**Solving Inverse Problems via Machine Learning Techniques**  A wide body of work has tried applying machine learning (ML) based techniques to tackle inverse problems. In particular, one class of such ML-based methods deploy the Maximum a posteriori (MAP) approach by directly modeling the inverse mapping via some neural network. In the context of physical sciences, examples of work include (Yoon et al., 2018; Khoo & Ying, 2019; Fan & Ying, 2019a;b; 2020; Fournier et al., 2020; Sun & Demanet, 2020; 2021; Li et al., 2021; 2022a; Zhou et al., 2023; Fan & Ying, 2023; Molinaro et al., 2023; Melia et al., 2025). For a more detailed overview of methods belonging to such class, one may refer to (Arridge et al., 2019; Ying, 2022b). Similar methodologies (Zhang & Ghanem, 2018; Gilton et al., 2019; Xiang et al., 2021) have also been applied to inverse problems in computational imaging and computer vision. The second class of ML-based methods (Hou et al., 2019; Zhang et al., 2021; Whang et al., 2021a; Park et al., 2024; Tao et al., 2025; Dasgupta et al., 2025), however, employ a Bayesian approach by leveraging generative priors like normalizing flows and diffusion models. Such methods have been widely applied in various areas like medical imaging (Song et al., 2021b; Chung & Ye, 2022; Tu et al., 2025), cryo-electron microscopy (Kreis et al., 2022; Levy et al., 2024), PDE-constrained inverse problems (Jiang et al., 2025), sampling marginal densities (Lindsey, 2025), inverse scattering (Zhang et al., 2024a), traveltime tomography (Cao & Zhang,

2024), nonlinear data assimilation (Ding et al., 2024), inverse protein folding (Hsu et al., 2022; Zhu et al., 2024b), as well as fluid dynamics (Chen et al., 2024d; Xu et al., 2025; Molinaro et al., 2024). For a complete review of applying diffusion models to solve inverse problems, one may refer to (Daras et al., 2024a). Moreover, for the second class of methods that deploy a posterior sampling approach, recent work have also tried to combine diffusion models with existing sampling methods like SMC (Wu et al., 2023; Cardoso et al., 2023; Dou & Song, 2024; Albergo & Vanden-Eijnden, 2024; Chen et al., 2024b; Vargas et al., 2023), SGS (Xu & Chi, 2024; Wu et al., 2024c; Wang et al., 2025), parallel tempering (Zhang et al., 2025c) and ensemble Kalman filtering (Zheng et al., 2024). For methods using gradients of the log-likelihood in their algorithm design, we note that they also relate to guidance-based methods (Dhariwal & Nichol, 2021; Wu et al., 2024a; Chidambaram et al., 2024; Ho & Salimans, 2022; Bansal et al., 2023; Song et al., 2023c; He et al., 2023; Guo et al., 2024; Ye et al., 2024) proposed for conditional sampling.

**Gradient Flows for Sampling and Generative Modeling**   Gradient flow perspectives, particularly those based on the Wasserstein metric with foundational insights stemming from optimal transport and the JKO scheme (Jordan et al., 1998), have been extensively studied for both sampling and variational inference. Recent works in this direction include (Gao et al., 2019; Ansari et al., 2020; Fan et al., 2021; Lambert et al., 2022; Diao et al., 2023), with ongoing developments like (Wild et al., 2023; Shaul et al., 2023; Zhang & Katsoulakis, 2023; Cheng et al., 2024b; Yao et al., 2024; Choi et al., 2024; Zhu et al., 2024a). A few other work (Vidal et al., 2023; Cheng et al., 2024a; Xu et al., 2024; Xie & Cheng, 2025; Boffi et al., 2024; Kassraie et al., 2024) also discuss algorithms formulated via proximal operators and local-map learning strategies. Related developments in quantum Monte Carlo (QMC), particularly diffusion Monte Carlo (DMC) (Caffarel & Claverie, 1988a;b), are reviewed in (Gubernatis et al., 2016; Becca & Sorella, 2017) with further applications to quantum many-body problems discussed in (Lu & Wang, 2020).

**(Stochastic) Weighted Particle Methods and Wasserstein-Fisher-Rao Dynamics**   Weighted particle methods, such as those based on the birth-death process and Wasserstein–Fisher–Rao (WFR) distances (Kondratyev et al., 2016; Liero et al., 2018; Chizat et al., 2018)has motivated a series of studies on ensemble-based sampling dynamics (Lindsey et al., 2022; Lu et al., 2019b; Maurais & Marzouk, 2024; Gabrié et al., 2022; Tan & Lu, 2023; Chen & Ying, 2024a; Pathiraja & Wacker, 2024; Ren et al., 2025b) that have been applied to solving high-dimensional Bayesian inverse problems (Qu et al., 2024; Chen et al., 2024e) and PDEs (Han et al., 2020; Zhang et al., 2024b; Neklyudov et al., 2024; Chen et al., 2024f). These techniques have also been applied to multi-objective optimization (Ren et al., 2024b), density estimation via Gaussian mixtures (Chen et al., 2023b; Yan et al., 2024), and reinforcement learning and MDPs (Müller et al., 2024). Their connection to min-max optimization is explored in (Domingo-Enrich et al., 2020; Ying, 2022a; Lascu et al., 2024).

## A.2   Notations

We use $\nabla_{\boldsymbol{x}}, \nabla_{\boldsymbol{x}}\cdot$ and $\Delta_{\boldsymbol{x}}$ to denote the gradient, divergence, and Laplacian operators with respect to any fixed variable $\boldsymbol{x}$. The set of positive real numbers is denoted by $\mathbb{R}^+$. We further use $\delta$ for the Dirac delta function. For measuring distances between probability distributions, we use the Kullback-Leibler (KL) divergence $D_{\mathrm{KL}}$, Total Variation (TV) divergence TV, and Wasserstein-$p$ distance $\mathcal{W}_p$. The $l_2$ norm is denoted by $\|\cdot\|_2^2$. In general, for any real number $p \in \mathbb{R}^+ \cup \{\infty\}$, we use $\|\cdot\|_{L^p}$ to denote the $L^p$ norm.

## A.3   Further Discussion on Related Work

### A.3.1   Intuitive Explanations of the Posterior Score Approximations

This subsection is devoted to describing the intuitions behind the approximations adopted in existing work like ILVR (Choi et al., 2021) and DPS (Chung et al., 2022). We start with deriving the approximation used in ILVR first. In fact, the approximation in ILVR can be understood as a preconditioned version of the approximation used in (Song et al., 2020b), which we will derive and explain first. Based on the Bayes' formula $p_t(\boldsymbol{x}_t|\boldsymbol{y}) \propto p_t(\boldsymbol{x}_t)p_t(\boldsymbol{y}|\boldsymbol{x}_t)$, which implies $\nabla_{\boldsymbol{x}_t} \log p(\boldsymbol{y}|\boldsymbol{x}_t) = \nabla_{\boldsymbol{x}_t} \log p(\boldsymbol{x}_t|\boldsymbol{y}_0) - \nabla_{\boldsymbol{x}_t} \log p(\boldsymbol{x}_t)$, we have that it suffices to derive an approximation of the term $\nabla_{\boldsymbol{x}_t} \log p(\boldsymbol{x}_t|\boldsymbol{y}_0)$. Following the derivation in Appendix

I.4 of (Song et al., 2020b), we have that $\boldsymbol{y}_t$ is almost the same as $\boldsymbol{y}_0$ when $t$ is small, which gives us the approximation $p(\boldsymbol{x}_t|\boldsymbol{y}_t, \boldsymbol{y}_0) \approx p(\boldsymbol{x}_t|\boldsymbol{y}_t)$. Moreover, when $t$ is relatively large, we have that $\boldsymbol{x}_t$ is almost the same as Gaussian noise, which is away from $\boldsymbol{y}_0$ and again implies $p(\boldsymbol{x}_t|\boldsymbol{y}_t, \boldsymbol{y}_0) \approx p(\boldsymbol{x}_t|\boldsymbol{y}_t)$. Combining the two cases further yields the following approximation:

$$p(\boldsymbol{x}_t|\boldsymbol{y}_0) = \int p(\boldsymbol{x}_t|\boldsymbol{y}_t, \boldsymbol{y}_0)p(\boldsymbol{y}_t|\boldsymbol{y}_0)\mathrm{d}\boldsymbol{y}_t \approx \int p(\boldsymbol{x}_t|\boldsymbol{y}_t)p(\boldsymbol{y}_t|\boldsymbol{y}_0)\mathrm{d}\boldsymbol{y}_t \approx p\left(\boldsymbol{x}_t|\hat{\boldsymbol{y}}_t\right).$$

where $\hat{\boldsymbol{y}}_t$ above denotes a realization of $\boldsymbol{y}_t$, $i.e.$, $\hat{\boldsymbol{y}}_t$ can be treated as being sampled from $p(\boldsymbol{y}_t|\boldsymbol{y}_0)$. Then we can plug in the approximation $p(\boldsymbol{x}_t|\boldsymbol{y}_0) \approx p(\boldsymbol{x}_t|\hat{\boldsymbol{y}}_t)$ derived above and apply Bayes' formula again, which indicate

$$\nabla_{\boldsymbol{x}_t} \log p(\boldsymbol{y}|\boldsymbol{x}_t) \approx \nabla_{\boldsymbol{x}_t} \log p(\boldsymbol{x}_t|\hat{\boldsymbol{y}}_t) - \nabla_{x_t} \log p(x_t) = \nabla_{\boldsymbol{x}_t} \log \frac{p(\boldsymbol{x}_t|\hat{\boldsymbol{y}}_t)}{p(\boldsymbol{x}_t)} = \nabla_{x_t} \log p(\hat{\boldsymbol{y}}_t|x_t)$$

Furthermore, from our definition of $\boldsymbol{y}_t$ above we have that

$$\boldsymbol{y}_t - A\boldsymbol{x}_t = \boldsymbol{y}_0 + \sigma(t)\eta - A(\boldsymbol{x}_0 + \sigma(t)\boldsymbol{w}) = \boldsymbol{y}_0 - A\boldsymbol{x}_0 = \boldsymbol{n} \sim \mathcal{N}(0, \sigma^2 I_m) \Rightarrow p(\hat{\boldsymbol{y}}_t|x_t) \propto e^{-\frac{1}{2\kappa^2}\|\hat{\boldsymbol{y}}_t - Ax_t\|^2}$$

Then we can further deduce that

$$\nabla_{\boldsymbol{x}_t} \log p(\boldsymbol{y}|\boldsymbol{x}_t) \approx \nabla_{\boldsymbol{x}_t} \log p(\hat{\boldsymbol{y}}_t|\boldsymbol{x}_t) = -\frac{1}{2\kappa^2}\nabla_{\boldsymbol{x}_t}\|\hat{\boldsymbol{y}}_t - A\boldsymbol{x}_t\|^2 = \frac{1}{\kappa^2}A^T(\hat{\boldsymbol{y}}_t - A\boldsymbol{x}_t)$$

which is exactly the approximation adopted in (Song et al., 2020b). The ILVR approximation can then be interpreted as a preconditioned version of the approximation in (Song et al., 2020b) derived above, where $(A^T A)^{-1}$ is the preconditioner.

Then we proceed to derive the DPS-based approximation used in (Chung et al., 2022), which is mainly based on the following factorization:

$$p(\boldsymbol{y}|\boldsymbol{x}_t) = \int p(\boldsymbol{y}, \boldsymbol{x}_0|\boldsymbol{x}_t)\mathrm{d}\boldsymbol{x}_0 = \int p(\boldsymbol{y}|\boldsymbol{x}_0, \boldsymbol{x}_t)p(\boldsymbol{x}_0|\boldsymbol{x}_t)\mathrm{d}\boldsymbol{x}_0$$

$$= \int p(\boldsymbol{y}|\boldsymbol{x}_0)p(\boldsymbol{x}_0|\boldsymbol{x}_t)\mathrm{d}\boldsymbol{x}_0 = \mathbb{E}_{\boldsymbol{z} \sim p(\boldsymbol{x}_0|\boldsymbol{x}_t)}[p(\boldsymbol{y}|\boldsymbol{z})] \approx p\left(\boldsymbol{y}|\hat{\boldsymbol{x}}_0(t)\right)$$

where $\hat{\boldsymbol{x}}_0(t) = \mathbb{E}_{\boldsymbol{z} \sim p(\boldsymbol{x}_0|\boldsymbol{x}_t)}[\boldsymbol{z}] = \mathbb{E}[\boldsymbol{x}_0|\boldsymbol{x}_t]$ denotes the conditional expectation of $\boldsymbol{x}_0$ given $\boldsymbol{x}_t$. Specifically, we note that the second equation above follows from Bayes' formula and the third equation above is derived based on the fact that $\boldsymbol{y}$ is conditionally independent of $\boldsymbol{x}_t$ given $\boldsymbol{x}_0$. Moreover, the last equation above follows from the fact that the expectation of a function can be approximated via its evaluation at the expectation, $i.e.$, $\mathbb{E}[f(Z)] \approx f(\mathbb{E}[Z])$ for any random variable $Z$. Furthermore, applying Tweedie's formula to $\boldsymbol{x}_t = \boldsymbol{x}_0 + \sigma(t)\boldsymbol{w}$ yields that

$$\hat{\boldsymbol{x}}_0(t) = \mathbb{E}[\boldsymbol{x}_0|\boldsymbol{x}_t] = \boldsymbol{x}_t + \sigma(t)^2\nabla_{\boldsymbol{x}_t} \log p_t(\boldsymbol{x}_t)$$

Then we can apply the operator $\nabla_{\boldsymbol{x}_t} \log(\cdot)$ on both sides of the approximation $p(\boldsymbol{y}|\boldsymbol{x}_t) \approx p\left(\boldsymbol{y}|\hat{\boldsymbol{x}}_0(t)\right)$ above to deduce that

$$\nabla_{\boldsymbol{x}_t} \log p(\boldsymbol{y}|\boldsymbol{x}_t) \approx \nabla_{\boldsymbol{x}_t} \log p(\boldsymbol{y}|\hat{\boldsymbol{x}}_0(t)) = -\frac{1}{2\kappa^2}\nabla_{\boldsymbol{x}_t}\|\boldsymbol{y} - A\hat{\boldsymbol{x}}_0(t)\|^2$$

$$= \frac{1}{\kappa^2}(\nabla_{\boldsymbol{x}_t}(A\hat{\boldsymbol{x}}_0(t)))^T(\boldsymbol{y} - A\hat{\boldsymbol{x}}_0(t)) = \frac{1}{\kappa^2}(\nabla_{\boldsymbol{x}_t}\hat{\boldsymbol{x}}_0(t))^T A^T(\boldsymbol{y} - A\hat{\boldsymbol{x}}_0(t))$$

Plugging in the expression of $\hat{\boldsymbol{x}}_0(t)$ then gives us the DPS approximation of the posterior score function:

$$\nabla_{\boldsymbol{x}_t} \log p(\boldsymbol{y}|\boldsymbol{x}_t) \approx \frac{1}{\kappa^2}(\nabla_{\boldsymbol{x}_t}\hat{\boldsymbol{x}}_0(t))^T A^T(\boldsymbol{y} - A\hat{\boldsymbol{x}}_0(t)) = \frac{1}{\kappa^2}(\boldsymbol{I}_n + \sigma(t)^2\nabla^2_{\boldsymbol{x}_t} \log p_t(\boldsymbol{x}_t))^T A^T(\boldsymbol{y} - A\mathbb{E}[\boldsymbol{x}_0|\boldsymbol{x}_t])$$

For a complete derivation of the ILVR and DPS approximations described above, as well as a list of all related approximations, we refer the readers to Figure 1 and Subsection 3.1 of (Daras et al., 2024a).

### A.3.2 Comparison with SMC-based methods

Below we discuss the key novelties of our method compared to a class of existing works that solve inverse problems by combining SMC with diffusion-based priors, which can be summarized from the following two aspects. The first aspect is our derivation from a continuous-time perspective, which yields a PDE-based formulation of the algorithm. In contrast, prior approaches such as (Wu et al., 2023; Cardoso et al., 2023; Dou & Song, 2024) are typically defined via proposal kernels based on the discrete-time formulation. Our PDE-based viewpoint may facilitate future work on designing more efficient numerical schemes and extending the framework to incorporating with more complicated diffusion processes for the prior evolution, such as continuous-time Markov chains or Lévy processes. A more detailed discussion on these future directions is postponed to the end of section 6. The second aspect concerns the choice of likelihood functions, which determines the induced time-dependent distribution paths. Here we expand on the choices used in the representative examples (Wu et al., 2023; Cardoso et al., 2023; Dou & Song, 2024). Specifically, (Cardoso et al., 2023; Dou & Song, 2024) adopted the setting of particle filtering by generating a corresponding noisy observation $\boldsymbol{y}_t$ based on the observed $\boldsymbol{y}$ for any $\boldsymbol{x}_t$, which in turn leads to the following distribution path $l_t^{(1)}(\boldsymbol{x}) \propto \widehat{\overline{p}}_t(\boldsymbol{x})e^{-\mu_{\boldsymbol{y}_t}(\boldsymbol{x})}$. For a detailed description of how $\boldsymbol{y}_t$ is generated, the readers may refer to Appendix A.3.1 above. Furthermore, (Wu et al., 2023) considers a sequence of evolving posterior distributions of the form below:

$$l_t^{(2)}(\boldsymbol{x}) \propto \widehat{\overline{p}}_t(\boldsymbol{x})p(\boldsymbol{y}|\boldsymbol{x}_t = \boldsymbol{x}) = \widehat{\overline{p}}_t(\boldsymbol{x}) \left( \int p(\boldsymbol{y}|\boldsymbol{x}_0 = \boldsymbol{z})p(\boldsymbol{x}_0 = \boldsymbol{z}|\boldsymbol{x}_t = \boldsymbol{x})\mathrm{d}\boldsymbol{z} \right)$$

$$\propto \widehat{\overline{p}}_t(\boldsymbol{x}) \left( \int p(\boldsymbol{x}_0 = \boldsymbol{z}|\boldsymbol{x}_t = \boldsymbol{x})e^{-\frac{1}{2\sigma^2}\|\boldsymbol{y}-\mathcal{A}(\boldsymbol{z})\|^2}\mathrm{d}\boldsymbol{z} \right).$$

where Tweedie's formula is further applied to compute $p(\boldsymbol{x}_0|\boldsymbol{x}_t)$ and approximate the integral above. From the formulas of $l_t$'s listed above, we have that the two kinds of distribution paths used in (Wu et al., 2023; Cardoso et al., 2023; Dou & Song, 2024) are all different from $\hat{q}_{\alpha,\boldsymbol{y}}(\boldsymbol{x}, t) \propto \widehat{\overline{p}}_t(\boldsymbol{x})e^{-\alpha_t\mu_{\boldsymbol{y}}(\boldsymbol{x})}$ considered in our work. Empirically, numerical evidence are also provided in section 5 to demonstrate the advantages of our method.

## B Supplementary Proofs and Justifications for Section 3

In this section, we provide detailed proofs and justifications for claims listed in Section 3. We will use the shorthand notation $f_{\alpha,\boldsymbol{y}}(\boldsymbol{x}) = \exp(\alpha_t\mu_{\boldsymbol{y}}(\boldsymbol{x}))$ for the time-dependent likelihood factor.

**Lemma B.1.** *The PDE dynamics governing the evolution of the unnormalized posterior distribution* $\widehat{Q}_{\alpha,\boldsymbol{y}}(\boldsymbol{x}, t) : \mathbb{R}^n \times [0, T] \to \mathbb{R}^+$ *is given by*

$$
\begin{aligned}
\frac{\partial}{\partial t}\widehat{Q}_{\alpha,\boldsymbol{y}} = &- \nabla_{\boldsymbol{x}} \cdot \left( \left( \widehat{\boldsymbol{H}}(\boldsymbol{x}, t) - \alpha_t V(t)^2 \nabla_{\boldsymbol{x}}\mu_{\boldsymbol{y}} \right) \widehat{Q}_{\alpha,\boldsymbol{y}} \right) + \tfrac{1}{2}V(t)^2 \Delta_x \widehat{Q}_{\alpha,\boldsymbol{y}} \\
&+ \left( \frac{1}{2}V(t)^2 \left( \alpha_t^2 \|\nabla_{\boldsymbol{x}}\mu_{\boldsymbol{y}}\|_2^2 - \alpha_t \Delta_{\boldsymbol{x}}\mu_{\boldsymbol{y}} \right) - \alpha_t \widehat{\boldsymbol{H}}(\boldsymbol{x}, t)^{\mathsf{T}}\nabla_{\boldsymbol{x}}\mu_{\boldsymbol{y}} - \alpha_t'\mu_{\boldsymbol{y}} \right) \widehat{Q}_{\alpha,\boldsymbol{y}},
\end{aligned}
\tag{B.1}
$$

*where*

$$\widehat{\boldsymbol{H}}(\boldsymbol{x}, t) := -F(t)\boldsymbol{x} + \frac{G(t)^2 + V(t)^2}{2}\phi_\theta(\boldsymbol{x}, t) \tag{B.2}$$

*denotes the original drift in the prior diffusion.*

*Proof.* We begin by rewriting the PDE dynamics that needs simplification:

$$
\begin{aligned}
\left( \frac{\partial}{\partial t}\widehat{Q}_{\alpha,\boldsymbol{y}} \right) f_{\alpha,\boldsymbol{y}} + (\alpha_t'\mu_{\boldsymbol{y}}f_{\alpha,\boldsymbol{y}})\widehat{Q}_{\alpha,\boldsymbol{y}} &= \frac{\partial}{\partial t}\left( \widehat{Q}_{\alpha,\boldsymbol{y}}f_{\alpha,\boldsymbol{y}} \right) \\
&= -\nabla_{\boldsymbol{x}} \cdot \left( \widehat{\boldsymbol{H}}(\boldsymbol{x}, t)\widehat{Q}_{\alpha,\boldsymbol{y}}f_{\alpha,\boldsymbol{y}} \right) + \frac{1}{2}V(t)^2\Delta_x(\widehat{Q}_{\alpha,\boldsymbol{y}}f_{\alpha,\boldsymbol{y}}),
\end{aligned}
\tag{B.3}
$$

where the first identity above follows from the product rule and the second identity above is derived by substituting $\widehat{p}_t = \widehat{Q}_{\alpha,\boldsymbol{y}} f_{\alpha,\boldsymbol{y}}$ into the Fokker-Planck PDE (3.2) satisfied by $\widehat{p}_t$. Rearranging the equation above further implies

$$\frac{\partial}{\partial t}\widehat{Q}_{\alpha,\boldsymbol{y}} = -\frac{1}{f_{\alpha,\boldsymbol{y}}}\nabla_{\boldsymbol{x}} \cdot \left(\widehat{\boldsymbol{H}}(\boldsymbol{x},t)\widehat{Q}_{\alpha,\boldsymbol{y}} f_{\alpha,\boldsymbol{y}}\right) + \frac{1}{2f_{\alpha,\boldsymbol{y}}}V(t)^2\Delta_x(\widehat{Q}_{\alpha,\boldsymbol{y}} f_{\alpha,\boldsymbol{y}}) - \alpha'_t\mu_{\boldsymbol{y}}\widehat{Q}_{\alpha,\boldsymbol{y}}. \tag{B.4}$$

Let $I_1$ and $I_2$ denote the first two terms on the right-hand side:

$$I_1 := -\frac{1}{f_{\alpha,\boldsymbol{y}}}\nabla_{\boldsymbol{x}} \cdot \left(\widehat{\boldsymbol{H}}(\boldsymbol{x},t)\widehat{Q}_{\alpha,\boldsymbol{y}} f_{\alpha,\boldsymbol{y}}\right), \ I_2 := \frac{1}{2f_{\alpha,\boldsymbol{y}}}V(t)^2\Delta_x(\widehat{Q}_{\alpha,\boldsymbol{y}} f_{\alpha,\boldsymbol{y}}). \tag{B.5}$$

Note that $\widehat{\boldsymbol{H}}(\boldsymbol{x},t) : \mathbb{R}^{n+1} \to \mathbb{R}^n$ is vector-valued, while both $\widehat{Q}_{\alpha,\boldsymbol{y}} : \mathbb{R}^{n+1} \to \mathbb{R}$ and $f_{\alpha,\boldsymbol{y}} : \mathbb{R}^n \to \mathbb{R}$ are scalar-valued. A direct computation shows that the first term $I_1$ simplifies to:

$$
\begin{aligned}
I_1 &= -\frac{1}{f_{\alpha,\boldsymbol{y}}}\nabla_{\boldsymbol{x}} \cdot \left(\widehat{\boldsymbol{H}}(\boldsymbol{x},t)\widehat{Q}_{\alpha,\boldsymbol{y}} f_{\alpha,\boldsymbol{y}}\right) \\
&= -\frac{1}{f_{\alpha,\boldsymbol{y}}}\left(\nabla_{\boldsymbol{x}} \cdot \left(\widehat{\boldsymbol{H}}(\boldsymbol{x},t)\right)\widehat{Q}_{\alpha,\boldsymbol{y}} f_{\alpha,\boldsymbol{y}} + \widehat{\boldsymbol{H}}(\boldsymbol{x},t)^{\mathsf{T}}\nabla_{\boldsymbol{x}}\left(\widehat{Q}_{\alpha,\boldsymbol{y}} f_{\alpha,\boldsymbol{y}}\right)\right) \\
&= -\nabla_{\boldsymbol{x}} \cdot \left(\widehat{\boldsymbol{H}}(\boldsymbol{x},t)\right)\widehat{Q}_{\alpha,\boldsymbol{y}} - \frac{1}{f_{\alpha,\boldsymbol{y}}}\widehat{\boldsymbol{H}}(\boldsymbol{x},t)^{\mathsf{T}}\left(\left(\nabla_{\boldsymbol{x}}\widehat{Q}_{\alpha,\boldsymbol{y}}\right)f_{\alpha,\boldsymbol{y}} + \widehat{Q}_{\alpha,\boldsymbol{y}}\left(\nabla_{\boldsymbol{x}} f_{\alpha,\boldsymbol{y}}\right)\right) \\
&= -\nabla_{\boldsymbol{x}} \cdot \left(\widehat{\boldsymbol{H}}(\boldsymbol{x},t)\right)\widehat{Q}_{\alpha,\boldsymbol{y}} - \widehat{\boldsymbol{H}}(\boldsymbol{x},t)^{\mathsf{T}}\nabla_{\boldsymbol{x}}\widehat{Q}_{\alpha,\boldsymbol{y}} - \alpha_t\left(\widehat{\boldsymbol{H}}(\boldsymbol{x},t)^{\mathsf{T}}\nabla_{\boldsymbol{x}}\mu_{\boldsymbol{y}}\right)\widehat{Q}_{\alpha,\boldsymbol{y}},
\end{aligned} \tag{B.6}
$$

where the last equality above follows from the fact that $\frac{1}{f_{\alpha,\boldsymbol{y}}}\nabla_{\boldsymbol{x}} f_{\alpha,\boldsymbol{y}} = \alpha_t\nabla_{\boldsymbol{x}}\mu_{\boldsymbol{y}}$ for $f_{\alpha,\boldsymbol{y}} = \exp(\alpha_t\mu_{\boldsymbol{y}})$. Similarly, expanding the Laplacian term $\Delta_{\boldsymbol{x}}(\widehat{Q}_{\alpha,\boldsymbol{y}} f_{\alpha,\boldsymbol{y}})$ allows us to simplify the second term $I_2$ as follows:

$$
\begin{aligned}
I_2 &= \frac{1}{2f_{\alpha,\boldsymbol{y}}}V(t)^2\left(\left(\Delta_{\boldsymbol{x}}\widehat{Q}_{\alpha,\boldsymbol{y}}\right)f_{\alpha,\boldsymbol{y}} + 2\left(\nabla_{\boldsymbol{x}}\widehat{Q}_{\alpha,\boldsymbol{y}}\right)^{\mathsf{T}}\nabla_{\boldsymbol{x}} f_{\alpha,\boldsymbol{y}} + \widehat{Q}_{\alpha,\boldsymbol{y}}\left(\Delta_{\boldsymbol{x}} f_{\alpha,\boldsymbol{y}}\right)\right) \\
&= \frac{1}{2}V(t)^2\Delta_{\boldsymbol{x}}\widehat{Q}_{\alpha,\boldsymbol{y}} + \alpha_t V(t)^2\left(\nabla_{\boldsymbol{x}}\widehat{Q}_{\alpha,\boldsymbol{y}}\right)^{\mathsf{T}}\nabla_{\boldsymbol{x}}\mu_{\boldsymbol{y}} + \frac{1}{2}V(t)^2\left(\alpha_t\Delta_{\boldsymbol{x}}\mu_{\boldsymbol{y}} + \alpha_t^2\|\nabla_{\boldsymbol{x}}\mu_{\boldsymbol{y}}\|_2^2\right)\widehat{Q}_{\boldsymbol{y}},
\end{aligned} \tag{B.7}
$$

where the last equality above follows from the fact that $\frac{1}{f_{\alpha,\boldsymbol{y}}}\Delta_{\boldsymbol{x}} f_{\alpha,\boldsymbol{y}} = \alpha_t\Delta_{\boldsymbol{x}}\mu_{\boldsymbol{y}} + \alpha_t^2\|\nabla_{\boldsymbol{x}}\mu_{\boldsymbol{y}}\|_2^2$ for $f_{\alpha,\boldsymbol{y}} = \exp(\alpha_t\mu_{\boldsymbol{y}})$. Summing the two expressions in (B.6) and (B.7) then yields

$$
\begin{aligned}
\frac{\partial}{\partial t}\widehat{Q}_{\alpha,\boldsymbol{y}} = I_1 + I_2 &= -\nabla_{\boldsymbol{x}} \cdot \left(\widehat{\boldsymbol{H}}(\boldsymbol{x},t)\right)\widehat{Q}_{\alpha,\boldsymbol{y}} + \left(\alpha_t V(t)^2\nabla_{\boldsymbol{x}}\mu_{\boldsymbol{y}} - \widehat{\boldsymbol{H}}(\boldsymbol{x},t)\right)^{\mathsf{T}}\nabla_{\boldsymbol{x}}\widehat{Q}_{\alpha,\boldsymbol{y}} \\
&\quad + \frac{1}{2}V(t)^2\Delta_{\boldsymbol{x}}\widehat{Q}_{\alpha,\boldsymbol{y}} + \left(\frac{1}{2}V(t)^2\left(\alpha_t\Delta_{\boldsymbol{x}}\mu_{\boldsymbol{y}} + \alpha_t^2\|\nabla_{\boldsymbol{x}}\mu_{\boldsymbol{y}}\|_2^2\right) - \alpha_t\widehat{\boldsymbol{H}}(\boldsymbol{x},t)^{\mathsf{T}}\nabla_{\boldsymbol{x}}\mu_{\boldsymbol{y}}\right)\widehat{Q}_{\boldsymbol{y}} - \alpha'_t\mu_{\boldsymbol{y}}\widehat{Q}_{\alpha,\boldsymbol{y}} \\
&= -\nabla_{\boldsymbol{x}} \cdot \left(\left(\widehat{\boldsymbol{H}}(\boldsymbol{x},t) - \alpha_t V(t)^2\nabla_{\boldsymbol{x}}\mu_{\boldsymbol{y}}\right)\widehat{Q}_{\boldsymbol{y}}\right) - \alpha_t V(t)^2\Delta_{\boldsymbol{x}}\mu_{\boldsymbol{y}}\widehat{Q}_{\boldsymbol{y}} \\
&\quad + \frac{1}{2}V(t)^2\Delta_{\boldsymbol{x}}\widehat{Q}_{\alpha,\boldsymbol{y}} + \left(\frac{1}{2}V(t)^2\left(\alpha_t^2\|\nabla_{\boldsymbol{x}}\mu_{\boldsymbol{y}}\|_2^2 + \alpha_t\Delta_{\boldsymbol{x}}\mu_{\boldsymbol{y}}\right) - \alpha_t\widehat{\boldsymbol{H}}(\boldsymbol{x},t)^{\mathsf{T}}\nabla_{\boldsymbol{x}}\mu_{\boldsymbol{y}} - \alpha'_t\mu_{\boldsymbol{y}}\right)\widehat{Q}_{\alpha,\boldsymbol{y}}
\end{aligned} \tag{B.8}
$$

which is exactly the dynamics given in (B.1), as desired. $\qquad\square$

**Lemma B.2.** *Consider the following PDE dynamics governing the evolution of some unnormalized density* $\widehat{Q}(\boldsymbol{x},t) : \mathbb{R}^n \times [0,T] \to \mathbb{R}^+$

$$\frac{\partial}{\partial t}\widehat{Q}(\boldsymbol{x},t) = -\nabla_{\boldsymbol{x}} \cdot \left(K(\boldsymbol{x},t)\widehat{Q}(\boldsymbol{x},t)\right) + \zeta(t)\Delta_{\boldsymbol{x}}\widehat{Q}(\boldsymbol{x},t) + J(\boldsymbol{x},t)\widehat{Q}(\boldsymbol{x},t), \tag{B.9}$$

*where* $\zeta : [0,T] \to \mathbb{R}^+$ *and* $K, J : \mathbb{R}^d \times [0,T] \to \mathbb{R}$. *Then we consider the normalized density* $\widehat{q}(\boldsymbol{x},t) : \mathbb{R}^n \times [0,T] \to [0,1]$ *defined as below*

$$\widehat{q}(\boldsymbol{x},t) := \frac{\widehat{Q}(\boldsymbol{x},t)}{\int_{\mathbb{R}^n} \widehat{Q}(\boldsymbol{x},t)\mathrm{d}\boldsymbol{x}}, \ t \in [0,T]. \tag{B.10}$$

*The PDE dynamics governing the evolution of the normalized density $\widehat{q}(\boldsymbol{x}, t)$ is then given by*

$$\frac{\partial}{\partial t}\widehat{q}(\boldsymbol{x}, t) = -\nabla_{\boldsymbol{x}} \cdot (K(\boldsymbol{x}, t)\widehat{q}(\boldsymbol{x}, t)) + \zeta(t)\Delta_{\boldsymbol{x}}\widehat{q}(\boldsymbol{x}, t) + \left( J(\boldsymbol{x}, t) - \int_{\mathbb{R}^n} J(\boldsymbol{x}, t)\widehat{q}(\boldsymbol{x}, t)\mathrm{d}\boldsymbol{x} \right)\widehat{q}(\boldsymbol{x}, t). \tag{B.11}$$

*Proof.* By using $Z(t) := \int_{\mathbb{R}^n} \widehat{Q}(\boldsymbol{x}, t)\mathrm{d}\boldsymbol{x}$ to denote the normalizing constant for any $t \in [0, T]$, we can then compute the time derivative of $Z(t)$ by plugging in (B.9) as follows

$$\begin{aligned}
\frac{\partial}{\partial t}Z(t) &= \frac{\partial}{\partial t}\left( \int_{\mathbb{R}^n} \widehat{Q}(\boldsymbol{x}, t)\mathrm{d}\boldsymbol{x} \right) = \int_{\mathbb{R}^n} \left( \frac{\partial}{\partial t}\widehat{Q}(\boldsymbol{x}, t) \right)\mathrm{d}\boldsymbol{x} \\
&= \int_{\mathbb{R}^n} \left( -\nabla_{\boldsymbol{x}} \cdot \left( K(\boldsymbol{x}, t)\widehat{Q}(\boldsymbol{x}, t) \right) + \zeta(t)\Delta_{\boldsymbol{x}}\widehat{Q}(\boldsymbol{x}, t) + J(\boldsymbol{x}, t)\widehat{Q}(\boldsymbol{x}, t) \right)\mathrm{d}\boldsymbol{x} \\
&= \int_{\mathbb{R}^n} J(\boldsymbol{x}, t)\widehat{Q}(\boldsymbol{x}, t)\mathrm{d}\boldsymbol{x} + \int_{\mathbb{R}^n} \nabla_{\boldsymbol{x}} \cdot \left( \zeta(t)\nabla_{\boldsymbol{x}}\widehat{Q}(\boldsymbol{x}, t) - K(\boldsymbol{x}, t)\widehat{Q}(\boldsymbol{x}, t) \right)\mathrm{d}\boldsymbol{x} \\
&= \int_{\mathbb{R}^n} J(\boldsymbol{x}, t)\widehat{Q}(\boldsymbol{x}, t)\mathrm{d}\boldsymbol{x}.
\end{aligned} \tag{B.12}$$

Furthermore, we may rewrite the normalized density as $\widehat{q}(\boldsymbol{x}, t) = \frac{1}{Z(t)}\widehat{Q}(\boldsymbol{x}, t)$ and differentiate the expression with respect to $t$, which yields

$$\begin{aligned}
\frac{\partial}{\partial t}\widehat{q}(\boldsymbol{x}, t) &= \frac{1}{Z(t)^2}\left( \left( \frac{\partial}{\partial t}\widehat{Q}(\boldsymbol{x}, t) \right)Z(t) - \left( \frac{\partial}{\partial t}Z(t) \right)\widehat{Q}(\boldsymbol{x}, t) \right) \\
&= \frac{1}{Z(t)}\left( \frac{\partial}{\partial t}\widehat{Q}(\boldsymbol{x}, t) \right) - \frac{1}{Z(t)}\left( \frac{\partial}{\partial t}Z(t) \right)\left( \frac{1}{Z(t)}\widehat{Q}(\boldsymbol{x}, t) \right) \\
&= \frac{1}{Z(t)}\left( -\nabla_{\boldsymbol{x}} \cdot \left( K(\boldsymbol{x}, t)\widehat{Q}(\boldsymbol{x}, t) \right) + \zeta(t)\Delta_{\boldsymbol{x}}\widehat{Q}(\boldsymbol{x}, t) + J(\boldsymbol{x}, t)\widehat{Q}(\boldsymbol{x}, t) \right) \\
&\quad - \frac{1}{Z(t)}\left( \int_{\mathbb{R}^n} J(\boldsymbol{x}, t)\widehat{Q}(\boldsymbol{x}, t)\mathrm{d}\boldsymbol{x} \right)\widehat{q}(\boldsymbol{x}, t) \\
&= -\nabla_{\boldsymbol{x}} \cdot (K(\boldsymbol{x}, t)\widehat{q}(\boldsymbol{x}, t)) + \zeta(t)\Delta_{\boldsymbol{x}}\widehat{q}(\boldsymbol{x}, t) + \left( J(\boldsymbol{x}, t) - \int_{\mathbb{R}^n} J(\boldsymbol{x}, t)\widehat{q}(\boldsymbol{x}, t)\mathrm{d}\boldsymbol{x} \right)\widehat{q}(\boldsymbol{x}, t).
\end{aligned}$$

where the second last equality above follows from (B.9) and (B.12) the last equality is deduced from the definition of the normalized density $\widehat{q}(\boldsymbol{x}, t)$. This concludes our proof. $\square$

**Remark B.3.** *By setting*

$$K(\boldsymbol{x}, t) := \widehat{\boldsymbol{H}}(\boldsymbol{x}, t) - \alpha_t V(t)^2 \nabla_{\boldsymbol{x}}\mu_{\boldsymbol{y}}(\boldsymbol{x}), \quad \zeta(t) := \frac{1}{2}V(t)^2,$$

*and*

$$J(\boldsymbol{x}, t) := \frac{1}{2}V(t)^2 \left( \alpha_t^2\|\nabla_{\boldsymbol{x}}\mu_{\boldsymbol{y}}(\boldsymbol{x})\|_2^2 - \alpha_t\Delta_{\boldsymbol{x}}\mu_{\boldsymbol{y}}(\boldsymbol{x}) \right) - \alpha_t\widehat{\boldsymbol{H}}(\boldsymbol{x}, t)^{\mathsf{T}}\nabla_{\boldsymbol{x}}\mu_{\boldsymbol{y}}(\boldsymbol{x}) - \alpha_t'\mu_{\boldsymbol{y}},$$

*one can use Lemma B.2 to deduce (3.4) from (3.3).*

**Lemma B.4.** *Consider a single particle $(\boldsymbol{x}_t, \beta_t)$ governed by*

$$\begin{cases}
\mathrm{d}\boldsymbol{x}_t &= \left( \widehat{\boldsymbol{H}}(\boldsymbol{x}_t, t) - \alpha_t V(t)^2\nabla_{\boldsymbol{x}}\mu_{\boldsymbol{y}}(\boldsymbol{x}_t) \right)\mathrm{d}t + V(t)\mathrm{d}\boldsymbol{w}_t, \\
\mathrm{d}\beta_t &= \left( U_{\alpha,\boldsymbol{y}}(\boldsymbol{x}_t, t) - \alpha_t\widehat{\boldsymbol{H}}(\boldsymbol{x}_t, t)^{\mathsf{T}}\nabla_{\boldsymbol{x}}\mu_{\boldsymbol{y}}(\boldsymbol{x}_t) - \alpha_t'\mu_{\boldsymbol{y}}(\boldsymbol{x}_t) \right)\beta_t\mathrm{d}t \\
&\quad - \left( \int_{\mathbb{R}^n} \left( U_{\alpha,\boldsymbol{y}}(\boldsymbol{x}_t, t) - \alpha_t\widehat{\boldsymbol{H}}(\boldsymbol{x}_t, t)^{\mathsf{T}}\nabla_{\boldsymbol{x}}\mu_{\boldsymbol{y}}(\boldsymbol{x}_t) - \alpha_t'\mu_{\boldsymbol{y}}(\boldsymbol{x}_t) \right)\left( P_\beta\gamma_t \right)(\boldsymbol{x})\mathrm{d}\boldsymbol{x} \right)\beta_t\mathrm{d}t,
\end{cases} \tag{B.13}$$

*with initial condition $\boldsymbol{x}_0 = \boldsymbol{x}^*$ and $\beta_0 = 1$, where $\boldsymbol{x}^*$ is sampled from the initial posterior distribution $\widehat{q}_{\boldsymbol{y}}(\boldsymbol{x}, 0)$, $(\boldsymbol{w}_t)_{t \geq 0}$ is a standard Brownian motion in $\mathbb{R}^n$, $\gamma_t(\boldsymbol{x}, \beta)$ denotes the joint probability distribution of $(\boldsymbol{x}_t, \beta_t)$ on $\mathbb{R}^n \times \mathbb{R}$,*

$$P_\beta\gamma_t(\boldsymbol{x}) := \int_{\mathbb{R}} \beta\gamma_t(\boldsymbol{x}, \beta)\mathrm{d}\beta$$

denotes the weighted projection of $\gamma_t$ onto $\boldsymbol{x}$, and $U_{\alpha,\boldsymbol{y}}(\boldsymbol{x}_t, t) := \frac{1}{2}V(t)^2 \left( \alpha_t^2 \|\nabla_{\boldsymbol{x}}\mu_{\boldsymbol{y}}\|_2^2 - \alpha_t \Delta_{\boldsymbol{x}}\mu_{\boldsymbol{y}} \right)$. Below we further define

$$
\begin{aligned}
W_{\alpha,\boldsymbol{y}}(\boldsymbol{x}, t) &:= \frac{1}{2}V(t)^2 \left( \alpha_t^2 \|\nabla_{\boldsymbol{x}}\mu_{\boldsymbol{y}}\|_2^2 - \alpha_t \Delta_{\boldsymbol{x}}\mu_{\boldsymbol{y}} \right) - \alpha_t \widehat{\boldsymbol{H}}(\boldsymbol{x}, t)^\intercal \nabla_{\boldsymbol{x}}\mu_{\boldsymbol{y}} - \alpha_t'\mu_{\boldsymbol{y}}(\boldsymbol{x}) \\
&= U_{\alpha,\boldsymbol{y}}(\boldsymbol{x}_t, t) - \alpha_t \widehat{\boldsymbol{H}}(\boldsymbol{x}, t)^\intercal \nabla_{\boldsymbol{x}}\mu_{\boldsymbol{y}} - \alpha_t'\mu_{\boldsymbol{y}}(\boldsymbol{x}).
\end{aligned}
\tag{B.14}
$$

Then we have that $P_\beta\gamma_t(\boldsymbol{x}) = \widehat{q}_{\boldsymbol{y}}(\boldsymbol{x}, t)$ for any $\boldsymbol{x} \in \mathbb{R}^n$ and $t \in [0, T]$, i.e. $P_\beta\gamma_t(\cdot)$ solves the following PDE:

$$
\begin{aligned}
\frac{\partial}{\partial t}\widehat{q}_{\alpha,\boldsymbol{y}} =& -\nabla_{\boldsymbol{x}} \cdot \left( \left( \widehat{\boldsymbol{H}}(\boldsymbol{x}, t) - \alpha_t V(t)^2 \nabla_{\boldsymbol{x}}\mu_{\boldsymbol{y}} \right) \widehat{q}_{\alpha,\boldsymbol{y}} \right) + \tfrac{1}{2}V(t)^2 \Delta_x \widehat{q}_{\alpha,\boldsymbol{y}} \\
&+ \left( W_{\alpha,\boldsymbol{y}}(\boldsymbol{x}, t) - \int_{\mathbb{R}^n} W_{\alpha,\boldsymbol{y}}(\boldsymbol{z}, t)\widehat{q}_{\alpha,\boldsymbol{y}}(\boldsymbol{z})\mathrm{d}\boldsymbol{z} \right) \widehat{q}_{\alpha,\boldsymbol{y}}.
\end{aligned}
\tag{B.15}
$$

The main idea behind our proof of the Lemma above is to derive the PDE governing the evolution of the joint distribution $\gamma_t(\boldsymbol{x}, \beta)$ first, which then leads to a PDE for its weighted projection $P_\beta\gamma_t(\boldsymbol{x})$. Our derivation here is mainly based on the theory of semigroups.

**Definition B.5** (Two-Parameter Semigroup Operator). *Given fixed time $s > 0$, consider a single particle $(\boldsymbol{x}_t, \beta_t)$ with initial condition $(\boldsymbol{x}_s, \beta_s) = (\boldsymbol{x}^*, \beta^*)$ for any $t > s$. Then the corresponding semigroup operator $\mathcal{T}_{s,t}^{(\boldsymbol{x},\beta)}$ is defined via*

$$
\mathcal{T}_{s,t}^{(\boldsymbol{x},\beta)}\phi(\boldsymbol{x}^*, \beta^*) := \mathbb{E}\left[ \phi(\boldsymbol{x}_t, \beta_t) \mid (\boldsymbol{x}_s, \beta_s) = (\boldsymbol{x}^*, \beta^*) \right],
\tag{B.16}
$$

*where $\phi : \mathbb{R}^n \times \mathbb{R} \to \mathbb{R}$ above denotes an arbitrary test function. For the special case when $s = 0$, we write $\mathcal{T}_{0,t}^{(\boldsymbol{x},\beta)} = \mathcal{T}_t^{(\boldsymbol{x},\beta)}$.*

**Definition B.6** (Time-Dependent Infinitesimal Generator). *Let $\mathbb{I}$ be the identity operator. Then for any fixed time $s > 0$ and suitable test function $\phi$, the infinitesimal generator $\mathcal{L}_s^{(\boldsymbol{x},\beta)}$ associated with the semigroup $\mathcal{T}_{s,t}^{(\boldsymbol{x},\beta)}$ is defined by*

$$
\mathcal{L}_s^{(\boldsymbol{x},\beta)}\phi\left(\boldsymbol{x}^*, \beta^*\right) := \lim_{\Delta s \to 0^+} \frac{1}{\Delta s}\left( \mathcal{T}_{s,s+\Delta s}^{(\boldsymbol{x},\beta)}\phi(\boldsymbol{x}^*, \beta^*) - \phi(\boldsymbol{x}^*, \beta^*) \right).
\tag{B.17}
$$

Moreover, for any test function $\phi : \mathbb{R}^n \times \mathbb{R} \to \mathbb{R}$ and input $(\boldsymbol{x}^*, \beta^*)$, we have

$$
\begin{aligned}
\mathcal{T}_{t_1+t_2,t_1+t_2+t_3}^{(\boldsymbol{x},\beta)} \circ \mathcal{T}_{t_1,t_1+t_2}^{(\boldsymbol{x},\beta)}\phi(\boldsymbol{x}^*, \beta^*) &= \mathbb{E}\left[ \phi(\boldsymbol{x}_{t_1+t_2+t_3}, \beta_{t_1+t_2+t_3}) \mid (\boldsymbol{x}_{t_1}, \beta_{t_1}) = (\boldsymbol{x}^*, \beta^*) \right] \\
&= \mathcal{T}_{t_1,t_1+t_2+t_3}^{(\boldsymbol{x},\beta)}\phi(\boldsymbol{x}^*, \beta^*)
\end{aligned}
\tag{B.18}
$$

demonstrating that $\mathcal{T}_{t_1+t_2,t_1+t_2+t_3}^{(\boldsymbol{x},\beta)} \circ \mathcal{T}_{t_1,t_1+t_2}^{(\boldsymbol{x},\beta)} = \mathcal{T}_{t_1,t_1+t_2+t_3}^{(\boldsymbol{x},\beta)}$ for any time $t_1, t_2, t_3 > 0$.

Furthermore, combining (B.18) with the definition of the infinitesimal generator in (B.17), we can directly deduce the following equation for any time $0 < s < t$, input $(\boldsymbol{x}^*, \beta^*)$ and test function $\phi : \mathbb{R}^n \times \mathbb{R} \to \mathbb{R}$,

$$
\begin{aligned}
\frac{\partial}{\partial t}\mathcal{T}_{s,t}^{(\boldsymbol{x},\beta)}\phi &= \lim_{\Delta t \to 0^+} \frac{1}{\Delta t}\left( \mathcal{T}_{s,t+\Delta t}^{(\boldsymbol{x},\beta)} - \mathcal{T}_{s,t}^{(\boldsymbol{x},\beta)} \right)\phi\left(\boldsymbol{x}^*, \beta^*\right) = \lim_{\Delta t \to 0^+} \frac{1}{\Delta t}\mathcal{T}_{s,t}^{(\boldsymbol{x},\beta)} \circ \left( \mathcal{T}_{t,t+\Delta t}^{(\boldsymbol{x},\beta)} - \mathbb{I} \right)\phi\left(\boldsymbol{x}^*, \beta^*\right) \\
&= \mathcal{T}_{s,t}^{(\boldsymbol{x},\beta)} \circ \left( \lim_{\Delta t \to 0^+} \frac{1}{\Delta t}\left( \mathcal{T}_{t,t+\Delta t}^{(\boldsymbol{x},\beta)} - \mathbb{I} \right) \right)\phi\left(\boldsymbol{x}^*, \beta^*\right) = \left( \mathcal{T}_{s,t}^{(\boldsymbol{x},\beta)} \circ \mathcal{L}_t^{(\boldsymbol{x},\beta)} \right)\phi\left(\boldsymbol{x}^*, \beta^*\right)
\end{aligned}
\tag{B.19}
$$

which is essentially the forward Kolgomorov equation expressed in terms of semigroups and infinitesimal generators. Moreover, for any $d \in \mathbb{Z}^+$ and two functions $\varphi^{(1)}, \varphi^{(2)} : \mathbb{R}^d \to \mathbb{R}$, we use

$$
\left\langle \varphi^{(1)}, \varphi^{(2)} \right\rangle_{L^2(\mathbb{R}^d)} := \int_{\mathbb{R}^d} \varphi^{(1)}(\boldsymbol{x})\varphi^{(2)}(\boldsymbol{x})\mathrm{d}\boldsymbol{x}
$$

to denote the inner product between $\varphi^{(1)}$ and $\varphi^{(2)}$. Should no confusion arise, we omit the subscript $L^2(\mathbb{R}^d)$ in all derivations below.

**Proposition B.7.** *The joint distribution $\gamma_t = \gamma_t(\boldsymbol{x}, \beta)$ satisfies the following PDE:*

$$\frac{\partial}{\partial t}\gamma_t = -\nabla_{\boldsymbol{x}} \cdot (\boldsymbol{K}_{\alpha,t}\gamma_t) - \frac{\partial}{\partial \beta}(b_{\alpha,t}\gamma_t) + \frac{1}{2}V(t)^2 \Delta_{\boldsymbol{x}}\gamma_t, \tag{B.20}$$

*with initial condition $\gamma_0(\boldsymbol{x}, \beta) = \widehat{q}_{\alpha,\boldsymbol{y}}(\boldsymbol{x}, 0) \times \delta_{\beta=1}$, where the two functions $\boldsymbol{K}_{\alpha,t} : \mathbb{R}^d \to \mathbb{R}$ and $b_{\alpha,t} : \mathbb{R}^d \times \mathbb{R} \to \mathbb{R}$ correspond to the drift and reweighting terms in (3.5) and (B.13), i.e.,*

$$\boldsymbol{K}_{\alpha,t}(\boldsymbol{x}) = \widehat{\boldsymbol{H}}(\boldsymbol{x}, t) - \alpha_t V(t)^2 \nabla_{\boldsymbol{x}}\mu_{\boldsymbol{y}}(\boldsymbol{x}),$$

$$b_{\alpha,t}(\boldsymbol{x}, \beta) = \left( U_{\alpha,\boldsymbol{y}}(\boldsymbol{x}, t) - \alpha_t \widehat{\boldsymbol{H}}(\boldsymbol{x}, t)^{\mathsf{T}}\nabla_{\boldsymbol{x}}\mu_{\boldsymbol{y}}(\boldsymbol{x}) - \alpha_t'\mu_{\boldsymbol{y}}(\boldsymbol{x}) \right) \beta \tag{B.21}$$

$$- \left( \int_{\mathbb{R}^n} \left( U_{\alpha,\boldsymbol{y}}(\boldsymbol{x}^*, t) - \alpha_t \widehat{\boldsymbol{H}}(\boldsymbol{x}^*, t)^{\mathsf{T}}\nabla_{\boldsymbol{x}}\mu_{\boldsymbol{y}}(\boldsymbol{x}^*) - \alpha_t'\mu_{\boldsymbol{y}}(\boldsymbol{x}) \right) (P_\beta \gamma_t)(\boldsymbol{x}^*)\mathrm{d}\boldsymbol{x}^* \right) \beta.$$

*Proof.* We note that our proof here is mainly based on the weak formulation of PDEs. Specifically, for any fixed time $t$ and test function $\varphi : \mathbb{R}^n \times \mathbb{R} \to \mathbb{R}$,, integrating the function $\mathcal{T}_t^{(\boldsymbol{x},\beta)}\varphi$ over the initial joint distribution $\gamma_0(\boldsymbol{x}, \beta)$ yields

$$\left\langle \mathcal{T}_t^{(\boldsymbol{x},\beta)}\varphi, \gamma_0 \right\rangle = \int_{\mathbb{R}^n \times \mathbb{R}} \mathcal{T}_t^{(\boldsymbol{x},\beta)}\varphi(\boldsymbol{x}^*, \beta^*)\gamma_0(\boldsymbol{x}^*, \beta^*)\,\mathrm{d}\boldsymbol{x}^*\mathrm{d}\beta^*$$

$$= \int_{\mathbb{R}^n \times \mathbb{R}} \mathbb{E}\left[\varphi(\boldsymbol{x}_t, \beta_t) \mid (\boldsymbol{x}_0, \beta_0) = (\boldsymbol{x}^*, \beta^*)\right] \gamma_0(\boldsymbol{x}^*, \beta^*)\,\mathrm{d}\boldsymbol{x}^*\mathrm{d}\beta^* \tag{B.22}$$

$$= \int_{\mathbb{R}^n \times \mathbb{R}} \varphi(\boldsymbol{x}^*, \beta^*)\gamma_t(\boldsymbol{x}^*, \beta^*)\mathrm{d}\boldsymbol{x}^*\mathrm{d}\beta^* = \langle \varphi, \gamma_t \rangle.$$

We integrate on both sides of (B.19) over the initial joint distribution $\gamma_0(\boldsymbol{x}, \beta)$ and plug in (B.22), which gives us that for any test function $\varphi : \mathbb{R}^n \times \mathbb{R} \to \mathbb{R}$,

$$\left\langle \varphi, \frac{\partial}{\partial t}\gamma_t \right\rangle = \frac{\mathrm{d}}{\mathrm{d}t}\langle \varphi, \gamma_t \rangle = \frac{\mathrm{d}}{\mathrm{d}t}\left\langle \mathcal{T}_t^{(\boldsymbol{x},\beta)}\varphi, \gamma_0 \right\rangle = \left\langle \frac{\partial}{\partial t}\mathcal{T}_{0,t}^{(\boldsymbol{x},\beta)}\varphi, \gamma_0 \right\rangle$$

$$= \left\langle \mathcal{T}_{0,t}^{(\boldsymbol{x},\beta)} \circ \mathcal{L}_t^{(\boldsymbol{x},\beta)}\varphi, \gamma_0 \right\rangle = \left\langle \mathcal{T}_t^{(\boldsymbol{x},\beta)} \circ \mathcal{L}_t^{(\boldsymbol{x},\beta)}\varphi, \gamma_0 \right\rangle = \left\langle \mathcal{L}_t^{(\boldsymbol{x},\beta)}\varphi, \gamma_t \right\rangle. \tag{B.23}$$

To further simplify the term on the RHS above, we need to compute the explicit form of the infinitesimal generator defined in (B.17). In fact, applying Itô's formula to the joint SDE (3.5) yields the following identity for any test function $\varphi : \mathbb{R}^n \times \mathbb{R} \to \mathbb{R}$,

$$\mathrm{d}\varphi(\boldsymbol{x}_t, \beta_t) = \left( (\nabla_{\boldsymbol{x}}\varphi)^{\mathsf{T}}\boldsymbol{K}_{\alpha,t} + \frac{\partial\varphi}{\partial\beta}b_{\alpha,t} + \frac{1}{2}V(t)^2\operatorname{Tr}\left(\nabla_{\boldsymbol{x}}^2\varphi\right) \right)\mathrm{d}t + V(t)\left( (\nabla_{\boldsymbol{x}}\varphi)^{\mathsf{T}}\mathrm{d}\boldsymbol{w}_t \right), \tag{B.24}$$

where $(\boldsymbol{w}_t)_{t\geq 0}$ is a standard Brownian motion on $\mathbb{R}^n$ and the two functions. Taking expectation on both sides of (B.24) then yields the explicit expression of the infinitesimal generator for any test function $\varphi : \mathbb{R}^n \times \mathbb{R} \to \mathbb{R}$ as below:

$$\mathcal{L}_t^{(\boldsymbol{x},\beta)}\varphi = (\nabla_{\boldsymbol{x}}\varphi)^{\mathsf{T}}\boldsymbol{K}_{\alpha,t} + \frac{\partial\varphi}{\partial\beta}b_{\alpha,t} + \frac{1}{2}V(t)^2\Delta_{\boldsymbol{x}}\varphi. \tag{B.25}$$

Below we use $x_i$ and $\boldsymbol{K}_{\alpha,t,i}$ to denote the $i$-th component of $\boldsymbol{x}$ and $\boldsymbol{K}_{\alpha,t}$ for any $i \in [n]$. By substituting (B.25) into the RHS of (B.23), we obtain that for any test function $\varphi : \mathbb{R}^n \times \mathbb{R} \to \mathbb{R}$,

$$\left\langle \mathcal{L}_t^{(\boldsymbol{x},\beta)}\varphi, \gamma_t \right\rangle = \left\langle (\nabla_{\boldsymbol{x}}\varphi)^{\mathsf{T}}\boldsymbol{K}_{\alpha,t} + \frac{\partial\varphi}{\partial\beta}b_{\alpha,t} + \frac{1}{2}V(t)^2\Delta_{\boldsymbol{x}}\varphi, \gamma_t \right\rangle$$

$$= \sum_{i=1}^n \left\langle \frac{\partial\varphi}{\partial x_i}, \boldsymbol{K}_{\alpha,t,i}\gamma_t \right\rangle + \left\langle \frac{\partial\varphi}{\partial\beta}, b_{\alpha,t}\gamma_t \right\rangle + \frac{1}{2}V(t)^2 \sum_{i=1}^n \left\langle \frac{\partial^2\varphi}{\partial x_i^2}, \gamma_t \right\rangle$$

$$= \left\langle \varphi, -\sum_{i=1}^n \frac{\partial}{\partial x_i}(\boldsymbol{K}_{\alpha,t,i}\gamma_t) - \frac{\partial}{\partial\beta}(b_{\alpha,t}\gamma_t) + \frac{1}{2}V(t)^2\Delta_{\boldsymbol{x}}\gamma_t \right\rangle \tag{B.26}$$

$$= \left\langle \varphi, -\nabla_{\boldsymbol{x}} \cdot (\boldsymbol{K}_{\alpha,t}\gamma_t) - \frac{\partial}{\partial\beta}(b_{\alpha,t}\gamma_t) + \frac{1}{2}V(t)^2\Delta_{\boldsymbol{x}}\gamma_t \right\rangle,$$

where the second last equality above follows from integration by parts. Substituting the last expression in (B.26) above into (B.23) then gives us the weak form of the PDE associated with the joint distribution $\gamma_t$ in (B.20), which concludes the proof. □

*Proof of Lemma B.4.* By defining

$$\gamma_t^{\boldsymbol{P}}(\boldsymbol{x}) := P_\beta \gamma_t(\boldsymbol{x}) = \int_{\mathbb{R}} \beta \gamma_t(\boldsymbol{x}, \beta) \mathrm{d}\beta$$

to be the weighted projection of $\gamma_t$, we then have that $\gamma_0^{\boldsymbol{P}}(\boldsymbol{x}) = \widehat{q}_{\alpha,\boldsymbol{y}}(\boldsymbol{x}, 0)$. Below we proceed to derive the PDE govering the evolution of $\gamma_t^{\boldsymbol{P}}$ based on (B.20). For any test function $\psi : \mathbb{R}^n \to \mathbb{R}$, taking $\varphi(\boldsymbol{x}, \beta) = \beta \psi(\boldsymbol{x}) : \mathbb{R}^n \times \mathbb{R} \to \mathbb{R}$ in the weak form derived in (B.23) and (B.26) yields

$$\begin{aligned}
\left\langle \psi, \frac{\partial}{\partial t} \gamma_t^{\boldsymbol{P}} \right\rangle &= \frac{\mathrm{d}}{\mathrm{d}t} \left\langle \psi, \gamma_t^{\boldsymbol{P}} \right\rangle = \frac{\mathrm{d}}{\mathrm{d}t} \left( \int_{\mathbb{R}^n} \psi(\boldsymbol{x}) \left( \int_{\mathbb{R}} \beta \gamma_t(\boldsymbol{x}, \beta) \mathrm{d}\beta \right) \mathrm{d}\boldsymbol{x} \right) \\
&= \frac{\mathrm{d}}{\mathrm{d}t} \left\langle \varphi, \gamma_t \right\rangle = \left\langle \varphi, \frac{\partial}{\partial t} \gamma_t \right\rangle = \left\langle \varphi, -\nabla_{\boldsymbol{x}} \cdot (\boldsymbol{K}_{\alpha,t} \gamma_t) - \frac{\partial}{\partial \beta} (b_{\alpha,t} \gamma_t) + \frac{1}{2} V(t)^2 \Delta_{\boldsymbol{x}} \gamma_t \right\rangle \\
&= - \left\langle \varphi, \nabla_{\boldsymbol{x}} \cdot (\boldsymbol{K}_{\alpha,t} \gamma_t) \right\rangle - \left\langle \varphi, \frac{\partial}{\partial \beta} (b_{\alpha,t} \gamma_t) \right\rangle + \frac{1}{2} V(t)^2 \left\langle \varphi, \Delta_{\boldsymbol{x}} \gamma_t \right\rangle.
\end{aligned} \tag{B.27}$$

For the first and third terms in the last expression of (B.27), we can further simplify them as follows

$$\begin{aligned}
\left\langle \varphi, \nabla_{\boldsymbol{x}} \cdot (\boldsymbol{K}_{\alpha,t} \gamma_t) \right\rangle &= \int_{\mathbb{R}} \beta \left( \int_{\mathbb{R}^n} \psi(\boldsymbol{x}) \left( \nabla_{\boldsymbol{x}} \cdot (\boldsymbol{K}_{\alpha,t}(\boldsymbol{x}) \gamma_t(\boldsymbol{x}, \beta)) \right) \mathrm{d}\boldsymbol{x} \right) \mathrm{d}\beta \\
&= \int_{\mathbb{R}^n} \psi(\boldsymbol{x}) \left( \nabla_{\boldsymbol{x}} \cdot \left( \boldsymbol{K}_{\alpha,t}(\boldsymbol{x}) \left( \int_{\mathbb{R}} \beta \gamma_t(\boldsymbol{x}, \beta) \mathrm{d}\beta \right) \right) \right) \mathrm{d}\boldsymbol{x} \\
&= \int_{\mathbb{R}^n} \psi(\boldsymbol{x}) \left( \nabla_{\boldsymbol{x}} \cdot (\boldsymbol{K}_{\alpha,t}(\boldsymbol{x}) \gamma_t^{\boldsymbol{P}}(\boldsymbol{x})) \right) \mathrm{d}\boldsymbol{x} \\
&= \left\langle \psi, \nabla_{\boldsymbol{x}} \cdot (\boldsymbol{K}_{\alpha,t} \gamma_t^{\boldsymbol{P}}) \right\rangle
\end{aligned} \tag{B.28}$$

and

$$\begin{aligned}
\left\langle \varphi, \Delta_{\boldsymbol{x}} \gamma_t \right\rangle &= \int_{\mathbb{R}} \beta \left( \int_{\mathbb{R}^n} \psi(\boldsymbol{x}) \left( \Delta_{\boldsymbol{x}} \gamma_t(\boldsymbol{x}, \beta) \right) \mathrm{d}\boldsymbol{x} \right) \mathrm{d}\beta \\
&= \int_{\mathbb{R}^n} \psi(\boldsymbol{x}) \left( \Delta_{\boldsymbol{x}} \left( \int_{\mathbb{R}} \beta \gamma_t(\boldsymbol{x}, \beta) \mathrm{d}\beta \right) \right) \mathrm{d}\boldsymbol{x} \\
&= \int_{\mathbb{R}^n} \psi(\boldsymbol{x}) \left( \Delta_{\boldsymbol{x}} \gamma_t^{\boldsymbol{P}} \right) (\boldsymbol{x}) \mathrm{d}\boldsymbol{x} = \left\langle \psi, \Delta_{\boldsymbol{x}} \gamma_t^{\boldsymbol{P}} \right\rangle,
\end{aligned} \tag{B.29}$$

respectively. Moreover, for the second term in the last expression of (B.27), we may plug in the expression of $b_{\alpha,t}$ and $W_{\alpha,\boldsymbol{y}}(\boldsymbol{x}, t)$ defined in (B.21) and (B.14) above and apply integration by parts to deduce that

$$\begin{aligned}
\left\langle \varphi, \frac{\partial}{\partial \beta} (b_{\alpha,t} \gamma_t) \right\rangle &= - \left\langle \frac{\partial}{\partial \beta} \varphi, b_{\alpha,t} \gamma_t \right\rangle = - \left\langle \psi(\boldsymbol{x}), b_t \gamma_t \right\rangle \\
&= - \int_{\mathbb{R}} \psi(\boldsymbol{x}) \gamma_t(\boldsymbol{x}, \beta) \left( W_{\alpha,\boldsymbol{y}}(\boldsymbol{x}, t) - \int_{\mathbb{R}^n} W_{\alpha,\boldsymbol{y}}(\boldsymbol{x}^*, t) \gamma_t^{\boldsymbol{P}}(\boldsymbol{x}^*) \mathrm{d}\boldsymbol{x}^* \right) \beta \mathrm{d}\boldsymbol{x} \mathrm{d}\beta \\
&= - \int_{\mathbb{R}} \psi(\boldsymbol{x}) \gamma_t^{\boldsymbol{P}}(\boldsymbol{x}) \left( W_{\alpha,\boldsymbol{y}}(\boldsymbol{x}, t) - \int_{\mathbb{R}^n} W_{\alpha,\boldsymbol{y}}(\boldsymbol{x}^*, t) \gamma_t^{\boldsymbol{P}}(\boldsymbol{x}^*) \mathrm{d}\boldsymbol{x}^* \right) \mathrm{d}\boldsymbol{x} \\
&= - \left\langle \psi(\boldsymbol{x}), \gamma_t^{\boldsymbol{P}}(\boldsymbol{x}) \left( W_{\alpha,\boldsymbol{y}}(\boldsymbol{x}, t) - \int_{\mathbb{R}^n} W_{\alpha,\boldsymbol{y}}(\boldsymbol{x}^*, t) \gamma_t^{\boldsymbol{P}}(\boldsymbol{x}^*) \mathrm{d}\boldsymbol{x}^* \right) \right\rangle.
\end{aligned} \tag{B.30}$$

Substituting (B.28), (B.29), and (B.30) into (B.27) then gives us the weak form of the PDE governing the evolution of the projected measure $\gamma_t^{\boldsymbol{P}} = P_\beta \gamma_t$, *i.e.*, $\gamma_t^{\boldsymbol{P}}(\boldsymbol{x}) = P_\beta \gamma_t(\boldsymbol{x})$ satisfies the following PDE

$$\frac{\partial}{\partial t} \gamma_t^{\boldsymbol{P}} = -\nabla_{\boldsymbol{x}} \cdot (\boldsymbol{K}_{\alpha,t} \gamma_t^{\boldsymbol{P}}) + \frac{1}{2} V(t)^2 \Delta_{\boldsymbol{x}} \gamma_t^{\boldsymbol{P}} + \left( W_{\alpha,\boldsymbol{y}}(\boldsymbol{x}, t) - \int_{\mathbb{R}^n} W_{\alpha,\boldsymbol{y}}(\boldsymbol{x}^*, t) \gamma_t^{\boldsymbol{P}}(\boldsymbol{x}^*) \mathrm{d}\boldsymbol{x}^* \right) \gamma_t^{\boldsymbol{P}}, \tag{B.31}$$

with initial condition $\gamma_0^{\boldsymbol{P}}(\boldsymbol{x}) = \widehat{q}_{\alpha,\boldsymbol{y}}(\boldsymbol{x},0)$. This is exactly the PDE in (B.15), which concludes our proof. $\square$

**Remark B.8** (Comparison with Concurrent Work (Skreta et al., 2025))**.** *We note that an alternative approach to derive the dynamics (3.5) for a weighted particle from the PDE (3.4) is to use the Feynman-Kac formula under the formulation of path integrals, as presented in the concurrent work (Skreta et al., 2025, Appendix A). Here we adopt the approach used for proving (Domingo-Enrich et al., 2020, Lemma 1 and 10), which is mainly based on the idea of lifting the projected measure to the joint measure and the weak form of PDE solutions.*

*We adapt the FK Corrector dynamics from (Skreta et al., 2025, Proposition D.5) to provide a direct comparison with our dynamics of a weighted particle (derived from the PDE (3.4) and presented as (3.5)) for the setting of posterior sampling. This is achieved by setting the parameters in their notations as $\beta_t = \alpha_t$, the noise intensity $\sigma_t = V(t)^2$, and the reward function $r = -\mu_{\boldsymbol{y}}$. The resulting drift and reweighting terms for both methods are juxtaposed in Table 3.*

Table 3: Drift and Reweighting Terms of AFDPS and FK Corrector

| Term | AFDPS (Ours) | FK Corrector |
|---|---|---|
| Drift | $-F(t)\boldsymbol{x} + V(t)^2\boldsymbol{\phi}_\theta(\boldsymbol{x},t)$ $-\alpha_t V(t)^2 \nabla_{\boldsymbol{x}}\mu_{\boldsymbol{y}}$ | $-F(t)\boldsymbol{x} + V(t)^2\boldsymbol{\phi}_\theta(\boldsymbol{x},t)$ |
| Reweighting | $\frac{1}{2}V(t)^2\left(\alpha_t^2\|\nabla_{\boldsymbol{x}}\mu_{\boldsymbol{y}}\|_2^2 - \alpha_t\Delta_{\boldsymbol{x}}\mu_{\boldsymbol{y}}\right)$ $+\alpha_t\left(F(t)\boldsymbol{x} - V(t)^2\boldsymbol{\phi}_\theta(\boldsymbol{x},t)\right)^\intercal \nabla_{\boldsymbol{x}}\mu_{\boldsymbol{y}} - \alpha_t'\mu_{\boldsymbol{y}}$ | $-\frac{1}{2}V(t)^2\left(\alpha_t^2\|\nabla_{\boldsymbol{x}}\mu_{\boldsymbol{y}}\|_2^2 - \alpha_t\Delta_{\boldsymbol{x}}\mu_{\boldsymbol{y}}\right)$ $+\alpha_t F(t)\boldsymbol{x}^\intercal\nabla_{\boldsymbol{x}}\mu_{\boldsymbol{y}} - \alpha_t'\mu_{\boldsymbol{y}}$ |

*It is noteworthy that if $V(t) = 0$ (i.e., in the absence of the diffusion-based corrector $\boldsymbol{\phi}_\theta$ and the gradient guidance $\nabla_{\boldsymbol{x}}\mu_{\boldsymbol{y}}$), both AFDPS and the FK Corrector would simplify to the same ODE dynamics, with their drift terms reducing to $-F(t)\boldsymbol{x}$. However, in the more general SDE case where $V(t) \neq 0$, the $-V(t)^2\nabla_{\boldsymbol{x}}\mu_{\boldsymbol{y}}$ term in our AFDPS drift marks a critical difference. Our empirical results, detailed in Section 5, demonstrate that this specific term plays a vital role in effectively guiding the sampler towards regions of high likelihood, thereby enhancing performance.*

*In fact, by using $Q_{\alpha,\boldsymbol{y}}(\boldsymbol{x}) := \breve{p}_t(\boldsymbol{x})e^{-\alpha_t\mu_{\boldsymbol{y}}(\boldsymbol{x})}$ to denote the unnormalized posterior associated with the ground-truth backward SDE (2.4) with $G(t) = V(t)$, we can directly differentiate $Q_{\alpha,\boldsymbol{y}}$ with respect to $\boldsymbol{x}$ to obtain that:*

$$\begin{aligned}
\nabla_{\boldsymbol{x}}Q_{\alpha,\boldsymbol{y}} &= \nabla_{\boldsymbol{x}}\left(\breve{p}_t e^{-\alpha_t\mu_{\boldsymbol{y}}}\right) = \left(\nabla_{\boldsymbol{x}}\breve{p}_t\right)e^{-\alpha_t\mu_{\boldsymbol{y}}} - \alpha_t\breve{p}_t e^{-\alpha_t\mu_{\boldsymbol{y}}}\left(\nabla_{\boldsymbol{x}}\mu_{\boldsymbol{y}}\right) \\
&= \breve{p}_t e^{-\alpha_t\mu_{\boldsymbol{y}}}\left(\nabla_{\boldsymbol{x}}\log\breve{p}_t - \alpha_t\nabla_{\boldsymbol{x}}\mu_{\boldsymbol{y}}\right) = Q_{\alpha,\boldsymbol{y}}\left(\nabla_{\boldsymbol{x}}\log\breve{p}_t - \alpha_t\nabla_{\boldsymbol{x}}\mu_{\boldsymbol{y}}\right)
\end{aligned} \tag{B.32}$$

*Moreover, a derivation similar to the proof of Lemma B.1 yields that the PDE dynamics governing the evolution of $Q_{\boldsymbol{y}}$ is given by*

$$\begin{aligned}
\frac{\partial}{\partial t}Q_{\alpha,\boldsymbol{y}} = &-\nabla_{\boldsymbol{x}}\cdot\left(\left(\boldsymbol{H}(\boldsymbol{x},t) - \alpha_t V(t)^2\nabla_{\boldsymbol{x}}\mu_{\boldsymbol{y}}\right)Q_{\alpha,\boldsymbol{y}}\right) + \frac{1}{2}V(t)^2\Delta_x Q_{\alpha,\boldsymbol{y}} \\
&+ \left(\frac{1}{2}V(t)^2\left(\alpha_t^2\|\nabla_{\boldsymbol{x}}\mu_{\boldsymbol{y}}\|_2^2 - \alpha_t\Delta_{\boldsymbol{x}}\mu_{\boldsymbol{y}}\right) - \alpha_t\boldsymbol{H}(\boldsymbol{x},t)^\intercal\nabla_{\boldsymbol{x}}\mu_{\boldsymbol{y}} - \alpha_t'\mu_{\boldsymbol{y}}\right)Q_{\alpha,\boldsymbol{y}}
\end{aligned} \tag{B.33}$$

*where $\boldsymbol{H}(\boldsymbol{x},t) = -F(t)\boldsymbol{x} + V(t)^2\nabla_{\boldsymbol{x}}\log\breve{p}_t(\boldsymbol{x})$ is essentially obtained by replacing the neural network-based approximation $\boldsymbol{\phi}_\theta(\boldsymbol{x},t)$ in the expression of $\widehat{\boldsymbol{H}}(\boldsymbol{x},t)$ defined above with the true score function $\nabla_{\boldsymbol{x}}\log\breve{p}_t(\boldsymbol{x})$. For any fixed scalar $\eta \in \mathbb{R}$, we may further decompose the term $\nabla_{\boldsymbol{x}}\mu_{\boldsymbol{y}}$ above as the sum of $\eta\nabla_{\boldsymbol{x}}\mu_{\boldsymbol{y}}$ and*

$(1 - \eta)\nabla_{\boldsymbol{x}}\mu_{\boldsymbol{y}}$ *and directly simplify the RHS above as follows:*

$$
\begin{aligned}
\frac{\partial}{\partial t}Q_{\alpha,\boldsymbol{y}} = &-\nabla_{\boldsymbol{x}}\cdot\left(\left(\boldsymbol{H}(\boldsymbol{x},t) - \eta\alpha_t V(t)^2\nabla_{\boldsymbol{x}}\mu_{\boldsymbol{y}}\right)Q_{\alpha,\boldsymbol{y}}\right) + (1-\eta)\alpha_t V(t)^2\nabla_{\boldsymbol{x}}\cdot\left(Q_{\alpha,\boldsymbol{y}}\nabla_{\boldsymbol{x}}\mu_{\boldsymbol{y}}\right) \\
&+\frac{1}{2}V(t)^2\Delta_{\boldsymbol{x}}Q_{\alpha,\boldsymbol{y}} + \left(\frac{1}{2}V(t)^2\left(\alpha_t^2\|\nabla_{\boldsymbol{x}}\mu_{\boldsymbol{y}}\|_2^2 - \alpha_t\Delta_{\boldsymbol{x}}\mu_{\boldsymbol{y}}\right) - \alpha_t\boldsymbol{H}(\boldsymbol{x},t)^\intercal\nabla_{\boldsymbol{x}}\mu_{\boldsymbol{y}} - \alpha_t'\mu_{\boldsymbol{y}}\right)Q_{\alpha,\boldsymbol{y}} \\
= &-\nabla_{\boldsymbol{x}}\cdot\left(\left(\boldsymbol{H}(\boldsymbol{x},t) - \eta\alpha_t V(t)^2\nabla_{\boldsymbol{x}}\mu_{\boldsymbol{y}}\right)Q_{\alpha,\boldsymbol{y}}\right) + (1-\eta)\alpha_t V(t)^2\left(\nabla_{\boldsymbol{x}}\mu_{\boldsymbol{y}}\right)^\intercal\nabla_{\boldsymbol{x}}Q_{\alpha,\boldsymbol{y}} \\
&+(1-\eta)\alpha_t V(t)^2 Q_{\alpha,\boldsymbol{y}}\left(\Delta_{\boldsymbol{x}}\mu_{\boldsymbol{y}}\right) + \frac{1}{2}V(t)^2\Delta_{\boldsymbol{x}}Q_{\alpha,\boldsymbol{y}} \\
&+\left(\frac{1}{2}V(t)^2\left(\alpha_t^2\|\nabla_{\boldsymbol{x}}\mu_{\boldsymbol{y}}\|_2^2 - \alpha_t\Delta_{\boldsymbol{x}}\mu_{\boldsymbol{y}}\right) - \alpha_t\boldsymbol{H}(\boldsymbol{x},t)^\intercal\nabla_{\boldsymbol{x}}\mu_{\boldsymbol{y}} - \alpha_t'\mu_{\boldsymbol{y}}\right)Q_{\alpha,\boldsymbol{y}} \\
= &-\nabla_{\boldsymbol{x}}\cdot\left(\left(\boldsymbol{H}(\boldsymbol{x},t) - \eta\alpha_t - V(t)^2\nabla_{\boldsymbol{x}}\mu_{\boldsymbol{y}}\right)Q_{\alpha,\boldsymbol{y}}\right) - \alpha_t'\mu_{\boldsymbol{y}}Q_{\alpha,\boldsymbol{y}} \\
&+(1-\eta)\alpha_t V(t)^2\left(\nabla_{\boldsymbol{x}}\mu_{\boldsymbol{y}}\right)^\intercal\left(\nabla_{\boldsymbol{x}}\log\breve{p}_t - \alpha_t\nabla_{\boldsymbol{x}}\mu_{\boldsymbol{y}}\right)Q_{\alpha,\boldsymbol{y}} + \frac{1}{2}V(t)^2\Delta_{\boldsymbol{x}}Q_{\alpha,\boldsymbol{y}} \\
&+\left(\frac{1}{2}V(t)^2\alpha_t^2\|\nabla_{\boldsymbol{x}}\mu_{\boldsymbol{y}}\|_2^2 + \left(\frac{1}{2}-\eta\right)\alpha_t V(t)^2\Delta_{\boldsymbol{x}}\mu_{\boldsymbol{y}} - \alpha_t\boldsymbol{H}(\boldsymbol{x},t)^\intercal\nabla_{\boldsymbol{x}}\mu_{\boldsymbol{y}} - \alpha_t'\mu_{\boldsymbol{y}}\right)Q_{\alpha,\boldsymbol{y}} \\
= &-\nabla_{\boldsymbol{x}}\cdot\left(\left(\boldsymbol{H}(\boldsymbol{x},t) - \eta\alpha_t V(t)^2\nabla_{\boldsymbol{x}}\mu_{\boldsymbol{y}}\right)Q_{\alpha,\boldsymbol{y}}\right) + \frac{1}{2}V(t)^2\Delta_{\boldsymbol{x}}Q_{\alpha,\boldsymbol{y}} - \alpha_t'\mu_{\boldsymbol{y}}Q_{\alpha,\boldsymbol{y}} \\
&+\left(\eta - \frac{1}{2}\right)V(t)^2\left(\alpha_t^2\|\nabla_{\boldsymbol{x}}\mu_{\boldsymbol{y}}\|_2^2 - \alpha_t\Delta_{\boldsymbol{x}}\mu_{\boldsymbol{y}}\right)Q_{\alpha,\boldsymbol{y}} \\
&+\alpha_t\left(F(t)\boldsymbol{x} - \eta V(t)^2\nabla_{\boldsymbol{x}}\log\breve{p}_t\left(\boldsymbol{x}\right)\right)^\intercal\left(\nabla_{\boldsymbol{x}}\mu_{\boldsymbol{y}}\right)Q_{\alpha,\boldsymbol{y}}.
\end{aligned}
$$

(B.34)

*where the second last equality above follows from plugging in (B.32).*

*By replacing the true score function $\nabla_{\boldsymbol{x}}\log\breve{p}_t(\boldsymbol{x})$ in the RHS above with the neural network-based estimator $\boldsymbol{\phi}_\theta(\boldsymbol{x},t)$, one then obtains the dynamics that can be used in practice. Specifically, for any fixed $\eta \in \mathbb{R}$, the drift term used in practice is given by*

$$
-F(t)\boldsymbol{x} + V(t)^2\boldsymbol{\phi}_\theta(\boldsymbol{x},t) - \eta\alpha_t V(t)^2\nabla_{\boldsymbol{x}}\mu_{\boldsymbol{y}},
$$

(B.35)

*while the reweighting term used in practice is given by*

$$
\left(\eta - \frac{1}{2}\right)V(t)^2\left(\alpha_t\|\nabla_{\boldsymbol{x}}\mu_{\boldsymbol{y}}\|_2^2 - \alpha_t\Delta_{\boldsymbol{x}}\mu_{\boldsymbol{y}}\right) + \alpha_t\left(F(t)\boldsymbol{x} - \eta V(t)^2\boldsymbol{\phi}_\theta\left(\boldsymbol{x},t\right)\right)^\intercal\nabla_{\boldsymbol{x}}\mu_{\boldsymbol{y}} - \alpha_t'\mu_{\boldsymbol{y}}Q_{\alpha,\boldsymbol{y}}
$$

(B.36)

*By comparing the two terms above with Table 3, we note that $\eta = 0$ yields the FK Corrector dynamics while $\eta = 1$ yields the AFDPS dynamics. Therefore, for more difficult nonlinear inverse problems, we may control the magnitude of the term $V(t)^2\nabla_{\boldsymbol{x}}\mu_{\boldsymbol{y}}$ by tuning the parameter $\eta$ in practice. This also conforms to strategies used in existing practical work on guidance like (Dhariwal & Nichol, 2021; Ho & Salimans, 2022; Bansal et al., 2023; Song et al., 2023c; He et al., 2023; Guo et al., 2024; Ye et al., 2024). Finally, it would be of independent question to mathematically analyze how the discrepancy between the true dynamics (B.34) and the practical dynamics given by (B.35) and (B.36) depends on the parameter $\eta$ in future work.*

## C   Supplementary Proofs and Justifications for Section 4

In this section, we provide detailed proofs and justifications for claims listed in Section 4.

### C.1   Proof of Theorem 4.1

We begin by decomposing the total variation error via triangle's inequality. Specifically, here we slightly abuse the notation by taking $\widetilde{q}_{\alpha,\boldsymbol{y},t}(\cdot) := \widehat{q}_{\alpha,\boldsymbol{y}}(\cdot,t)$ for any $t \in [0,T]$, *i.e.*, $\widetilde{q}_{\alpha,\boldsymbol{y},t}$ satisfies the PDE (3.4):

$$
\begin{aligned}
\frac{\partial}{\partial t}\widetilde{q}_{\alpha,\boldsymbol{y},t} = &-\nabla_{\boldsymbol{x}}\cdot\left(\left(\widehat{\boldsymbol{H}}(\boldsymbol{x},t) - \alpha_t V(t)^2\nabla_{\boldsymbol{x}}\mu_{\boldsymbol{y}}\right)\widetilde{q}_{\alpha,\boldsymbol{y},t}\right) + \frac{1}{2}V(t)^2\Delta_x\widetilde{q}_{\alpha,\boldsymbol{y},t} \\
&+\left(W_{\alpha,\boldsymbol{y}}(\boldsymbol{x},t) - \int_{\mathbb{R}^n}W_{\alpha,\boldsymbol{y}}(\boldsymbol{z},t)\widetilde{q}_{\alpha,\boldsymbol{y},t}(\boldsymbol{z})\mathrm{d}\boldsymbol{z}\right)\widetilde{q}_{\alpha,\boldsymbol{y},t}
\end{aligned}
$$

(C.1)

with initial condition $\widetilde{q}_{\alpha,\boldsymbol{y},0}(\boldsymbol{x}) = \widehat{q}_{\alpha,\boldsymbol{y}}(\boldsymbol{x},0)$. A direct application of triangle's inequality yields

$$\mathrm{TV}\left(\widehat{q}_{\alpha,\boldsymbol{y},T}, q_{\boldsymbol{y},0}\right) \leq \mathrm{TV}\left(\widetilde{q}_{\alpha,\boldsymbol{y},T}, q_{\boldsymbol{y},0}\right) + \mathrm{TV}\left(\widehat{q}_{\alpha,\boldsymbol{y},T}, \widetilde{q}_{\alpha,\boldsymbol{y},T}\right). \tag{C.2}$$

We note that our proof in this section can be divided into two parts, which provide upper bounds on the two terms on the RHS above respectively.

### C.1.1 Bounding the First Term on the RHS of (C.2)

We start off with bounding the first term $\mathrm{TV}\left(\widetilde{q}_{\alpha,\boldsymbol{y},T}, q_{\boldsymbol{y},0}\right)$ on the RHS above, which requires the following two lemmas. Specifically, the first lemma below provides a quantitative bound on the discrepancy between two diffusion processes with different drift functions, while the second lemma describes the convergence of the forward process towards the target distribution when Gaussian noise is added.

**Lemma C.1.** *For any pair of diffusion processes* $(\boldsymbol{x}_t)_{t\in[0,T]}$ *and* $(\widetilde{\boldsymbol{x}}_t)_{t\in[0,T]}$ *on* $\mathbb{R}^n$ *defined as follows*

$$\begin{aligned}
\mathrm{d}\boldsymbol{x}_t &= \boldsymbol{b}(\boldsymbol{x}_t,t)\mathrm{d}t + c(t)\mathrm{d}\boldsymbol{w}_t \\
\text{and} \quad \mathrm{d}\widetilde{\boldsymbol{x}}_t &= \widetilde{\boldsymbol{b}}(\widetilde{\boldsymbol{x}}_t,t)\mathrm{d}t + c(t)\mathrm{d}\boldsymbol{w}_t
\end{aligned} \tag{C.3}$$

*where* $\boldsymbol{b}, \widetilde{\boldsymbol{b}} : \mathbb{R}^n \times [0,T] \to \mathbb{R}^n$ *are the two drift functions,* $c : [0,T] \to \mathbb{R}^+$ *and* $(\boldsymbol{w}_t)_{t\in[0,T]}$ *is a standard Brownian motion. Let* $\rho_t$ *and* $\widetilde{\rho}_t$ *denote the distribution of* $\boldsymbol{x}_t$ *and* $\widetilde{\boldsymbol{x}}_t$ *respectively for any* $t \in [0,T]$, *then we have*

$$D_{\mathrm{KL}}(\rho_T \| \widetilde{\rho}_T) \leq D_{\mathrm{KL}}(\rho_0 \| \widetilde{\rho}_0) + \int_0^T \int_{\mathbb{R}^n} \frac{1}{2c(t)^2} \left\| \boldsymbol{b}(\boldsymbol{x},t) - \widetilde{\boldsymbol{b}}(\boldsymbol{x},t) \right\|_2^2 \rho_t(\boldsymbol{x})\mathrm{d}\boldsymbol{x}\mathrm{d}t. \tag{C.4}$$

*Proof.* We remark that the proof of this lemma is essentially the same as the derivations in many previous works on the theoretical analysis of DMs and variants. Examples include, but are not limited to, (Chen et al., 2023a, Lemma C.1), (Albergo et al., 2023b, Lemma 2.22), and (Wu et al., 2024c, Lemma A.4). For the sake of completeness, we include a detailed derivation here.

The main idea is to use the Fokker-Planck equations associated with the diffusion processes in (C.3) and differentiate the KL divergence between the two evolving densities with respect to time. Specifically, we have that $\rho_t$ and $\widetilde{\rho}_t$ satisfy the following Fokker-Planck equations:

$$\begin{aligned}
\frac{\partial}{\partial t}\rho_t &= -\nabla_{\boldsymbol{x}} \cdot (\boldsymbol{b}(\boldsymbol{x},t)\rho_t) + \frac{1}{2}c(t)^2 \Delta_{\boldsymbol{x}}\rho_t, \\
\text{and} \quad \frac{\partial}{\partial t}\widetilde{\rho}_t &= -\nabla_{\boldsymbol{x}} \cdot \left(\widetilde{\boldsymbol{b}}(\boldsymbol{x},t)\widetilde{\rho}_t\right) + \frac{1}{2}c(t)^2 \Delta_{\boldsymbol{x}}\widetilde{\rho}_t.
\end{aligned} \tag{C.5}$$

From the definition of the KL divergence

$$D_{\mathrm{KL}}(\rho_t \| \widetilde{\rho}_t) = \int_{\mathbb{R}^n} \log \frac{\rho_t(\boldsymbol{x})}{\widetilde{\rho}_t(\boldsymbol{x})} \rho_t(\boldsymbol{x})\mathrm{d}\boldsymbol{x},$$

we can differentiate it with respect to the time variable $t$, which yields

$$\begin{aligned}
\frac{\mathrm{d}}{\mathrm{d}t}D_{\mathrm{KL}}(\rho_t \| \widetilde{\rho}_t) &= \int_{\mathbb{R}^n} \log \frac{\rho_t}{\widetilde{\rho}_t} \frac{\partial \rho_t}{\partial t}\mathrm{d}\boldsymbol{x} + \int_{\mathbb{R}^n} \left( \frac{\partial}{\partial t}\log \rho_t - \frac{\partial}{\partial t}\log \widetilde{\rho}_t \right) \rho_t\mathrm{d}\boldsymbol{x} \\
&= \int_{\mathbb{R}^n} \log \frac{\rho_t}{\widetilde{\rho}_t} \frac{\partial \rho_t}{\partial t}\mathrm{d}\boldsymbol{x} + \int_{\mathbb{R}^n} \left( \frac{1}{\rho_t}\frac{\partial \rho_t}{\partial t} - \frac{1}{\widetilde{\rho}_t}\frac{\partial \widetilde{\rho}_t}{\partial t} \right) \rho_t\mathrm{d}\boldsymbol{x} \\
&= \int_{\mathbb{R}^n} \log \frac{\rho_t}{\widetilde{\rho}_t} \frac{\partial \rho_t}{\partial t}\mathrm{d}\boldsymbol{x} - \int_{\mathbb{R}^n} \frac{\rho_t}{\widetilde{\rho}_t}\frac{\partial \widetilde{\rho}_t}{\partial t}\mathrm{d}\boldsymbol{x}
\end{aligned} \tag{C.6}$$

For the first term in (C.6) above, we plug in (C.5) and use integration by parts, which yields

$$
\begin{aligned}
&\int_{\mathbb{R}^n} \log \frac{\rho_t}{\widetilde{\rho}_t} \frac{\partial \rho_t}{\partial t} \mathrm{d}\boldsymbol{x} \\
&= -\int_{\mathbb{R}^n} (\log \rho_t - \log \widetilde{\rho}_t) \nabla_{\boldsymbol{x}} \cdot \left( \left( \boldsymbol{b} - \frac{c(t)^2}{2} \nabla_{\boldsymbol{x}} \log \rho_t \right) \rho_t \right) \mathrm{d}\boldsymbol{x} \\
&= \int_{\mathbb{R}^n} (\nabla_{\boldsymbol{x}} \log \rho_t - \nabla_{\boldsymbol{x}} \log \widetilde{\rho}_t)^{\mathsf{T}} \left( \boldsymbol{b} - \frac{c(t)^2}{2} \nabla_{\boldsymbol{x}} \log \rho_t \right) \rho_t \mathrm{d}\boldsymbol{x}.
\end{aligned}
\tag{C.7}
$$

To simplify the second term in (C.6), we plug in (C.5) apply integration by parts again to obtain that

$$
\begin{aligned}
-\int_{\mathbb{R}^n} \frac{\rho_t}{\widetilde{\rho}_t} \frac{\partial \widetilde{\rho}_t}{\partial t} \mathrm{d}\boldsymbol{x} &= \int_{\mathbb{R}^n} \frac{\rho_t}{\widetilde{\rho}_t} \nabla_{\boldsymbol{x}} \cdot \left( \left( \widetilde{\boldsymbol{b}} - \frac{c(t)^2}{2} \nabla_{\boldsymbol{x}} \log \widetilde{\rho}_t \right) \widetilde{\rho}_t \right) \mathrm{d}\boldsymbol{x} \\
&= \int_{\mathbb{R}^n} \left[ \nabla_{\boldsymbol{x}} \cdot \left( \widetilde{\boldsymbol{b}} - \frac{c(t)^2}{2} \nabla_{\boldsymbol{x}} \log \widetilde{\rho}_t \right) \rho_t + \left( \widetilde{\boldsymbol{b}} - \frac{c(t)^2}{2} \nabla_{\boldsymbol{x}} \log \widetilde{\rho}_t \right)^{\mathsf{T}} \frac{\nabla_{\boldsymbol{x}} \widetilde{\rho}_t}{\widetilde{\rho}_t} \rho_t \right] \mathrm{d}\boldsymbol{x} \\
&= \int_{\mathbb{R}^n} \left( \widetilde{\boldsymbol{b}} - \frac{c(t)^2}{2} \nabla_{\boldsymbol{x}} \log \widetilde{\rho}_t \right)^{\mathsf{T}} (\nabla_{\boldsymbol{x}} \log \widetilde{\rho}_t - \nabla_{\boldsymbol{x}} \log \rho_t) \rho_t \mathrm{d}\boldsymbol{x}
\end{aligned}
\tag{C.8}
$$

Furthermore, substituting and into then yields

$$
\begin{aligned}
\frac{\mathrm{d}}{\mathrm{d}t} D_{\mathrm{KL}}(\rho_t \| \widetilde{\rho}_t) &= \int_{\mathbb{R}^n} (\nabla_{\boldsymbol{x}} \log \rho_t - \nabla_{\boldsymbol{x}} \log \widetilde{\rho}_t)^{\mathsf{T}} \left( \boldsymbol{b} - \frac{c(t)^2}{2} \nabla_{\boldsymbol{x}} \log \rho_t \right) \rho_t \mathrm{d}\boldsymbol{x} \\
&\quad + \int_{\mathbb{R}^n} (\nabla_{\boldsymbol{x}} \log \widetilde{\rho}_t - \nabla_{\boldsymbol{x}} \log \rho_t)^{\mathsf{T}} \left( \widetilde{\boldsymbol{b}} - \frac{c(t)^2}{2} \nabla_{\boldsymbol{x}} \log \widetilde{\rho}_t \right) \rho_t \mathrm{d}\boldsymbol{x} \\
&= -\frac{c(t)^2}{2} \int_{\mathbb{R}^n} \| \nabla_{\boldsymbol{x}} \log \widetilde{\rho}_t - \nabla_{\boldsymbol{x}} \log \rho_t \|_2^2 \rho_t \mathrm{d}\boldsymbol{x} \\
&\quad + \int_{\mathbb{R}^n} \left( \boldsymbol{b} - \widetilde{\boldsymbol{b}} \right)^{\mathsf{T}} (\nabla_{\boldsymbol{x}} \log \rho_t - \nabla_{\boldsymbol{x}} \log \widetilde{\rho}_t) \rho_t \mathrm{d}\boldsymbol{x} \\
&\leq \frac{1}{2c(t)^2} \int_{\mathbb{R}^n} \left\| \boldsymbol{b} - \widetilde{\boldsymbol{b}} \right\|_2^2 \rho_t \mathrm{d}\boldsymbol{x} = \frac{1}{2c(t)^2} \int_{\mathbb{R}^n} \left\| \boldsymbol{b}(\boldsymbol{x}, t) - \widetilde{\boldsymbol{b}}(\boldsymbol{x}, t) \right\|_2^2 \rho_t(\boldsymbol{x}) \mathrm{d}\boldsymbol{x}
\end{aligned}
\tag{C.9}
$$

where the last inequality follows from the AM-GM inequality, *i.e.* $\boldsymbol{x}^{\mathsf{T}} \boldsymbol{y} \leq \frac{1}{2c(t)^2} \|\boldsymbol{x}\|_2^2 + \frac{c(t)^2}{2} \|\boldsymbol{y}\|_2^2$ for any vectors $\boldsymbol{x}, \boldsymbol{y} \in \mathbb{R}^n$ and $t \in [0, T]$.

Integrating (C.9) from $t = 0$ to $t = T$ then yields (C.4), which concludes our proof. □

**Lemma C.2.** *For any distribution $p$ on $\mathbb{R}^n$ with bounded second moment $m_2^2$, i.e., $\mathbb{E}_{\boldsymbol{x} \sim p}[\|\boldsymbol{x}\|_2^2] \leq m_2^2$, we have* $D_{\mathrm{KL}}\left( p * \mathcal{N}(\boldsymbol{0}, \sigma^2 \boldsymbol{I}_n) \| \mathcal{N}(\boldsymbol{0}, \sigma^2 \boldsymbol{I}_n) \right) \leq \frac{m_2^2}{2\sigma^2}$, *where* $(p * q)(\boldsymbol{x}) := \int_{\mathbb{R}^n} p(\boldsymbol{y}) q(\boldsymbol{x} - \boldsymbol{y}) \mathrm{d}\boldsymbol{y}$ *denotes the convolution of the two probability distributions $p, q$.*

*Proof.* We remark that this is the same as (Wang et al., 2024, Lemma 10), where a complete proof is already provided. □

With Lemma C.1 and Lemma C.2 listed above, we then proceed to bound the term TV $(\widetilde{q}_{\alpha, \boldsymbol{y}, T}, q_{\boldsymbol{y}, 0})$. Consider the backward process associated with the true score function under the EDM framework, which can be formally written as

$$
\mathrm{d}\breve{\boldsymbol{x}}_t = \left[ -\frac{\dot{s}(t)}{s(t)} \breve{\boldsymbol{x}}_t + 2s(t)^2 \dot{\sigma}(t) \sigma(t) \nabla \log \breve{p}_t(\breve{\boldsymbol{x}}_t) \right] \mathrm{d}t + s(t) \sqrt{2\dot{\sigma}(t)\sigma(t)} \mathrm{d}\boldsymbol{w}_t.
\tag{C.10}
$$

with initial condition

$$
\breve{\boldsymbol{x}}_0 \sim \breve{p}_0 = p_T = p_0 * \mathcal{N}(\boldsymbol{0}, T^2 \boldsymbol{I}_n),
$$

where the last identity follows from results derived in Appendix B.1 in the paper (Karras et al., 2022) that proposes the EDM framework as well as our particular choices of the scaling functions $s(t) = 1$ and $\sigma(t) = t$.

Then we consider applying Lemma C.1 to compare the two diffusion processes $(\breve{x}_t)_{t \in [0,T]}$ and $(\breve{\widehat{x}}_t)_{t \in [0,T]}$ defined in (C.10) and (2.5) respectively.

By setting $c(t) = s(t)\sqrt{2\dot{\sigma}(t)\sigma(t)} = \sqrt{2}t$,

$$\boldsymbol{b}(\boldsymbol{x}, t) = -\frac{\dot{s}(t)}{s(t)}\boldsymbol{x} + 2s(t)^2\dot{\sigma}(t)\sigma(t)\nabla \log \breve{p}_t(\boldsymbol{x}) = 2t\nabla \log \breve{p}_t(\boldsymbol{x})$$

and

$$\widetilde{\boldsymbol{b}}(\boldsymbol{x}, t) = -\frac{\dot{s}(t)}{s(t)}\boldsymbol{x} + 2s(t)^2\dot{\sigma}(t)\sigma(t)\boldsymbol{\phi}_\theta(\boldsymbol{x}, t) = 2t\boldsymbol{\phi}_\theta(\boldsymbol{x}, t),$$

we have

$$\begin{aligned}
D_{\mathrm{KL}}(p_0\|\breve{\widehat{p}}_T) &= D_{\mathrm{KL}}(\breve{p}_T\|\breve{\widehat{p}}_T) \\
&\leq D_{\mathrm{KL}}(\breve{p}_0\|\breve{\widehat{p}}_0) + \int_0^T \int_{\mathbb{R}^n} \frac{1}{4t} \|2t(\boldsymbol{\phi}_\theta(\boldsymbol{x}, t) - \nabla_{\boldsymbol{x}} \log \breve{p}_t(\boldsymbol{x}))\|_2^2 \, \breve{p}_t(\boldsymbol{x})\mathrm{d}\boldsymbol{x}\mathrm{d}t \\
&= D_{\mathrm{KL}}\left(p_0 * \mathcal{N}(\boldsymbol{0}, T^2\boldsymbol{I}_n)\|\mathcal{N}(\boldsymbol{0}, T^2\boldsymbol{I}_n)\right) \\
&\quad + \int_0^T \int_{\mathbb{R}^n} t \|\boldsymbol{\phi}_\theta(\boldsymbol{x}, t) - \nabla_{\boldsymbol{x}} \log \breve{p}_t(\boldsymbol{x})\|_2^2 \, \breve{p}_t(\boldsymbol{x})\mathrm{d}\boldsymbol{x}\mathrm{d}t \leq \frac{m_2^2}{2T^2} + \frac{1}{2}T^2\epsilon_{\boldsymbol{s}}^2,
\end{aligned} \tag{C.11}$$

where the second lest inequality above follows from Lemma C.1 and the last inequality follows from Assumption 4.2, Assumption 4.3 and Lemma C.2.

Applying Pinsker's inequality helps us further bound the TV divergence between $p_0$ and $\breve{\widehat{p}}_T$ as follows

$$\mathrm{TV}\left(\breve{\widehat{p}}_T, p_0\right) = \mathrm{TV}\left(p_0, \breve{\widehat{p}}_T\right) \leq \sqrt{\frac{1}{2}D_{\mathrm{KL}}(p_0\|\breve{\widehat{p}}_T)} \leq \frac{1}{2}\sqrt{\frac{m_2^2}{T^2} + T^2\epsilon_{\boldsymbol{s}}^2}. \tag{C.12}$$

Based on the bounds on the distance between the two prior distributions above, we proceed to bound the distance between the two associated posterior distributions. From our definition of $\widetilde{q}_{\alpha,\boldsymbol{y},t}$ in equation (C.1) above, we have that

$$\widetilde{q}_{\alpha,\boldsymbol{y},T}(\boldsymbol{x}) = \widehat{q}_{\alpha,\boldsymbol{y}}(\boldsymbol{x}, T) \propto \widehat{\breve{p}}_T(\boldsymbol{x})e^{-\alpha_t\mu_{\boldsymbol{y}}(\boldsymbol{x})} = \widehat{\breve{p}}_T(\boldsymbol{x})e^{-\mu_{\boldsymbol{y}}(\boldsymbol{x})} \quad \text{and} \quad q_{\boldsymbol{y},0}(\boldsymbol{x}) \propto p_0(x)e^{-\mu_{\boldsymbol{y}}(\boldsymbol{x})},$$

By using

$$\widehat{Z}(\boldsymbol{y}) := \int_{\mathbb{R}^n} \widehat{\breve{p}}_T(\boldsymbol{x})e^{-\mu_{\boldsymbol{y}}(\boldsymbol{x})}\mathrm{d}\boldsymbol{x} \quad \text{and} \quad Z(\boldsymbol{y}) := \int_{\mathbb{R}^n} p_0(x)e^{-\mu_{\boldsymbol{y}}(\boldsymbol{x})}\mathrm{d}\boldsymbol{x}$$

to denote the two corresponding normalizing constants, we can further deduce that

$$\left|\widehat{Z}(\boldsymbol{y}) - Z(\boldsymbol{y})\right| = \left|\int_{\mathbb{R}^n} e^{-\mu_{\boldsymbol{y}}(\boldsymbol{x})}\left(\widehat{\breve{p}}_T(\boldsymbol{x}) - p_0(\boldsymbol{x})\right)\mathrm{d}\boldsymbol{x}\right| \leq 2e^{-C_{\boldsymbol{y}}^{(1)}}\mathrm{TV}\left(\widehat{\breve{p}}_T, p_0\right) \tag{C.13}$$

where the inequality above follows from Assumption 4.1. Then we can use the bound on the difference between the normalizing constants above to further obtain that

$$
\begin{aligned}
\mathrm{TV}(\widetilde{q}_{\alpha,\boldsymbol{y},T}, q_{\boldsymbol{y},0}) &= \frac{1}{2}\int_{\mathbb{R}^n}\left|\frac{1}{\widehat{Z}(\boldsymbol{y})}\widehat{\widetilde{p}}_T(\boldsymbol{x})e^{-\mu_{\boldsymbol{y}}(\boldsymbol{x})} - \frac{1}{Z(\boldsymbol{y})}p_0(\boldsymbol{x})e^{-\mu_{\boldsymbol{y}}(\boldsymbol{x})}\right|\mathrm{d}\boldsymbol{x} \\
&\leq \frac{1}{2}\int_{\mathbb{R}^n}\left|\frac{1}{\widehat{Z}(\boldsymbol{y})}\widehat{\widetilde{p}}_T(\boldsymbol{x})e^{-\mu_{\boldsymbol{y}}(\boldsymbol{x})} - \frac{1}{Z(\boldsymbol{y})}\widehat{\widetilde{p}}_T(\boldsymbol{x})e^{-\mu_{\boldsymbol{y}}(\boldsymbol{x})}\right|\mathrm{d}\boldsymbol{x} \\
&\quad + \frac{1}{2}\int_{\mathbb{R}^n}\left|\frac{1}{Z(\boldsymbol{y})}\widehat{\widetilde{p}}_T(\boldsymbol{x})e^{-\mu_{\boldsymbol{y}}(\boldsymbol{x})} - \frac{1}{Z(\boldsymbol{y})}p_0(\boldsymbol{x})e^{-\mu_{\boldsymbol{y}}(\boldsymbol{x})}\right|\mathrm{d}\boldsymbol{x} \\
&= \frac{|Z(\boldsymbol{y}) - \widehat{Z}(\boldsymbol{y})|}{2Z(\boldsymbol{y})\widehat{Z}(\boldsymbol{y})}\left(\int_{\mathbb{R}^n}\widehat{\widetilde{p}}_T(\boldsymbol{x})e^{-\mu_{\boldsymbol{y}}(\boldsymbol{x})}\mathrm{d}\boldsymbol{x}\right) \\
&\quad + \frac{1}{2Z(\boldsymbol{y})}\left|\int_{\mathbb{R}^n}e^{-\mu_{\boldsymbol{y}}(\boldsymbol{x})}\left(\widehat{\widetilde{p}}_T(\boldsymbol{x}) - p_0(\boldsymbol{x})\right)\mathrm{d}\boldsymbol{x}\right| \\
&\leq \frac{1}{2Z(\boldsymbol{y})}\left(\left|\widehat{Z}(\boldsymbol{y}) - Z(\boldsymbol{y})\right| + 2e^{-C_{\boldsymbol{y}}^{(1)}}\mathrm{TV}\left(\widehat{\widetilde{p}}_T, p_0\right)\right) \\
&\leq \frac{2e^{-C_{\boldsymbol{y}}^{(1)}}}{Z(\boldsymbol{y})}\mathrm{TV}\left(\widehat{\widetilde{p}}_T, p_0\right) \leq \frac{e^{-C_{\boldsymbol{y}}^{(1)}}}{Z(\boldsymbol{y})}\sqrt{\frac{m_2^2}{T^2} + T^2\epsilon_{\boldsymbol{s}}^2},
\end{aligned}
$$

where the first inequality above follows from triangle inequality, the second inequality above follows from Assumption 4.1, the third inequality above follows from (C.13) and the last inequality above follows from (C.12).

By setting

$$
C_{\boldsymbol{y}}^{(2)} := \frac{e^{-C_{\boldsymbol{y}}^{(1)}}}{Z(\boldsymbol{y})}
$$

in the last expression above, which is some constant that only depends on $\boldsymbol{y}$, we finally obtain the following upper bound for the first term on the RHS of (C.2):

$$
\mathrm{TV}\left(\widetilde{q}_{\alpha,\boldsymbol{y},T}, q_{\boldsymbol{y},0}\right) \leq C_{\boldsymbol{y}}^{(2)}\sqrt{\frac{m_2^2}{T^2} + T^2\epsilon_{\boldsymbol{s}}^2}. \tag{C.14}
$$

### C.1.2 Bounding the Second Term on the RHS of (C.2)

Then we proceed to upper bound the second term $\mathrm{TV}\left(\widehat{q}_{\alpha,\boldsymbol{y},T}, \widetilde{q}_{\alpha,\boldsymbol{y},T}\right)$. Following the same set of notations used in Appendix B above, we define $\boldsymbol{K}(\boldsymbol{x},t) := \widehat{\boldsymbol{H}}(\boldsymbol{x},t) - \alpha_t V(t)^2\nabla_{\boldsymbol{x}}\mu_{\boldsymbol{y}}(\boldsymbol{x})$ and $\zeta(t) := \frac{1}{2}V(t)^2$. Moreover, we use $\mathcal{P}_t$ to denote the following time-dependent operator for any test function $\phi : \mathbb{R}^n \to \mathbb{R}$,

$$
(\mathcal{P}_t\phi)(\boldsymbol{x}) := \boldsymbol{K}(\boldsymbol{x},t)^{\mathsf{T}}\nabla_{\boldsymbol{x}}\phi(\boldsymbol{x}) + \zeta(t)\Delta_{\boldsymbol{x}}\phi(\boldsymbol{x}) \tag{C.15}
$$

A direct computation yields that the associated adjoint operator $\mathcal{P}_t^*$ is exactly the time-dependent infinitesimal generator given by the sum of the drift and diffusion term on the RHS of (C.1), *i.e.*,

$$
\left(\mathcal{P}_t^*\phi\right)(\boldsymbol{x}) := -\nabla_{\boldsymbol{x}}\cdot\left(\boldsymbol{K}(\boldsymbol{x},t)\phi(\boldsymbol{x})\right) + \zeta(t)\Delta_{\boldsymbol{x}}\phi(\boldsymbol{x}), \tag{C.16}
$$

for any test function $\phi : \mathbb{R}^n \to \mathbb{R}$. Furthermore, we use $\lambda := \lambda_t(\phi)$ to denote the functional formed by the linear term on the RHS of (C.2), *i.e.*,

$$
\lambda_t(\phi) := \left(W_{\alpha,\boldsymbol{y}}(\boldsymbol{x},t) - \int_{\mathbb{R}^n}W_{\alpha,\boldsymbol{y}}(\boldsymbol{z},t)\phi(\boldsymbol{z})\mathrm{d}\boldsymbol{z}\right)\phi(\boldsymbol{x}) = \left(tI(\boldsymbol{x},t) - t\int_{\mathbb{R}^n}I(\boldsymbol{z},t)\phi(\boldsymbol{z})\mathrm{d}\boldsymbol{z}\right)\phi(\boldsymbol{x}) \tag{C.17}
$$

for any test function $\phi : \mathbb{R}^n \to \mathbb{R}$, where the last identity above follows from our definition of $I(\boldsymbol{x},t)$ given in Assumption 4.4 and the special choice that $\sigma(t) = t$. Then we may use the notations introduced above to rewrite the PDE (C.1) as follows:

$$
\frac{\partial}{\partial t}q(\boldsymbol{x},t) = (\mathcal{P}_t^*q)(\boldsymbol{x}) + \lambda_t(q) \tag{C.18}
$$

Based on the time-dependent infinitesimal generator (C.16) and the PDE (C.18) above, we may further define the two-parameter semigroup operator $\mathcal{U}_{s,t}$ by

$$(\mathcal{U}_{s,t}\phi)(\boldsymbol{x}) = \mathbb{E}\left[\phi(X_t)|X_s = \boldsymbol{x}\right] \tag{C.19}$$

for any time interval $[s,t]$ and test function $\phi : \mathbb{R}^n \to \mathbb{R}$, where $(X_\tau)_{\tau \geq s}$ is driven by the following time-inhomogeneous SDE.

$$\mathrm{d}X_\tau = -\boldsymbol{K}(X_\tau, \tau)\mathrm{d}\tau + \sqrt{2\zeta(t)}\mathrm{d}W_\tau, X_s = \boldsymbol{x}. \tag{C.20}$$

Moreover, based on the theory of semigroups and PDEs, we have an alternative interpretation of $\mathcal{U}_{s,t}$. Specifically, for any test function $\phi : \mathbb{R}^n \to \mathbb{R}$, we have that $(\mathcal{U}_{s,t}\phi)(\boldsymbol{x}) = u(\boldsymbol{x}, t)$ is exactly the solution of the following forward PDE:

$$\frac{\partial}{\partial \tau}u(\boldsymbol{x}, \tau) = -\nabla_{\boldsymbol{x}} \cdot (\boldsymbol{K}(\boldsymbol{x}, \tau)u(\boldsymbol{x}, \tau)) + \zeta(\tau)\Delta_{\boldsymbol{x}}u(\boldsymbol{x}, \tau) = (\mathcal{P}_\tau^* u)(\boldsymbol{x}) \text{ for } \tau \in (s, t], \ u(\boldsymbol{x}, s) = \phi(\boldsymbol{x}). \tag{C.21}$$

From the PDE-based interpretation above, we also have that the adjoint two-parameter semigroup operator $\mathcal{U}_{s,t}^*$ of $\mathcal{U}_{s,t}$ can be defined via the backward PDE associated with $\mathcal{P}_t$. Specifically, for any time interval $[s,t]$ and test function $\phi : \mathbb{R}^n \to \mathbb{R}$, we have that $(\mathcal{U}_{s,t}^*\phi)(\boldsymbol{x}) = v(\boldsymbol{x}, s)$ is exactly the solution of the following backward PDE:

$$\frac{\partial}{\partial \tau}v(\boldsymbol{x}, \tau) = \boldsymbol{K}(\boldsymbol{x}, \tau)^{\mathsf{T}}\nabla_{\boldsymbol{x}}v(\boldsymbol{x}, \tau) + \zeta(\tau)\Delta_{\boldsymbol{x}}v(\boldsymbol{x}, \tau) = (\mathcal{P}_\tau v)(\boldsymbol{x}) \text{ for } \tau \in [s, t), \ v(\boldsymbol{x}, t) = \phi(\boldsymbol{x}). \tag{C.22}$$

Furthermore, we note that the two target distributions $\widehat{q}_{\alpha,\boldsymbol{y},T}$ and $\widetilde{q}_{\alpha,\boldsymbol{y},T}$ are essentially solutions to PDEs of the same form (C.18) but with two different initial conditions $\widehat{q}_{\alpha,\boldsymbol{y},0}(\cdot)$ and $\widetilde{q}_{\alpha,\boldsymbol{y},T}(\cdot) = \widehat{q}_{\alpha,\boldsymbol{y}}(\cdot, 0)$. Therefore, in order to upper bound the term, here we only need to prove the stability of PDE (C.18) above with respect to the initial conditions. Before proving the generic stability argument, we need to prove the following two lemmas beforehand. Specifically, the first lemma characterizes solution to the PDE (C.18) based on the generator $\mathcal{P}_t^*$ and initial condition, while the second lemma shows the contractiveness of the two-parameter semigroup $\mathcal{U}_{s,t}$ with respect to the $L^1$ norm.

**Lemma C.3** (Duhamel's Principle/Variation of Parameters Formula). *A generic formula of the solution $q(t, \boldsymbol{x}) = q_t(\boldsymbol{x})$ to the PDE (C.18) with initial condition $q(0, \boldsymbol{x}) = q_0(\boldsymbol{x})$ can be written as follows:*

$$q(\boldsymbol{x}, t) = \mathcal{U}_{0,t}q_0 + \int_0^t (\mathcal{U}_{s,t}\lambda_s(q_s))\mathrm{d}s \tag{C.23}$$

*Proof.* We note that the generic formula above directly follows from Duhamel's Principle/Variation of parameters formula. For the sake of completeness, we provide a derivation below. In fact, using the fact that $\mathcal{U}_{s,t}\phi$ satisfies the PDE (C.21) with initial condition $u(s, \boldsymbol{x}) = \phi(\boldsymbol{x})$ for any $\phi$, we can differentiate with respect to $t$ on both sides of (C.23) to deduce that

$$\frac{\partial}{\partial t}q(\boldsymbol{x}, t) = \frac{\partial}{\partial t}\left(\mathcal{U}_{0,t}q_0\right) + \frac{\mathrm{d}}{\mathrm{d}t}\left(\int_0^t (\mathcal{U}_{s,t}\lambda_s(q_s))\mathrm{d}s\right) = \frac{\partial}{\partial t}\left(\mathcal{U}_{0,t}q_0\right) + \mathcal{U}_{t,t}\lambda_t(q_t) + \int_0^t \frac{\partial}{\partial t}(\mathcal{U}_{s,t}\lambda_s(q_s))\mathrm{d}s$$

$$= \left(\mathcal{P}_t^*\mathcal{U}_{0,t}q_0\right)(\boldsymbol{x}) + \lambda_t(q(\boldsymbol{x}, t)) + \int_0^t \left(\mathcal{P}_t^*\mathcal{U}_{s,t}\lambda_s(q_s)\right)\mathrm{d}s \tag{C.24}$$

$$= \mathcal{P}_t^*\left(\mathcal{U}_{0,t}q_0(\boldsymbol{x}) + \int_0^t \mathcal{U}_{s,t}\lambda_s(q_s)\mathrm{d}s\right) + \lambda_t(q(\boldsymbol{x}, t)) = (\mathcal{P}_t^*q)(\boldsymbol{x}) + \lambda_t(q)$$

which is exactly the PDE (C.18), as desired. □

**Lemma C.4** (Contractiveness of the two-parameter semigroup operator). *For any time interval $[s,t] \subset [0,T]$ and test function $\phi : \mathbb{R}^n \to \mathbb{R}$, we have $\|\mathcal{U}_{s,t}\phi\|_{L^1} \leq \|\phi\|_{L^1}$.*

*Proof.* We note that it suffices to show that the operator norm $\|\mathcal{U}_{s,t}\|_{L^1 \to L^1}$ is bounded by 1. Given that the dual of the $L^1$ norm is exactly the $L^\infty$ norm, we have that

$$
\begin{aligned}
\|\mathcal{U}_{s,t}\|_{L^1 \to L^1} &= \sup_{\|\phi\|_{L^1} \leq 1} \|\mathcal{U}_{s,t}\phi\|_{L^1} = \sup_{\|\phi\|_{L^1} \leq 1} \sup_{\|\psi\|_{L^\infty} \leq 1} \langle \psi, \mathcal{U}_{s,t}\phi \rangle_{L^2} \\
&= \sup_{\|\psi\|_{L^\infty} \leq 1} \sup_{\|\phi\|_{L^1} \leq 1} \langle \phi, \mathcal{U}_{s,t}^*\psi \rangle_{L^2} = \sup_{\|\psi\|_{L^\infty} \leq 1} \left\|\mathcal{U}_{s,t}^*\psi\right\|_{L^\infty} = \left\|\mathcal{U}_{s,t}^*\right\|_{L^\infty \to L^\infty}
\end{aligned}
\tag{C.25}
$$

where $\langle \cdot, \cdot \rangle_{L^2}$ above denotes the inner product associated with the $L^2$ norm, *i.e.*, $\langle \psi, \phi \rangle_{L^2} = \int_{\mathbb{R}^n} \psi(\boldsymbol{x})\phi(\boldsymbol{x})\mathrm{d}\boldsymbol{x}$ for any test functions $\psi, \phi : \mathbb{R}^n \to \mathbb{R}$. Therefore, here we only need to show that

$$
\left\|\mathcal{U}_{s,t}^*\right\|_{L^\infty \to L^\infty} \leq 1,
\tag{C.26}
$$

*i.e.*, $\left\|\mathcal{U}_{s,t}^*\phi\right\|_{L^\infty} \leq \|\phi\|_{L^\infty}$ for any bounded test function $\phi : \mathbb{R}^n \to \mathbb{R}$. We note that this essentially follows from applying maximum principle to the backward PDE (C.22). For a detailed discussion on the maximum principle, the readers may refer to (Evans, 2022). Specifically, it suffices to show that the solution $v(\boldsymbol{x}, \tau)$ to the PDE (C.22) satisfies

$$
\begin{aligned}
\sup_{\tau \in [s,t], \ \boldsymbol{x} \in \mathbb{R}^n} v(\boldsymbol{x}, \tau) &\leq \sup_{\boldsymbol{x} \in \mathbb{R}^n} v(\boldsymbol{x}, t), \\
\inf_{\tau \in [s,t], \ \boldsymbol{x} \in \mathbb{R}^n} v(\boldsymbol{x}, \tau) &\geq \inf_{\boldsymbol{x} \in \mathbb{R}^n} v(\boldsymbol{x}, t),
\end{aligned}
\tag{C.27}
$$

when the initial function $v(\boldsymbol{x}, t)$ is bounded with respect to the $L^\infty$ norm. We begin by proving the first inequality regarding the supremum. For any fixed $\omega > 0$ that can be arbitrarily small, we consider the perturbed solution $v^\omega(\boldsymbol{x}, \tau) := v(\boldsymbol{x}, \tau) - \omega\tau$. We will first show that the supremum of $v^\omega$ must be attained at the boundary when $\tau = t$, *i.e.*,

$$
\sup_{\tau \in [s,t], \ \boldsymbol{x} \in \mathbb{R}^n} v^\omega(\boldsymbol{x}, \tau) \leq \sup_{\boldsymbol{x} \in \mathbb{R}^n} v^\omega(\boldsymbol{x}, t).
\tag{C.28}
$$

A direct computation yields that $v^\omega$ satisfies the following modified backward PDE:

$$
\begin{aligned}
\frac{\partial}{\partial \tau} v^\omega(\boldsymbol{x}, \tau) &= \frac{\partial}{\partial \tau} v(\boldsymbol{x}, \tau) - \omega = (\mathcal{P}_\tau v)(\boldsymbol{x}) - \omega = \boldsymbol{K}(\boldsymbol{x}, \tau)^\mathsf{T} \nabla_{\boldsymbol{x}} v(\boldsymbol{x}, \tau) + \zeta(\tau)\Delta_{\boldsymbol{x}} v(\boldsymbol{x}, \tau) - \omega \\
&= \boldsymbol{K}(\boldsymbol{x}, \tau)^\mathsf{T} \nabla_{\boldsymbol{x}} v^\omega(\boldsymbol{x}, \tau) + \zeta(\tau)\Delta_{\boldsymbol{x}} v^\omega(\boldsymbol{x}, \tau) - \omega = (\mathcal{P}_\tau v^\omega)(\boldsymbol{x}) - \omega.
\end{aligned}
\tag{C.29}
$$

For the sake of contradiction, assume that $\sup_{\tau \in [s,t], \ \boldsymbol{x} \in \mathbb{R}^n} v^\omega(\boldsymbol{x}, \tau) > \sup_{\boldsymbol{x} \in \mathbb{R}^n} v^\omega(\boldsymbol{x}, t)$ *i.e.*, the supremum of $v^\omega$ over $(\boldsymbol{x}, \tau) \in \mathbb{R}^n \times [s, t]$ is attained at some point $(x^*, \tau^*) \in \mathbb{R}^n \times [s, t)$. Since $(\boldsymbol{x}^*, \tau^*)$ is also a local maximum, we must have

$$
\frac{\partial}{\partial \tau} v^\omega(\boldsymbol{x}^*, \tau^*) \geq 0, \ \nabla_{\boldsymbol{x}} v^\omega(\boldsymbol{x}^*, \tau^*) = \boldsymbol{0}, \ \Delta_{\boldsymbol{x}} v^\omega(\boldsymbol{x}^*, \tau^*) \leq 0.
\tag{C.30}
$$

Substituting the inequalities and equality above into the modified PDE (C.29) then yields

$$
0 \leq \frac{\partial}{\partial \tau} v^\omega(\boldsymbol{x}^*, \tau^*) = \zeta(\tau)\Delta_{\boldsymbol{x}} v^\omega(\boldsymbol{x}^*, \tau^*) - \omega \leq -\omega < 0,
\tag{C.31}
$$

which leads to a contradiction. Therefore, the assumption is wrong and (C.28) is proved. Based on the definition of $v$ and $v^\omega$, we have the following inequality

$$
v^\omega(\boldsymbol{x}, \tau) \leq v(\boldsymbol{x}, \tau) = v^\omega(\boldsymbol{x}, \tau) + \omega\tau \leq v^\omega(\boldsymbol{x}, \tau) + \omega T,
\tag{C.32}
$$

for any $(\boldsymbol{x}, \tau) \in \mathbb{R}^n \times [s, t]$. Taking supremum with respect to $(\boldsymbol{x}, \tau)$ in the inequality above and plugging in (C.28) then imply

$$
\sup_{\tau \in [s,t], \ \boldsymbol{x} \in \mathbb{R}^n} v(\boldsymbol{x}, \tau) \leq \sup_{\tau \in [s,t], \ \boldsymbol{x} \in \mathbb{R}^n} v^\omega(\boldsymbol{x}, \tau) + \omega T \leq \sup_{\boldsymbol{x} \in \mathbb{R}^n} v^\omega(\boldsymbol{x}, t) + \omega T \leq \sup_{\boldsymbol{x} \in \mathbb{R}^n} v(\boldsymbol{x}, t) + \omega T.
$$

Taking the limit $\omega \to 0^+$ in the inequality above then implies $\sup_{\tau \in [s,t], \ \boldsymbol{x} \in \mathbb{R}^n} v(\boldsymbol{x}, \tau) \leq \sup_{\boldsymbol{x} \in \mathbb{R}^n} v(\boldsymbol{x}, t)$, which completes our proof of the first inequality in (C.27) above. For the second inequality regarding the infimum, we note that $\sup(-v) = -\inf v$. Hence, the second inequality can be proved in the same way by considering $-v$ and the corresponding perturbation $(-v)^\omega := -v + \omega\tau$. This concludes our proof of (C.26) and Lemma C.4. $\qquad\square$

With the two lemmas above, we may now prove stability of the PDE (C.21). In general, consider any two solutions $q^{(i)}(\boldsymbol{x}, t)$ $(i = 1, 2)$ to (C.21) with two different initial conditions $q^{(1)}(\boldsymbol{x}, 0) = q_0^{(1)}(\boldsymbol{x})$ and $q^{(2)}(\boldsymbol{x}, 0) = q_0^{(2)}(\boldsymbol{x})$, i.e.,

$$\frac{\partial}{\partial t} q^{(i)}(\boldsymbol{x}, t) = \left(\mathcal{P}_t^* q^{(i)}\right)(\boldsymbol{x}) + \lambda_t\left(q^{(i)}\right), (i = 1, 2) \tag{C.33}$$

Applying the formula (C.23) derived in Lemma above, we also have that the two solutions satisfy

$$q^{(i)}(\boldsymbol{x}, t) = \mathcal{U}_{0,t} q_0^{(i)} + \int_0^t \mathcal{U}_{s,t} \lambda_s\left(q_s^{(i)}\right) \mathrm{d}s \ (i = 1, 2). \tag{C.34}$$

We further use $r_t(\boldsymbol{x}) := q^{(1)}(\boldsymbol{x}, t) - q^{(2)}(\boldsymbol{x}, t)$ to denote the difference between the two solutions. Then it suffices to show that the total variation

$$\mathrm{TV}\left(q^{(1)}(\cdot, t), q^{(2)}(\cdot, t)\right) = \frac{1}{2}\left\|q^{(1)}(\cdot, t) - q^{(2)}(\cdot, t)\right\|_{L^1} = \frac{1}{2}\|r_t\|_{L^1}, \tag{C.35}$$

can be upper bounded by the initial error $\mathrm{TV}\left(q^{(1)}(\cdot, 0), q^{(2)}(\cdot, 0)\right) = \frac{1}{2}\|r_0\|_{L^1}$ up to some constant depending on $\boldsymbol{y}$ and $t$. In fact, taking the difference of the two equations in (C.34) and applying the linearity of the two-parameter semigroup operator $\mathcal{U}_{s,t}$, we have that

$$
\begin{aligned}
\|r_t\|_{L^1} &= \left\|\mathcal{U}_{0,t}\left(q_0^{(1)} - q_0^{(2)}\right) + \int_0^t \mathcal{U}_{s,t}\left(\lambda_s\left(q_s^{(1)}\right) - \lambda_s\left(q_s^{(2)}\right)\right) \mathrm{d}s\right\|_{L^1} \\
&\le \left\|\mathcal{U}_{0,t}\left(q_0^{(1)} - q_0^{(2)}\right)\right\|_{L^1} + \int_0^t \left\|\mathcal{U}_{s,t}\left(\lambda_s\left(q_s^{(1)}\right) - \lambda_s\left(q_s^{(2)}\right)\right)\right\|_{L^1} \mathrm{d}s \\
&\le \left\|q_0^{(1)} - q_0^{(2)}\right\|_{L^1} + \int_0^t \left\|\lambda_s\left(q_s^{(1)}\right) - \lambda_s\left(q_s^{(2)}\right)\right\|_{L^1} \mathrm{d}s,
\end{aligned}
\tag{C.36}
$$

where the first and second inequality above follows from triangle inequality and contractiveness of the semigroup operator proved in Lemma C.4 above, respectively. Moreover, plugging in the expression of $\lambda$ defined in (C.17) above yields

$$
\begin{aligned}
\left\|\lambda_s\left(q_s^{(1)}\right) - \lambda_s\left(q_s^{(2)}\right)\right\|_{L^1} &= s \left\|I(\boldsymbol{x}, s)\left(q_s^{(1)} - q_s^{(2)}\right) - \langle I(\cdot, s), q_s^{(1)}\rangle_{L^2} q_s^{(1)} + \langle I(\cdot, s), q_s^{(2)}\rangle_{L^2} q_s^{(2)}\right\|_{L^1} \\
&\le s \left\|I(\boldsymbol{x}, s)\left(q_s^{(1)} - q_s^{(2)}\right)\right\|_{L^1} + s \left\|\left(\langle I(\cdot, s), q_s^{(1)} - q_s^{(2)}\rangle_{L^2}\right) q_s^{(1)}\right\|_{L^1} \\
&\quad + s \left\|\left(\langle I(\cdot, s), q_s^{(2)}\rangle_{L^2}\right)\left(q_s^{(1)} - q_s^{(2)}\right)\right\|_{L^1}.
\end{aligned}
\tag{C.37}
$$

Furthermore, applying the upper bound on $I(\boldsymbol{x}, s)$ in Assumption 4.4 then yields that for any $\boldsymbol{x}$ and $s \in [0, t]$,

$$
\begin{aligned}
\left\|I(\boldsymbol{x}, s)\left(q_s^{(1)} - q_s^{(2)}\right)\right\|_{L^1} &\le B_{\boldsymbol{y}} \left\|q_s^{(1)} - q_s^{(2)}\right\|_{L^1}, \\
\left\|\left(\langle I(\cdot, s), q_s^{(1)} - q_s^{(2)}\rangle_{L^2}\right) q_s^{(1)}\right\|_{L^1} &\le \left(\int_{\mathbb{R}^n} |I(\boldsymbol{x}, s)| \left|q_s^{(1)}(\boldsymbol{x}) - q_s^{(2)}(\boldsymbol{x})\right| \mathrm{d}\boldsymbol{x}\right)\left(\int_{\mathbb{R}^n} q_s^{(1)}(\boldsymbol{x}) \mathrm{d}\boldsymbol{x}\right) \\
&\le B_{\boldsymbol{y}} \int_{\mathbb{R}^n} \left|q_s^{(1)}(\boldsymbol{x}) - q_s^{(2)}(\boldsymbol{x})\right| \mathrm{d}\boldsymbol{x} = B_{\boldsymbol{y}} \left\|q_s^{(1)} - q_s^{(2)}\right\|_{L^1}, \\
\left\|\left(\langle I(\cdot, s), q_s^{(2)}\rangle_{L^2}\right)\left(q_s^{(1)} - q_s^{(2)}\right)\right\|_{L^1} &\le \left|\langle I(\cdot, s), q_s^{(2)}\rangle_{L^2}\right| \left\|q_s^{(1)} - q_s^{(2)}\right\|_{L^1} \\
&= \left(\int_{\mathbb{R}^n} |I(\boldsymbol{x}, s)| q_s^{(2)}(\boldsymbol{x}) \mathrm{d}\boldsymbol{x}\right) | \left\|q_s^{(1)} - q_s^{(2)}\right\|_{L^1} \\
&\le B_{\boldsymbol{y}}\left(\int_{\mathbb{R}^n} q_s^{(2)}(\boldsymbol{x}) \mathrm{d}\boldsymbol{x}\right) \left\|q_s^{(1)} - q_s^{(2)}\right\|_{L^1} = B_{\boldsymbol{y}} \left\|q_s^{(1)} - q_s^{(2)}\right\|_{L^1}.
\end{aligned}
\tag{C.38}
$$

Substituting the three bounds derived above into (C.37) and (C.36) then gives us that

$$\|r_t\|_{L^1} \le \|r_0\|_{L^1} + \int_0^t 3 B_{\boldsymbol{y}} s \left\|q_s^{(1)} - q_s^{(2)}\right\|_{L^1} \mathrm{d}s \le \int_0^t 3 B_{\boldsymbol{y}} s \|r_s\| \mathrm{d}s + \|r_0\|_{L^1}. \tag{C.39}$$

Applying Grönwall's inequality then implies that $\|r_t\|_{L^1} \le \|r_0\|_{L^1} \, e^{\int_0^t 3B_{\boldsymbol{y}} s \mathrm{d}s} = \|r_0\|_{L^1} \, e^{\frac{3}{2}B_{\boldsymbol{y}}t^2}$, *i.e.*,

$$\mathrm{TV}\left(q^{(1)}(\cdot,t), q^{(2)}(\cdot,t)\right) = \frac{1}{2}\|r_t\|_{L^1} \le \frac{1}{2}\|r_0\|_{L^1} \, e^{\frac{3}{2}B_{\boldsymbol{y}}t^2} = e^{\frac{3}{2}B_{\boldsymbol{y}}t^2}\mathrm{TV}\left(q^{(1)}(\cdot,0), q^{(2)}(\cdot,0)\right). \tag{C.40}$$

By taking $q^{(1)}(\cdot,t) := \widehat{q}_{\alpha,\boldsymbol{y},t}$ and $q^{(2)}(\cdot,t) := \widetilde{q}_{\alpha,\boldsymbol{y},t}$ in the stability inequality above, we obtain that

$$\mathrm{TV}\left(\widehat{q}_{\alpha,\boldsymbol{y},T}, \widetilde{q}_{\alpha,\boldsymbol{y},T}\right) \le e^{\frac{3}{2}B_{\boldsymbol{y}}T^2}\mathrm{TV}\left(\widehat{q}_{\alpha,\boldsymbol{y},0}, \widetilde{q}_{\alpha,\boldsymbol{y},0}\right) =: C_{\boldsymbol{y},T}^{(3)}\mathrm{TV}\left(\widehat{q}_{\alpha,\boldsymbol{y},0}, \widehat{q}_{\alpha,\boldsymbol{y}}(\cdot,0)\right) = C_{\boldsymbol{y},T}^{(3)}\epsilon_I, \tag{C.41}$$

where $\epsilon_I$ is the initial error introduced before Theorem 4.1 above and $C_{\boldsymbol{y},T}^{(3)} := e^{\frac{3}{2}B_{\boldsymbol{y}}T^2}$ is some constant depending on $\boldsymbol{y}, T$ only. This concludes our proof of the upper bound for the second term on the RHS of (C.2).

Finally, substituting the two upper bounds proved in Appendix C.1.1 and C.1.2 above into (C.2) yields

$$\mathrm{TV}\left(\widehat{q}_{\alpha,\boldsymbol{y},T}, q_{\boldsymbol{y},0}\right) \le C_{\boldsymbol{y}}^{(2)}\sqrt{\frac{m_2^2}{T^2} + T^2\epsilon_{\boldsymbol{s}}^2} + C_{\boldsymbol{y},T}^{(3)}\epsilon_I,$$

which concludes our proof of Theorem 4.1. In particular, for the case when the initial sampling error $\epsilon_I = 0$, balancing the two terms in the last expression above also yields $\frac{m_2^2}{T^2} = T^2\epsilon_{\boldsymbol{s}}^2$, *i.e.*, $T = \sqrt{\frac{m_2}{\epsilon_s}}$ gives us the optimal upper bound

$$\mathrm{TV}(\widehat{q}_{\alpha,\boldsymbol{y},T}, q_{\boldsymbol{y},0}) \le C_{\boldsymbol{y}}^{(2)}\sqrt{m_2\epsilon_{\boldsymbol{s}}},$$

which is proportional to the square root of the score matching error defined in Assumption 4.3.

## C.2 Proof of Theorem 4.2

Our proof of Theorem 4.2 is mainly based on arguments from propagation of chaos (Sznitman, 1991; Lacker, 2018). Recall that

$$\gamma_\tau^N(\boldsymbol{x}, \beta) = \frac{1}{N}\sum_{i=1}^N \delta_{(\boldsymbol{x}_\tau^{(i)}, \beta_\tau^{(i)})}$$

denotes the joint measured formed by the $N$ weighted particles $\left\{\left(\boldsymbol{x}_\tau^{(i)}, \beta_\tau^{(i)}\right)\right\}_{i=1}^N$ given by (3.6) and $\gamma_\tau$ is the joint probability distribution of the single weighted particle $(\boldsymbol{x}_\tau, \beta_\tau)$ satisfying (3.5).

Now we consider an auxiliary system of $N$ weighted particles $\{(\widetilde{\boldsymbol{x}}_t^{(i)}, \widetilde{\beta}_t^{(i)})\}_{i=1}^N$ sampled identically and independently from the single particle dynamics (3.5), *i.e.*,

$$\begin{cases} \mathrm{d}\widetilde{\boldsymbol{x}}_t^{(i)} &= \left(\widehat{\boldsymbol{H}}(\widetilde{\boldsymbol{x}}_t^{(i)}, t) - \alpha_t V(t)^2 \nabla_{\boldsymbol{x}}\mu_{\boldsymbol{y}}(\widetilde{\boldsymbol{x}}_t^{(i)})\right)\mathrm{d}t + V(t)\mathrm{d}\boldsymbol{w}_t^{(i)}, \\ \mathrm{d}\widetilde{\beta}_t^{(i)} &= \left(U_{\alpha,\boldsymbol{y}}(\widetilde{\boldsymbol{x}}_t^{(i)}, t) - \alpha_t\widehat{\boldsymbol{H}}(\widetilde{\boldsymbol{x}}_t^{(i)}, t)^\intercal\nabla_{\boldsymbol{x}}\mu_{\boldsymbol{y}}(\widetilde{\boldsymbol{x}}_t^{(i)}) - \alpha_t'\mu_{\boldsymbol{y}}(\boldsymbol{x}_t^{(i)})\right)\widetilde{\beta}_t^{(i)}\mathrm{d}t \\ &\quad - \left(\int_{\mathbb{R}^n}\left(U_{\alpha,\boldsymbol{y}}(\boldsymbol{x}, t) - \alpha_t\widehat{\boldsymbol{H}}(\boldsymbol{x}, t)^\intercal\nabla_{\boldsymbol{x}}\mu_{\boldsymbol{y}}(\boldsymbol{x}) - \alpha_t'\mu_{\boldsymbol{y}}(\boldsymbol{x})\right)(P_\beta\gamma_t)(\boldsymbol{x})\mathrm{d}\boldsymbol{x}\right)\widetilde{\beta}_t^{(i)}\mathrm{d}t. \end{cases} \tag{C.42}$$

We note that the initial conditions of (C.42) are given by $\widetilde{\boldsymbol{x}}_0^{(i)} = \boldsymbol{x}_0^{(i)} \sim \widehat{q}_{\alpha,\boldsymbol{y}}(\cdot,0)$ and $\widetilde{\beta}_0^{(i)} = 1$ for $i \in [N]$. Moreover, we have that the $\left(\boldsymbol{w}_t^{(i)}\right)_{t \in [0,T]}$ is the same standard Brownian motion used in (3.6) for any $i \in [N]$, which implies $\boldsymbol{x}_t^{(i)} \equiv \widetilde{\boldsymbol{x}}_t^{(i)}$ for any $i \in [N]$ and $t \in [0,T]$. Then we consider the joint empirical measure

$$\widetilde{\gamma}_t^N(\boldsymbol{x}, \beta) = \frac{1}{N}\sum_{i=1}^N \delta_{(\widetilde{\boldsymbol{x}}_t^{(i)}, \widetilde{\beta}_t^{(i)})} \tag{C.43}$$

formed by the $N$ weighted particles $\left\{\left(\widetilde{\boldsymbol{x}}_t^{(i)}, \widetilde{\beta}_t^{(i)}\right)\right\}_{i=1}^N$ given by (C.42).

Before we proceed, we establish the following upper bound:

**Lemma C.5.** *The following upper bound on the absolute values of the weights holds for any $t \in [0, T]$*

$$\max_{i \in [N]} \left\{ |\beta_t|, \left| \beta_t^{(i)} \right|, \left| \widetilde{\beta}_t^{(i)} \right| \right\} \leq \exp \left( B_{\boldsymbol{y}} t^2 \right). \tag{C.44}$$

*Proof.* Below we will only prove the upper bound in (C.44) above for the weight $\widetilde{\beta}_t^{(i)}$ governed by (C.42), as the same upper bound for $\beta_t^{(i)}$ governed by (3.6) and $\beta_t$ governed by (3.5) can be proved via the same procedure. By integrating from $0$ to $t$ on both sides of (C.42) and applying the bound on $I$ provided in the statement of Theorem 4.2, we have that

$$
\begin{aligned}
\left| \widetilde{\beta}_t^{(i)} \right| &\leq \left| \int_0^t \left( U_{\alpha, \boldsymbol{y}} \left( \widetilde{\boldsymbol{x}}_\tau^{(i)}, \tau \right) - \alpha_\tau \widehat{\boldsymbol{H}} \left( \widetilde{\boldsymbol{x}}_\tau^{(i)}, \tau \right)^\mathsf{T} \nabla_{\boldsymbol{x}} \mu_{\boldsymbol{y}} \left( \widetilde{\boldsymbol{x}}_\tau^{(i)} \right) - \alpha_\tau' \mu_{\boldsymbol{y}} \left( \widetilde{\boldsymbol{x}}_\tau^{(i)} \right) \right) \widetilde{\beta}_\tau^{(i)} \mathrm{d}\tau \right| + \widetilde{\beta}_0^{(i)} \\
&\quad + \left| \int_0^t \left( \int_{\mathbb{R}^n} \left( U_{\alpha, \boldsymbol{y}}(\boldsymbol{x}, \tau) - \alpha_\tau \widehat{\boldsymbol{H}}(\boldsymbol{x}, \tau)^\mathsf{T} \nabla_{\boldsymbol{x}} \mu_{\boldsymbol{y}}(\boldsymbol{x}) - \alpha_\tau' \mu_{\boldsymbol{y}}(\boldsymbol{x}) \right) (P_\beta \gamma_\tau)(\boldsymbol{x}) \mathrm{d}\boldsymbol{x} \right) \widetilde{\beta}_\tau^{(i)} \mathrm{d}\tau \right| \\
&\leq \int_0^t \left| \tau I \left( \widetilde{\boldsymbol{x}}_\tau^{(i)}, \tau \right) \widetilde{\beta}_\tau^{(i)} \right| \mathrm{d}\tau + \int_0^t \left| \tau \left( \int_{\mathbb{R}^n} I(\boldsymbol{x}, \tau) (P_\beta \gamma_\tau)(\boldsymbol{x}) \mathrm{d}\boldsymbol{x} \right) \widetilde{\beta}_\tau^{(i)} \right| \mathrm{d}\tau + 1 \\
&\leq 2 \int_0^t B_{\boldsymbol{y}} \tau \left| \widetilde{\beta}_\tau^{(i)} \right| \mathrm{d}\tau + 1,
\end{aligned}
\tag{C.45}
$$

where the last inequality above follows from the assumed upper bound on $I$ and Lemma B.4, which shows that the weighted projection $P_\beta \gamma_t$ is a probability measure on $\mathbb{R}^n$ for any $t \in [0, T]$. Applying Gronwall's inequality to (C.45) then yields the upper bound in (C.44), as desired. $\square$

*Proof of Theorem 4.2.* By recalling the definition of the Wasserstein-2 distance as follows

$$\mathcal{W}_2^2(\mu, \nu) := \inf_{\Gamma \in \Pi(\mu, \nu)} \left( \int_{\mathbb{R}^d \times \mathbb{R}^d} \|\boldsymbol{x} - \boldsymbol{y}\|_2^2 \Gamma(\boldsymbol{x}, \boldsymbol{y}) \mathrm{d}\boldsymbol{x} \mathrm{d}\boldsymbol{y} \right), \tag{C.46}$$

where $\Pi(\mu, \nu)$ denotes the set of couplings between any two distributions $\mu, \nu$ on $\mathbb{R}^d$ for fixed $d$, we can apply triangle inequality

$$\|a + b\|_2^2 \leq 2 \left( \|a\|_2^2 + \|b\|_2^2 \right),$$

and take expectation on both sides to deduce that for any fixed $t \in [0, T]$ and $\tau \in [0, t]$, the following inequality

$$\mathbb{E} \left[ \mathcal{W}_2^2(\gamma_\tau^N, \gamma_\tau) \right] \leq 2 \mathbb{E} \left[ \mathcal{W}_2^2(\gamma_\tau^N, \widetilde{\gamma}_\tau^N) \right] + 2 \mathbb{E} \left[ \mathcal{W}_2^2(\widetilde{\gamma}_\tau^N, \gamma_\tau) \right] \tag{C.47}$$

holds for any $N$.

Taking supremum with respect to $\tau \in [0, t]$ on both sides above then yields

$$\sup_{\tau \in [0, t]} \mathbb{E} \left[ \mathcal{W}_2^2(\gamma_\tau^N, \gamma_\tau) \right] \leq 2 \sup_{\tau \in [0, t]} \mathbb{E} \left[ \mathcal{W}_2^2(\gamma_\tau^N, \widetilde{\gamma}_\tau^N) \right] + 2 \sup_{\tau \in [0, t]} \mathbb{E} \left[ \mathcal{W}_2^2(\widetilde{\gamma}_\tau^N, \gamma_\tau) \right]. \tag{C.48}$$

We then need to bound the two terms on the RHS of (C.47).

For the first term in (C.47), we note that the empirical measure

$$\frac{1}{N} \sum_{i=1}^N \delta_{(\boldsymbol{x}_\tau^{(i)}, \beta_\tau^{(i)}), (\widetilde{\boldsymbol{x}}_\tau^{(i)}, \widetilde{\beta}_\tau^{(i)})}$$

defined on $\mathbb{R}^{n+1} \times \mathbb{R}^{n+1}$ is a coupling between $\gamma_\tau^N$ and $\widetilde{\gamma}_\tau^N$ for any time $\tau \in [0, t]$. Setting $\Gamma$ in (C.43) to be such a coupling then gives us the following upper bound on the expected Wasserstein-2 distance:

$$
\begin{aligned}
\sup_{\tau \in [0, t]} \mathbb{E} \left[ \mathcal{W}_2^2(\gamma_\tau^N, \widetilde{\gamma}_\tau^N) \right] &\leq \sup_{\tau \in [0, t]} \left( \frac{1}{N} \sum_{i=1}^N \mathbb{E} \left[ \left\| \boldsymbol{x}_t^{(i)} - \widetilde{\boldsymbol{x}}_t^{(i)} \right\|_2^2 + \left| \beta_t^{(i)} - \widetilde{\beta}_t^{(i)} \right|^2 \right] \right) \\
&= \frac{1}{N} \sum_{i=1}^N \sup_{\tau \in [0, t]} \mathbb{E} \left[ \left| \beta_t^{(i)} - \widetilde{\beta}_t^{(i)} \right|^2 \right].
\end{aligned}
\tag{C.49}
$$

where the equality above follows from the observation $\boldsymbol{x}_t^{(i)} \equiv \widetilde{\boldsymbol{x}}_t^{(i)}$ for any $i \in [N]$ and $t \in [0, T]$.

Below, we use

$$L(\boldsymbol{x}, t) := U_{\alpha, \boldsymbol{y}}(\boldsymbol{x}, t) - \alpha_t \widehat{\boldsymbol{H}}(\boldsymbol{x}, t)^\mathsf{T} \nabla_{\boldsymbol{x}} \mu_{\boldsymbol{y}}(\boldsymbol{x}) - \alpha_t' \mu_{\boldsymbol{y}}(\boldsymbol{x})$$

to denote the drift function appearing in the dynamics (3.6) and (C.42). By plugging in the choices $s(t) = 1, \sigma(t) = t$ stated in Theorem 4.2, we then have

$$L = U_{\alpha, \boldsymbol{y}} - \alpha_t \widehat{\boldsymbol{H}}^\mathsf{T} \nabla_{\boldsymbol{x}} \mu_{\boldsymbol{y}} - \alpha_t' \mu_{\boldsymbol{y}} = t \left( \alpha_t^2 \|\nabla_{\boldsymbol{x}} \mu_{\boldsymbol{y}}\|_2^2 - \alpha_t \Delta_{\boldsymbol{x}} \mu_{\boldsymbol{y}} \right) - 2t\alpha_t \boldsymbol{\phi}_\theta^\mathsf{T} \nabla_{\boldsymbol{x}} \mu_{\boldsymbol{y}} - \alpha_t' \mu_{\boldsymbol{y}} = tI, \qquad \text{(C.50)}$$

where $I = I(\boldsymbol{x}, t)$ is defined in the statement of Theorem 4.2.

Now we return to bound the RHS of (C.49). By taking the difference between the two dynamics (3.6) and (C.42) and applying triangle inequality, we then plug in $\widetilde{\boldsymbol{x}}_t^{(i)} \equiv \boldsymbol{x}_t^{(i)}$ to obtain the following decomposed upper bound for any $\tau' \in [0, \tau]$ with fixed $\tau \in [0, t]$ and $i \in [N]$:

$$\begin{aligned}
\left| \frac{\mathrm{d}}{\mathrm{d}\tau'} \beta_{\tau'}^{(i)} - \frac{\mathrm{d}}{\mathrm{d}\tau'} \widetilde{\beta}_{\tau'}^{(i)} \right| &\leq \left| L\left( \boldsymbol{x}_{\tau'}^{(i)}, \tau' \right) \left( \beta_{\tau'}^{(i)} - \widetilde{\beta}_{\tau'}^{(i)} \right) \right| \\
&\quad + \left| \left( \int_{\mathbb{R}^n} L\left( \boldsymbol{x}, \tau' \right) \left( P_\beta \gamma_{\tau'} \right)(\boldsymbol{x}) \mathrm{d}\boldsymbol{x} \right) \left( \beta_{\tau'}^{(i)} - \widetilde{\beta}_{\tau'}^{(i)} \right) \right| \\
&\quad + \left| \int_{\mathbb{R}^{n+1}} \beta L\left( \boldsymbol{x}, \tau' \right) \left( \gamma_{\tau'}^N(\boldsymbol{x}, \beta) - \gamma_{\tau'}(\boldsymbol{x}, \beta) \right) \mathrm{d}\boldsymbol{x}\mathrm{d}\beta \right| \left| \beta_{\tau'}^{(i)} \right| \\
&\leq 2B_{\boldsymbol{y}} \tau' \left| \beta_{\tau'}^{(i)} - \widetilde{\beta}_{\tau'}^{(i)} \right| \\
&\quad + \left| \int_{\mathbb{R}^{n+1}} \beta I\left( \boldsymbol{x}, \tau' \right) \left( \gamma_{\tau'}^N(\boldsymbol{x}, \beta) - \gamma_{\tau'}(\boldsymbol{x}, \beta) \right) \mathrm{d}\boldsymbol{x}\mathrm{d}\beta \right| \tau' \exp\left( B_{\boldsymbol{y}} \tau'^2 \right),
\end{aligned} \qquad \text{(C.51)}$$

where the last inequality above follows from (C.44) and assumed upper bound on the function $I = \frac{1}{t} L$.

Furthermore, we recall the following property of the Wasserstein distances $\mathcal{W}_1$ and $\mathcal{W}_2$:

$$\mathcal{W}_1(\mu, \nu) := \sup_{g:\mathbb{R}^n \to \mathbb{R}, \ \mathrm{Lip}(g) \leq 1} \int_{\mathbb{R}^n} g(\boldsymbol{x}) \left( \mu(\boldsymbol{x}) - \nu(\boldsymbol{x}) \right) \mathrm{d}\boldsymbol{x} \leq \mathcal{W}_2(\mu, \nu). \qquad \text{(C.52)}$$

From the assumed upper bound on $\mathrm{Lip}(I)$ given in Theorem 4.2, we have $\mathrm{Lip}\left( \frac{1}{B_{\boldsymbol{y}}} I \right) \leq 1$. Setting

$$g(\boldsymbol{x}, \beta) := \frac{\beta I(\boldsymbol{x}, \tau')}{e^{B_{\boldsymbol{y}} \tau'^2} B_{\boldsymbol{y}}},$$

$\mu := \gamma_{\tau'}^N$, and $\nu := \gamma_{\tau'}$ in (C.52) above for any $\tau' \in [0, \tau]$ then implies

$$\begin{aligned}
\left| \int_{\mathbb{R}^n} \beta I\left( \boldsymbol{x}, \tau' \right) \left( \gamma_{\tau'}^N(\boldsymbol{x}, \beta) - \gamma_{\tau'}(\boldsymbol{x}, \beta) \right) \mathrm{d}\boldsymbol{x}\mathrm{d}\beta \right| &\leq B_{\boldsymbol{y}} e^{B_{\boldsymbol{y}} \tau'^2} \mathcal{W}_1\left( \gamma_{\tau'}^N, \gamma_{\tau'} \right) \\
&\leq B_{\boldsymbol{y}} e^{B_{\boldsymbol{y}} \tau'^2} \mathcal{W}_2\left( \gamma_{\tau'}^N, \gamma_{\tau'} \right).
\end{aligned} \qquad \text{(C.53)}$$

Substituting (C.53) into (C.51), squaring on both sides and applying AM-GM inequality indicate that for any $\tau' \in [0, \tau]$ and $i \in [N]$:

$$\begin{aligned}
\left| \frac{\mathrm{d}}{\mathrm{d}\tau'} \beta_{\tau'}^{(i)} - \frac{\mathrm{d}}{\mathrm{d}\tau'} \widetilde{\beta}_{\tau'}^{(i)} \right|^2 &\leq \left( 2B_{\boldsymbol{y}} \tau' \left| \beta_{\tau'}^{(i)} - \widetilde{\beta}_{\tau'}^{(i)} \right| + B_{\boldsymbol{y}} \tau' \exp\left( 2B_{\boldsymbol{y}} \tau'^2 \right) \mathcal{W}_2\left( \gamma_{\tau'}^N, \gamma_{\tau'} \right) \right)^2 \\
&\leq 8B_{\boldsymbol{y}}^2 \tau'^2 \left| \beta_{\tau'}^{(i)} - \widetilde{\beta}_{\tau'}^{(i)} \right|^2 + 2B_{\boldsymbol{y}}^2 \tau'^2 \exp\left( 4B_{\boldsymbol{y}} \tau'^2 \right) \mathcal{W}_2^2\left( \gamma_{\tau'}^N, \gamma_{\tau'} \right).
\end{aligned} \qquad \text{(C.54)}$$

Integrating from $\tau' = 0$ to $\tau' = \tau$ on both sides above and applying Cauchy-Schwarz inequality imply that for any $\tau \in [0, t]$ and $i \in [N]$:

$$
\begin{aligned}
\left| \beta_\tau^{(i)} - \widetilde{\beta}_\tau^{(i)} \right|^2 &= \left| \int_0^\tau \left( \frac{\mathrm{d}}{\mathrm{d}\tau'} \beta_{\tau'}^{(i)} - \frac{\mathrm{d}}{\mathrm{d}\tau'} \widetilde{\beta}_{\tau'}^{(i)} \right) \mathrm{d}\tau' \right|^2 \leq \tau \left( \int_0^\tau \left| \frac{\mathrm{d}}{\mathrm{d}\tau'} \beta_{\tau'}^{(i)} - \frac{\mathrm{d}}{\mathrm{d}\tau'} \widetilde{\beta}_{\tau'}^{(i)} \right|^2 \mathrm{d}\tau' \right) \\
&\leq 8 B_{\boldsymbol{y}}^2 \tau \int_0^\tau \tau'^2 \left| \beta_{\tau'}^{(i)} - \widetilde{\beta}_{\tau'}^{(i)} \right|^2 \mathrm{d}\tau' \\
&\quad + 2\tau \int_0^\tau B_{\boldsymbol{y}}^2 \tau'^2 \exp\left( 4 B_{\boldsymbol{y}} \tau'^2 \right) \mathcal{W}_2^2 \left( \gamma_{\tau'}^N, \gamma_{\tau'} \right) \mathrm{d}\tau'.
\end{aligned}
\tag{C.55}
$$

Applying Gronwall's inequality to the function $\frac{1}{\tau'} \left| \beta_{\tau'}^{(i)} - \widetilde{\beta}_{\tau'}^{(i)} \right|^2$ in (C.55) above then yields

$$
\frac{1}{\tau} \left| \beta_\tau^{(i)} - \widetilde{\beta}_\tau^{(i)} \right|^2 \leq 2 \left( \int_0^\tau B_{\boldsymbol{y}}^2 \tau'^2 \exp\left( 4 B_{\boldsymbol{y}} \tau'^2 \right) \mathcal{W}_2^2 \left( \gamma_{\tau'}^N, \gamma_{\tau'} \right) \mathrm{d}\tau' \right) e^{\int_0^\tau 8 B_{\boldsymbol{y}}^2 \tau'^3 \mathrm{d}\tau'}.
\tag{C.56}
$$

Then we multiply $\tau$ and take the expectation on both sides of (C.56). A direct application of Fubini's Theorem then indicates that for any $i \in [N]$ and $\tau \in [0, t]$:

$$
\begin{aligned}
\mathbb{E}\left[ \left| \beta_\tau^{(i)} - \widetilde{\beta}_\tau^{(i)} \right|^2 \right] &\leq 2\tau e^{2 B_{\boldsymbol{y}}^2 \tau^4} \int_0^\tau B_{\boldsymbol{y}}^2 \tau'^2 \exp\left( 4 B_{\boldsymbol{y}} \tau'^2 \right) \mathbb{E}\left[ \mathcal{W}_2^2 \left( \gamma_{\tau'}^N, \gamma_{\tau'} \right) \right] \mathrm{d}\tau' \\
&\leq 2 B_{\boldsymbol{y}}^2 \tau^3 e^{2 B_{\boldsymbol{y}}^2 \tau^4 + 4 B_{\boldsymbol{y}} \tau^2} \int_0^\tau \sup_{\tau'' \in [0, \tau']} \mathbb{E}\left[ \mathcal{W}_2^2 \left( \gamma_{\tau''}^N, \gamma_{\tau''} \right) \right] \mathrm{d}\tau'.
\end{aligned}
\tag{C.57}
$$

Taking supremum with respect to $\tau \in [0, t]$ on both sides of (C.57) further implies

$$
\sup_{\tau \in [0, t]} \mathbb{E}\left[ \left| \beta_\tau^{(i)} - \widetilde{\beta}_\tau^{(i)} \right|^2 \right] \leq 2 B_{\boldsymbol{y}}^2 t^3 e^{2 B_{\boldsymbol{y}}^2 t^4 + 4 B_{\boldsymbol{y}} t^2} \int_0^t \sup_{\tau' \in [0, \tau]} \mathbb{E}\left[ \mathcal{W}_2^2 \left( \gamma_{\tau'}^N, \gamma_{\tau'} \right) \right] \mathrm{d}\tau,
\tag{C.58}
$$

for any $i \in [N]$ and $t \in [0, T]$.

Substituting (C.58) above into (C.49) and then (C.48) indicates

$$
\begin{aligned}
\sup_{\tau \in [0, t]} \mathbb{E}\left[ \mathcal{W}_2^2(\gamma_\tau^N, \gamma_\tau) \right] &\leq 4 B_{\boldsymbol{y}}^2 t^3 e^{2 B_{\boldsymbol{y}}^2 t^4 + 4 B_{\boldsymbol{y}} t^2} \int_0^t \sup_{\tau' \in [0, \tau]} \mathbb{E}\left[ \mathcal{W}_2^2 \left( \gamma_{\tau'}^N, \gamma_{\tau'} \right) \right] \mathrm{d}\tau \\
&\quad + 2 \sup_{\tau \in [0, t]} \mathbb{E}\left[ \mathcal{W}_2^2(\widetilde{\gamma}_\tau^N, \gamma_\tau) \right] \\
&\leq \int_0^t 4 B_{\boldsymbol{y}}^2 T^3 e^{2 B_{\boldsymbol{y}}^2 T^4 + 4 B_{\boldsymbol{y}} T^2} \sup_{\tau' \in [0, \tau]} \mathbb{E}\left[ \mathcal{W}_2^2 \left( \gamma_{\tau'}^N, \gamma_{\tau'} \right) \right] \mathrm{d}\tau \\
&\quad + 2 \sup_{\tau \in [0, t]} \mathbb{E}\left[ \mathcal{W}_2^2(\widetilde{\gamma}_\tau^N, \gamma_\tau) \right],
\end{aligned}
\tag{C.59}
$$

for any $t \in [0, T]$.

Applying Gronwall's inequality again to the function $\sup_{\tau \in [0, t]} \mathbb{E}\left[ \mathcal{W}_2^2(\gamma_\tau^N, \gamma_\tau) \right]$ further implies that

$$
\sup_{\tau \in [0, t]} \mathbb{E}\left[ \mathcal{W}_2^2(\gamma_\tau^N, \gamma_\tau) \right] \leq 2 \exp\left( 4 B_{\boldsymbol{y}}^2 T^4 e^{2 B_{\boldsymbol{y}}^2 T^4 + 4 B_{\boldsymbol{y}} T^2} \right) \sup_{\tau \in [0, t]} \mathbb{E}\left[ \mathcal{W}_2^2(\widetilde{\gamma}_\tau^N, \gamma_\tau) \right]
\tag{C.60}
$$

for any $t \in [0, T]$.

By setting $t = T$ in (C.60) above and taking the limit $N \to \infty$, we then have

$$
\lim_{N \to \infty} \mathbb{E}\left[ \mathcal{W}_2^2(\gamma_\tau^N, \gamma_\tau) \right] = \lim_{N \to \infty} \mathbb{E}\left[ \mathcal{W}_2^2(\widetilde{\gamma}_\tau^N, \gamma_\tau) \right] = 0,
$$

for any $\tau \in [0, T]$ with $T$ fixed, where the last equality above follows from Lemma B.4 and the law of large numbers (See, for instance, (Lacker, 2018, Corollary 2.14)). This concludes our proof. $\qquad\square$

**Remark C.6.** *We note that one may also adopt similar arguments used in (Domingo-Enrich et al., 2020) to prove existence and uniqueness of solutions to the SDE systems (3.6) and (3.5). In fact, such type of mean field analysis based on arguments from propogation of chaos have been widely adopted for studying different types of PDEs arising from subfields of not only physical sciences but also data sciences, such as fluid dynamics (Goodman et al., 1990), kinetic theory (Carrillo & Vaes, 2021; Borghi & Pareschi, 2025), theory of two layer neural networks (Mei et al., 2019; Hu et al., 2021), ensemble-based sampling and variational inference (Lu et al., 2019a; Kelly et al., 2014; Schillings & Stuart, 2017; 2018; Ding & Li, 2021a;b). For some good reference on related mathematical models, one may refer to (Muntean et al., 2016). Therefore, it would be of independent interest to investigate whether we can develop more refined mathematical theory for the two sampling algorithms proposed in this paper by combining perspectives from gradient flows or numerical analysis. Moreover, it would also be interesting to investigate how existing mathematical theory (Eberle & Marinelli, 2006; Schweizer, 2012; Eberle & Marinelli, 2013; Beskos et al., 2014b;a; 2016; Giraud & Del Moral, 2017) developed for SMC can be applied to analyze Algorithm 2 and Algorithm 4 that we proposed here.*

# D  Additional Implementation Details for Linear Inverse Problems

## D.1  Datasets, model checkpoints and inverse problem setups

**Data usage**   We mainly test our methods and the baseline methods on the FFHQ-256 (Karras et al., 2019) dataset and the ImageNet-256 (Deng et al., 2009) dataset. All images used for the tests in this paper are in RGB. For FFHQ-256, the 100 testing images were selected to be the first 100 images in the dataset, whoses indexes range from 00000 to 00099. For ImageNet-256, the 100 testing images were selected to be the first 100 images in the ImageNet-1k validation set.

**Model checkpoints**   The two pretrained score functions for the FFHQ-256 and the ImageNet-256 datasets used in this paper were directly taken from the ones used in (Chung et al., 2022), which are available in the following Google Drive [1]. However, since these checkpoints were all trained based on the DDPM formulation (Ho et al., 2020), we adpoted the same transformation used in (Wu et al., 2024c) to convert the pretrained score function from the DDPM formulation to the EDM formulation (Karras et al., 2022). One may refer to the "Preconditioning" subsection in Appendix C.2 of (Wu et al., 2024c) for an explicit formula of the transformation deployed here.

**Inverse problem setups**   Below we provide a discussion on the mathematical formulations of the four inverse problems we tested on here.

**Super-resolution**   The forward model in (2.1) associated with the super-resolution problem we test on here can be written as

$$\boldsymbol{y} = P_f \boldsymbol{x} + \boldsymbol{n}$$

where $P_f \in \mathbb{R}^{\frac{n}{f} \times n}$ implements a block averaging filter that downscales each image by a factor of $f$ and $\boldsymbol{n} \sim \mathcal{N}(\boldsymbol{0}, 0.2\boldsymbol{I}_{\frac{n}{f}})$. Using similar setups as many previous work (Chung et al., 2022; Kawar et al., 2022; Wu et al., 2024c) on solving inverse problems via diffusion models, here we pick $f = 4$.

**Gaussian and motion deblurring**   The forward model associated with any deblurring problem can be summarized as

$$\boldsymbol{y} = B_k \boldsymbol{x} + \boldsymbol{n}$$

where $\boldsymbol{n} \sim \mathcal{N}(\boldsymbol{0}, 0.2\boldsymbol{I}_n)$ and $B_k \in \mathbb{R}^{n \times n}$ is a circulant matrix that realizes a convolution with the kernel $k$ under circular boundary condition. Again, we adopt the same settings used in most previous work (Chung et al., 2022; Kawar et al., 2022; Wu et al., 2024c).

Specifically, for the Gaussian deblurring problem, the convolutional kernel $k$ is fixed to be a Gaussian kernel of standard deviation 3.0 and size $61 \times 61$. For the motion deblurring problem, the kernel $k$ is randomly generated via code used in previous work (Kawar et al., 2022; Wu et al., 2024c), where the size is chosen to

---

[1]Pretrained score functions used in (Chung et al., 2022)

be $61 \times 61$ and the intensity is set to be 0.5. In order to ensure a fair comparison, we use the same motion kernel $k$ for each image across different methods.

**Box inpainting**  The forward model for the box inpainting problem is given by

$$\boldsymbol{y} = D\boldsymbol{x} + \boldsymbol{n}$$

where $\boldsymbol{n} \sim \mathcal{N}(\boldsymbol{0}, 0.2\boldsymbol{I}_n)$ and $D$ is a diagonal matrix with either 0 or 1 on its diagonal. In particular, here we choose $D$ such that a centered square patch of size $64 \times 64$ (*i.e.*, the side length is a quarter of the original image's side length) is masked out.

## D.2   Implementation details of AFDPS and all baseline methods

Regarding computing resources, all experiments included in this paper were conducted on NVIDIA RTX A100 and A6000 GPUs. A major part of the code implementing Algorithm 2 and 4 in this paper were adapted from the following Github repository[2]. Specifically, we used the same numerical discretization as that of the EDM framework (Karras et al., 2019), which is also deployed in (Wu et al., 2024c). One major difference is that we had tuned the terminal time to be $T = 8$ for both AFDPS-SDE (Alg. 2) and AFDPS-ODE (Alg. 4), while $T$ is set to be 80 for both the SGS-EDM method (Wu et al., 2024c) and the original EDM framework (Karras et al., 2022). Moreover, we increased the number of discretized timesteps as our methods avoids running multiple backward diffusion processes for different iterations. Specifically, for AFDPS-SDE the number particles and discretized timesteps were set to be 10 and 2000, respectively. For the AFDPS-ODE method, in order to control the total number of evaluations (NFEs), we set the number of particles, discretized timesteps and number of corrector steps at each time to be 5, 1000 and 4, respectively. Moreover, for both AFDPS-SDE (Algorithm 2) and AFDPS-ODE (Algorithm 4), we save computational cost by skipping the resampling step specified in Algorithm 1 in our implementation, which allows us to implement the dynamics of the particles' positions and weights in a parallel way. Finally, we return the particle associated with the largest weight as our best estimator of the recovered image. Given that we already take the logarithm of the weights in both AFDPS-SDE and AFDPS-ODE, they are guaranteed to remain numerically stable as time increases.

Here we further elaborate on the implementation details associated with the baseline methods. One thing to note is that two extra baselines are included in the extended numerical results presented in Tables 1 and 2 above. The following list provides an extended summary of these baselines and how we choose the parameters:

- *DPS (Chung et al., 2022)*: a method that performs posterior sampling by guiding the reverse diffusion process with manifold-constrained gradients derived from the measurement likelihood, enabling efficient inference in general noisy (non)linear inverse problems. We adopt most parameters usedin the default setting. The only difference is that we increase the number of discretized timesteps from 1000 to 1500, which helps make the method more tolerant of problems with higher observational noises
- *DCDP (Li et al., 2024b)*: a framework that alternates between data-consistent reconstruction and diffusion-based purification, which decouples data fidelity and prior sampling to improve flexibility and performance in image restoration tasks. In order to make the DCDP method adaptive to problems with higher observational noise, we change their settings by picking the number of iterations involved in both the data-reconstruction step and the diffusion-based purification step to be 100. Regarding the learning rates used for the data-reconstruction step, we have tuned them to yield the best possible performance. Specifically, the learning rates for the Gaussian deblurring, box inpainting, motion deblurring and super-resolution problems were set to be $10, 7, 10$ and $3$, respectively.
- *SGS-EDM (Wu et al., 2024c)*: a method that couples a split Gibbs sampler with a diffusion model, interpreting posterior inference as alternating between likelihood-based updates and Gaussian denoising via a learned generative prior. For the SGS-EDM method, we adopt the default setting used in (Wu et al., 2024c).
- *FK-Corrector (Skreta et al., 2025)*: a method that uses the Feynman-Kac formula to design corrector steps within a sequential Monte Carlo framework, improving the accuracy of samples from forward diffusion trajectories. We use the same set of parameters deployed in the AFDPS-SDE method by setting the number of particles and discretized timesteps to be 10 and 2000 as well, which ensures a fair comparison.

---

[2]Source code for (Wu et al., 2024c)

- *PF-SMC-DM (Dou & Song, 2024)*: a framework that formulates posterior sampling as a particle filtering problem, combining sequential Monte Carlo with diffusion models for efficient inference in high-dimensional spaces. Again, to ensure a fair comparison, we increase the number of particles and discretized timesteps to be 10 and 2000 for PF-SMC-DM as well.

# E  Additional Experimental Results and Discussions

In this section, we provide additional experimental results and detailed qualitative comparisons between our proposed methods and existing baselines.

**Summary.**  Across the diverse inverse problems evaluated on FFHQ-256 and ImageNet-256 (detailed in Table 1 and Table 2), the AFDPS framework consistently delivers strong results. The AFDPS-SDE variant, in particular, frequently distinguishes itself by producing visually compelling outcomes, excelling in the generation of sharp details and fine textures that contribute to high perceptual quality. This is evident in Figures 3-6, where AFDPS-SDE's reconstructions often appear more intricate and realistic. The AFDPS-ODE variant also provides coherent results, which are typically characterized by a notable smoothness. For tasks where capturing the utmost detail and textural accuracy is paramount, AFDPS-SDE often provides a particularly effective solution, frequently leading in or strongly competing for the best perceptual metrics (LPIPS).

**Gaussian Deblurring.**  In Gaussian deblurring, AFDPS-SDE showcases its ability to produce perceptually rich outputs, achieving the best LPIPS on ImageNet-256 (0.3925) and a competitive LPIPS on FFHQ-256 (0.2580). Figure 3 highlights SDE's strength in rendering sharp, defined textures like the dog's fur (ImageNet, row 2). Concurrently, AFDPS-ODE achieves high PSNR on both datasets and the best LPIPS on FFHQ-256 (24.98 PSNR, 0.2560 LPIPS), delivering notably clean and smooth outputs, for example, on the baby's facial skin (FFHQ, row 2).

**Motion Deblurring.**  For motion deblurring, AFDPS-SDE demonstrates strong perceptual quality, securing the best LPIPS score (0.2869) on FFHQ-256, while PF-SMC-DM leads in PSNR. Figure 4 emphasizes SDE's proficiency in transforming blurred images into sharp, detailed reconstructions, meticulously recovering fine details like individual hair strands in FFHQ portraits (e.g., row 5). AFDPS-ODE also effectively removes blur, yielding coherent results, typically with a characteristically smoother finish.

**Super-Resolution.**  AFDPS-SDE stands out in super-resolution, achieving the best PSNR and LPIPS scores on both FFHQ-256 (22.96 PSNR, 0.3063 LPIPS) and ImageNet-256 (20.97 PSNR, 0.4643 LPIPS). Figure 5 compellingly shows SDE generating sharp, highly detailed images from severely degraded inputs, adeptly reconstructing fine facial features (FFHQ, row 2 and 5) and intricate object textures like butterfly patterns (ImageNet, row 2). AFDPS-ODE also provides coherent upscaled outputs, especially for FFHQ dataset, reaffirming the metrics in the tables.

**Box Inpainting.**  In box inpainting combined with denoising, AFDPS-SDE shows robust performance, securing the highest PSNR on ImageNet-256 (23.15). Figure 6 highlights SDE's ability to generate detailed and realistically textured inpainted regions, such as the intricate dog fur (ImageNet, row 1) or sharp keyboard key structures (ImageNet, row 4). AFDPS-ODE also performs strongly, achieving best LPIPS on both datasets (FFHQ: 0.1969, ImageNet: 0.2716) and best PSNR on FFHQ (25.73), producing notably smooth and coherent fills, like seamless facial features (FFHQ, row 1).

Measurement  Ours (SDE)  Ours (ODE) Ground Truth  Measurement  Ours (SDE)  Ours (ODE) Ground Truth

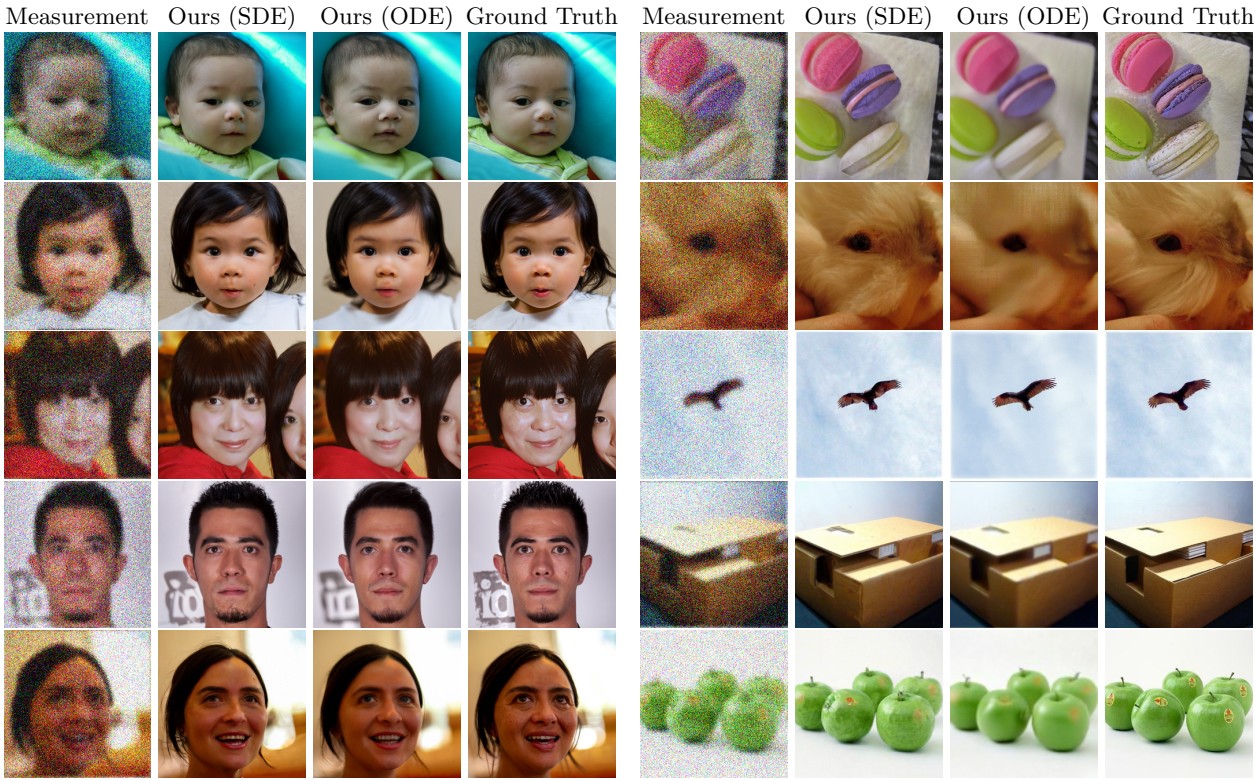

Figure 3: Additional visual examples for the Gaussian deblurring problem on FFHQ and ImageNet.

Measurement  Ours (SDE)  Ours (ODE) Ground Truth  Measurement  Ours (SDE)  Ours (ODE) Ground Truth

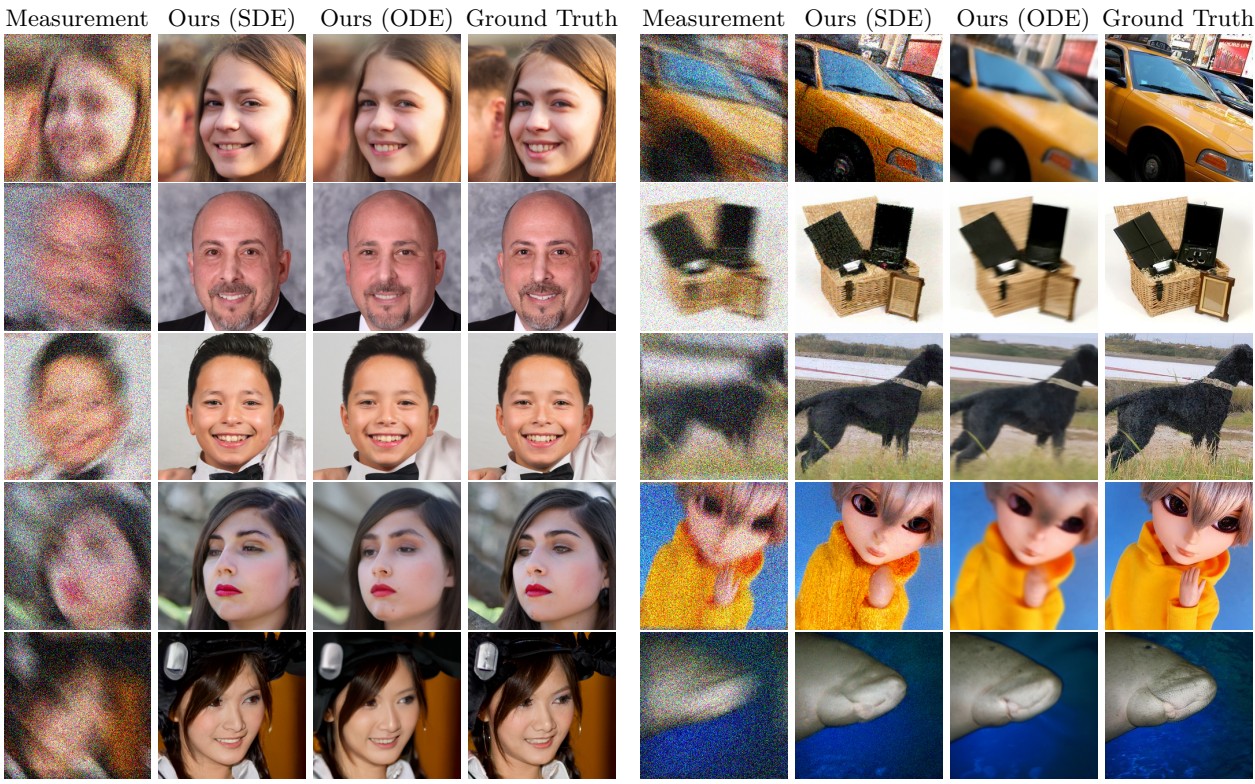

Figure 4: Additional visual examples for the motion deblurring problem on FFHQ and ImageNet.

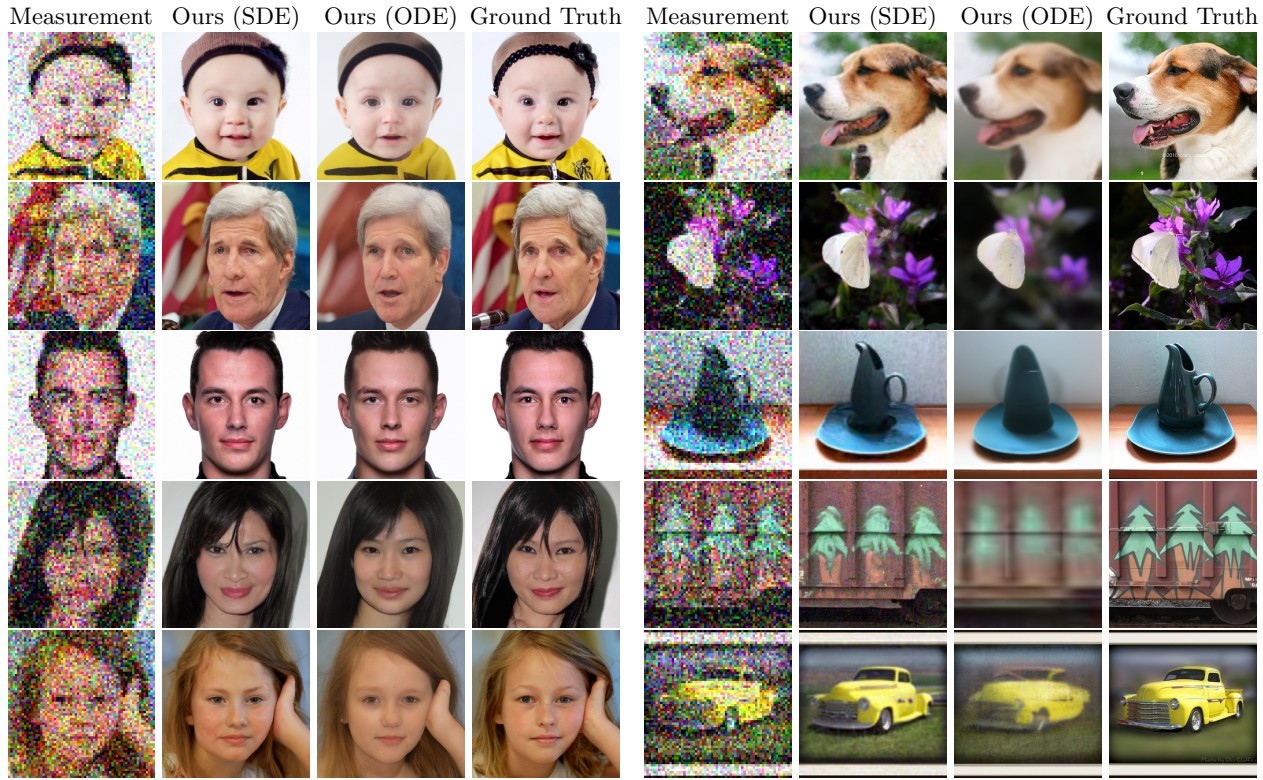

Figure 5: Additional visual examples for the super-resolution problem on FFHQ and ImageNet.

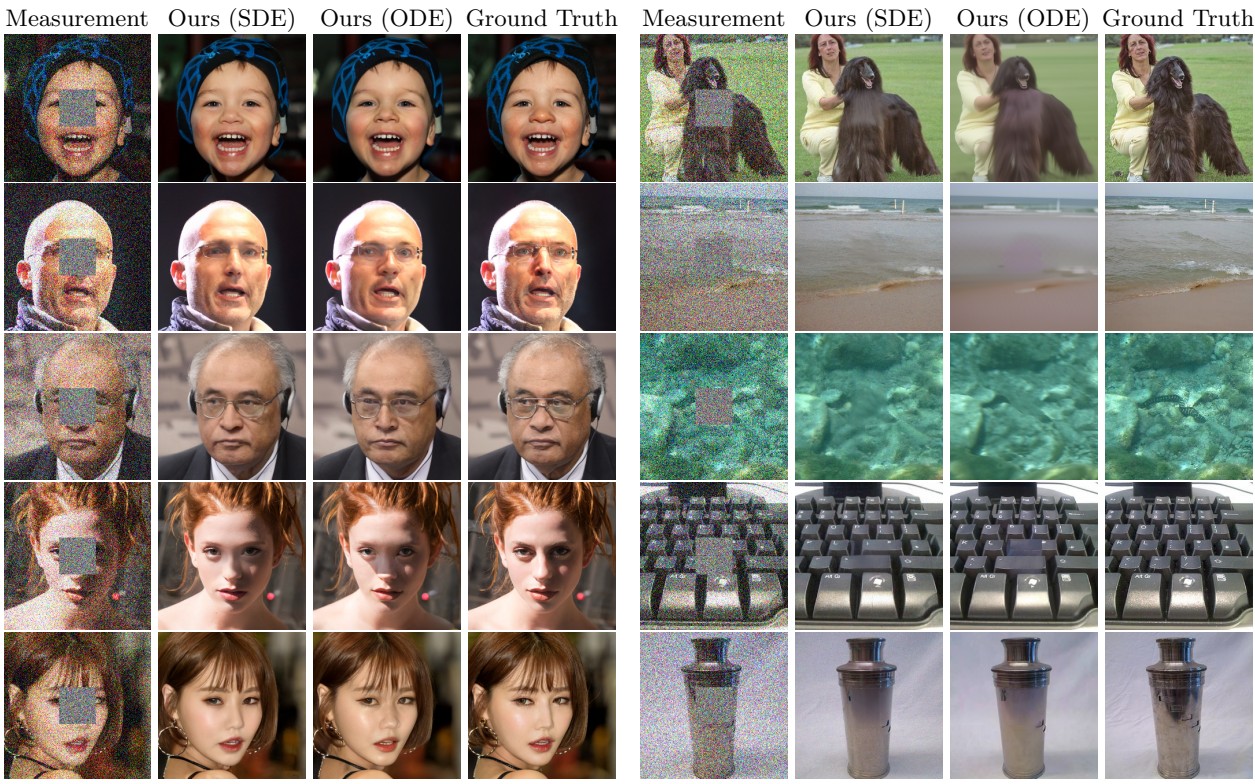

Figure 6: Additional visual examples for the box inpainting problem on FFHQ and ImageNet.

# F    Supplementary Experiments

In order to better justify the effectiveness of our proposed methodology, we tested our methods on an additional set of tasks, such as validation on a larger set of images, nonlinear inverse problem and tuning of hyperparameters. Details of both the experimental settings and results are included in the following four subsections. Moreover, we note all the AFDPS method implemented below are slightly different compared to the version we used earlier. Specifically, the parametrized curve $\alpha : [0, T] \to [0, 1]$ is picked to be the anenaling schedule $\alpha(t) := \frac{t}{T}$ instead of the uniform constant $\alpha(t) \equiv 1$. In the first subsection below, we evaluate the accuracy of AFDPS in sampling the posterior distribution on a simulated compressed sensing problem.

## F.1    Synthetic experiment on compressed sensing

Similar to the synthetic experiment conducted in subsection 4.1 of (Wu et al., 2024c),, the measurement model is constructed by $\boldsymbol{y} = \frac{1}{\sqrt{n}}A\boldsymbol{x} + \boldsymbol{n} \in \mathbb{R}^m$, where $\boldsymbol{n} \sim \mathcal{N}(\boldsymbol{0}, 0.5\boldsymbol{I}_m)$ and the matrix $A \in \mathbb{R}^{m \times n}$ satisfies $A_{ij} \sim \mathcal{N}(0, 1)$ for any $1 \leq i \leq m$ and $1 \leq j \leq n$. The prior distribution on $\boldsymbol{x}$ is further modeled as $\boldsymbol{x} \sim \mathcal{N}(\mu, \boldsymbol{I}_n)$, where $\mu$ is picked to be the first image in the MNIST dataset. Under this setting, we have that the posterior distribution $p(\boldsymbol{x}|\boldsymbol{y})$ is analytically tractable. Just as the experiment presented in section 4.1 of (Ren et al., 2025b), we evaluate the ensemble of weighted particles generated by AFDPS with respect to both the Maximum Mean Discrepancy (MMD) and Sliced Wsserstein Distance (SWD). Here we adopt the SDE-based implementation in Algorithm 2 and set the threshold on ESS for the resampling step to be 0.8. From Figure 7, we can see that AFDPS does converge to the target posterior distribution as the number of particles increases, as desired.

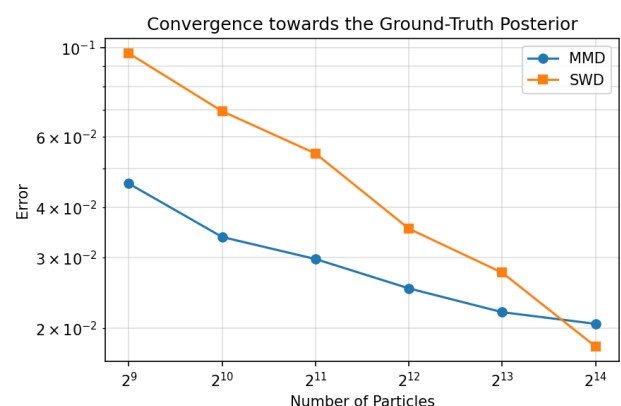

Figure 7: Convergence of AFDPS towards the ground-truth posterior versus number of particles.

| Ours (AFDPS) | Baseline (SGS) | Ground Truth | Ours (AFDPS) | Baseline (SGS) | Ground Truth |

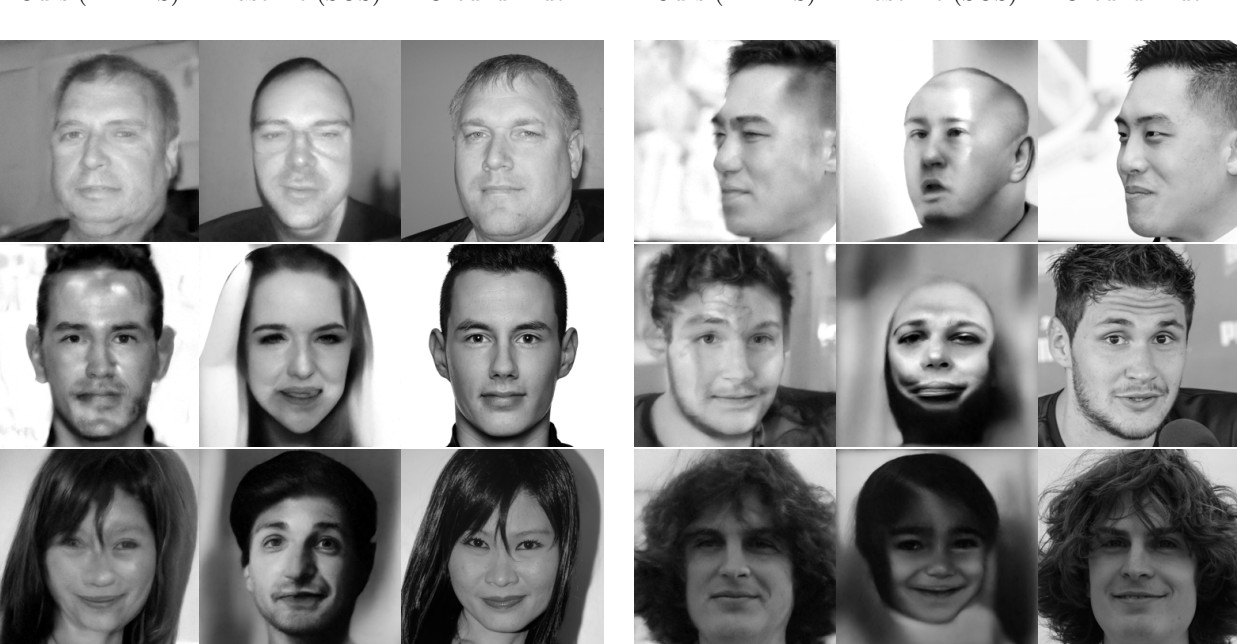

Figure 8: Comparison between AFDPS and SGS-EDM (Wu et al., 2024c) on Fourier Phase Retrieval.

### F.2 Nonlinear inverse problems

The second subsection here is devoted to testing the proposed AFDPS method on a challenging nonlinear inverse problem. Here we consider the Fourier phase retrieval (FPR) problem, which is also tested in (Chung & Ye, 2022; Wu et al., 2024c). Specifically, by using a matrix $F$ to represent the 2D Fourier Transform, we have that the measurement model can be expressed as $\boldsymbol{y} = \|FP\boldsymbol{x}\|_2 + \boldsymbol{n}$, where $P$ denotes the oversampling matrix that pads the input $\boldsymbol{x}$ with zero entries in 2D matrix form. Here we set the oversampling ratio to be 8 and the observed Gaussian noise $\mathcal{N}(\boldsymbol{0}, 0.2\boldsymbol{I}_m)$. We note that the inverse problem tested in this subsection is more challenging compared to the one tested on SGS-EDM in subsection 4.2 of (Wu et al., 2024c), as the observational noise has much larger variance. Both AFDPS and SGS-EDM are then tested on the first 100 images from the FFHQ dataset. The baseline SGS-EDM achieves an avrage PSNR value of 12.63 and an average LPIPS value of 0.4052. In contrast, our method AFDPS yields an average PSNR value of 12.63 and an average LPIPS value of 0.4680, which outperforms SGS-EDM. A few examples of recovered images returned by both methods are also exhibited in 8 above, which demonstrates the supremacy of our method compared to SGS-EDM.

### F.3 Extra baseline and validation on a larger dataset

The third subsection here focuses on comparing AFDPS with the Decoupled Annealing Posterior Sampling (DAPS) algorithm proposed in a recent work (Zhang et al., 2025a). Following the experimental setup described in Appendix D above, we evaluate the proposed method on Gaussian and motion deblurring problems using the first 100 images of the FFHQ-256 dataset. The corresponding PSNR and LPIPS scores are reported in Table 4. From the results we can see that AFDPS does surpass DAPS on the two deblurring problems.

Table 4: Results of the comparison between DAPS and AFDPS on two deblurring problems.

| Method | Gaussian Deblurring | | Motion Deblurring | |
|---|---|---|---|---|
| | PSNR ($\uparrow$) | LPIPS ($\downarrow$) | PSNR ($\uparrow$) | LPIPS ($\downarrow$) |
| DAPS (100 images) | 22.10 | 0.4580 | 23.02 | 0.4661 |
| **AFDPS-SDE (100 images)** | 25.19 | 0.1291 | 24.07 | 0.1530 |
| **AFDPS-SDE (1000 images)** | 25.11 | 0.1329 | 24.00 | 0.1571 |

To further validate the effectiveness of AFDPS, we extend our evaluation to the first 1,000 images from the FFHQ-256 dataset. We also include the super-resolution task alongside the two deblurring problems above. For the added super-resolution task, AFDPS achieves an average PSNR value of 23.13 and an average LPIPS value of 0.1775 on the first 1000 images of the FFHQ-256 dataset, which demonstrates the effectiveness of AFDPS. An additional set of visual examples from the first 1000 images of FFHQ-256 on the three tasks are also included in Figures 9, 10 and 11 below, respectively.

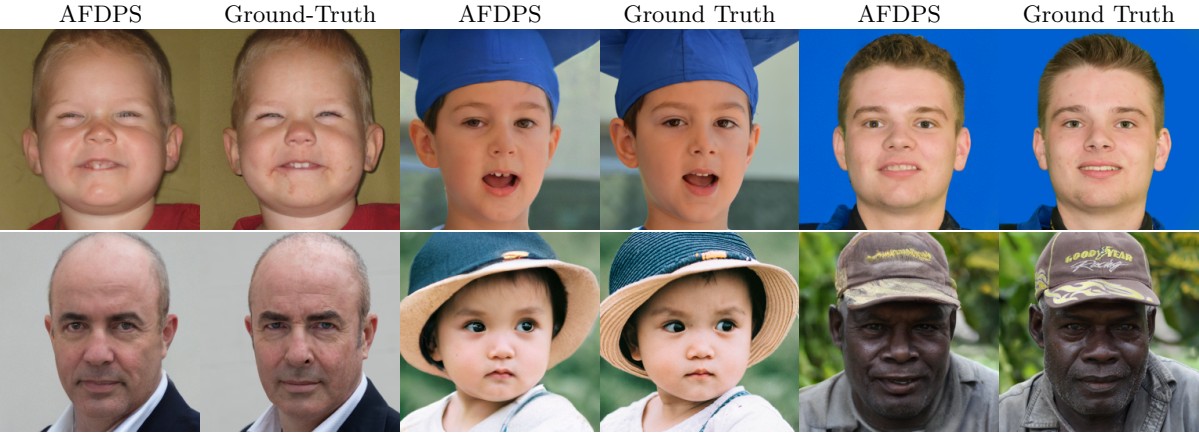

Figure 9: Additional visual examples for the Gaussian deblurring problem on 1000 images from FFHQ

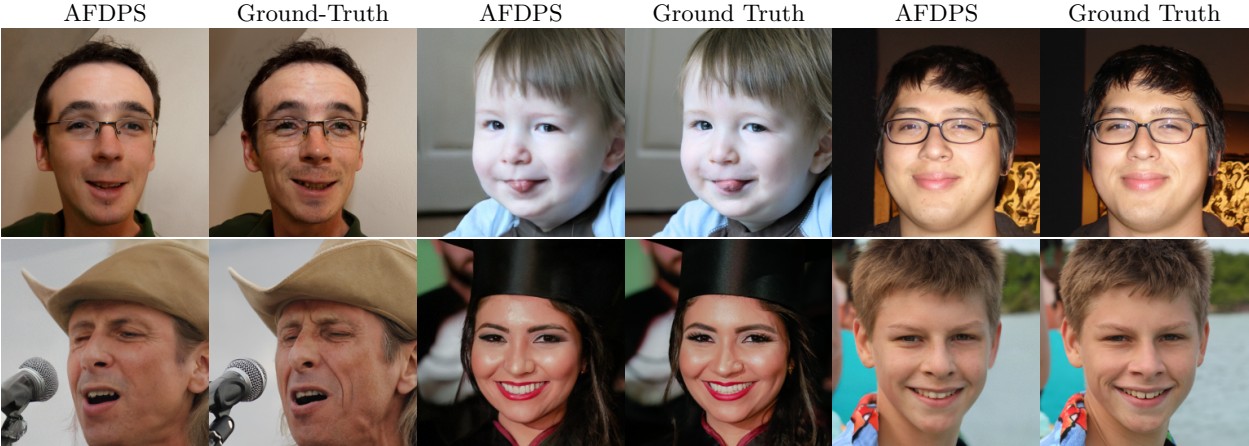

Figure 10: Additional visual examples for the motion deblurring problem on 1000 images from FFHQ

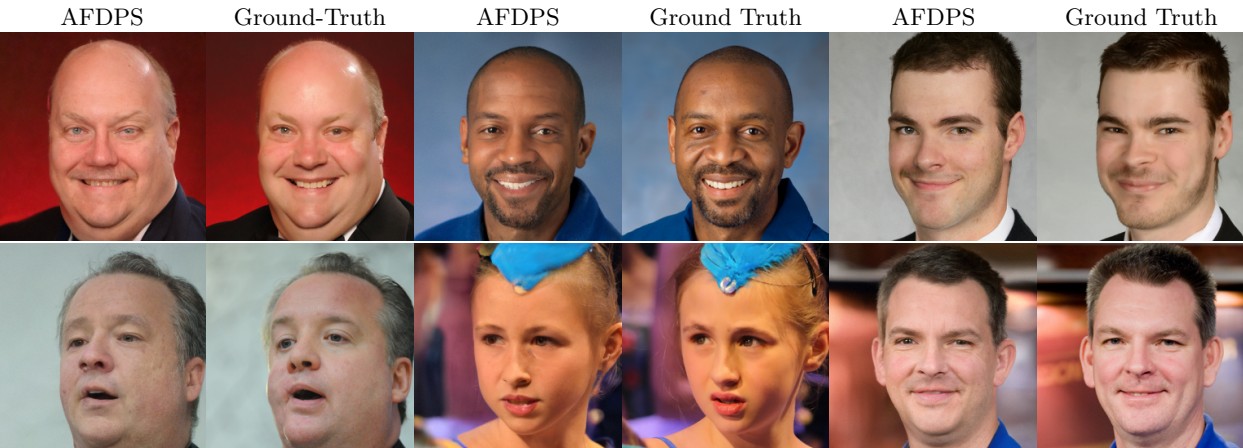

Figure 11: Additional visual examples for the super-resolution problem on 1000 images from FFHQ

## F.4 Tuning and studies of other hyperparameters

In the last subsection, we investigate the impact of a few important parameters on the performance of the AFDPS algorithm. Throughout this subsection, we focus on the Gaussian deblurring problem applied to the first 100 images of the FFHQ-256 dataset and fix the number of discretized timesteps to be 500.

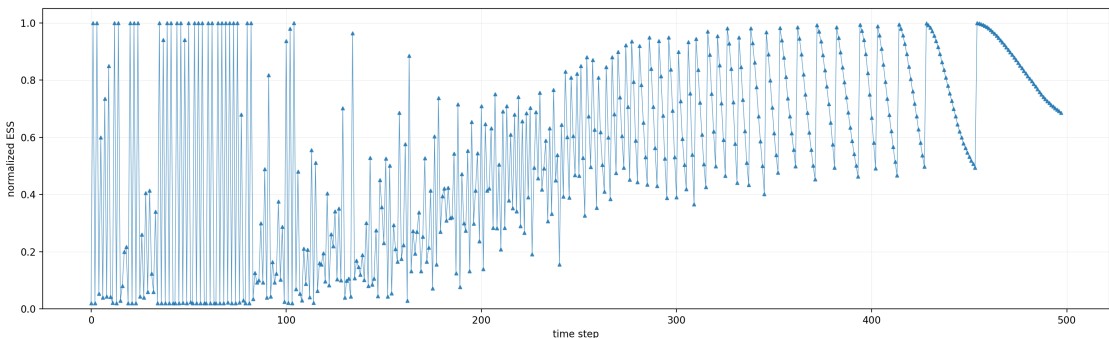

Figure 12: Plot of ESS with respect to time on the Gaussian deblurring task

The first parameter we investigate here is the Effective Sample Size (ESS). Here we fix the number of particles to be 50. By setting the threshold for resampling to be 0.5 and plotting the ESS with respect to the discretized timestep, we obtain Figure 12 above. From the plot we can see that the ESS oscillates during the first half of the inference time. For the second half of the inference time, ESS gradually decays and gets periodically restored through resampling, which is similar to the behavior exhibited in Figure 2 of (Ren et al., 2025b).

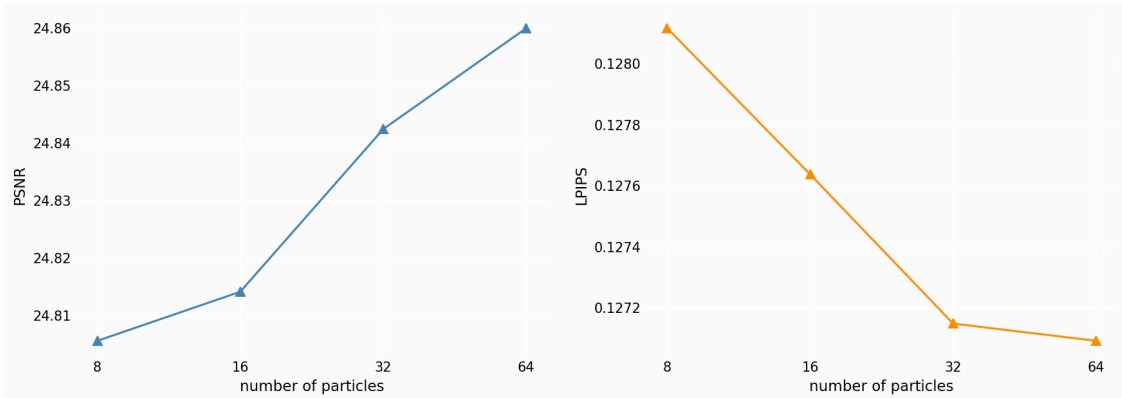

Figure 13: Tuning the number of particles in AFDPS-SDE on the Gaussian deblurring task

Secondly, we study how the change in the number of particles might impact the AFDPS-SDE algorithm's performance in terms of the PSNR and LPIPS metrics. From the two plots in Figure 12, we can see that image quality gets improved as the number of particles increases. Moreover, the gains are marginal as the number of particles becomes sufficiently large, suggesting that around 50 particles can yield a balance between the reconstruction quality and computational cost.

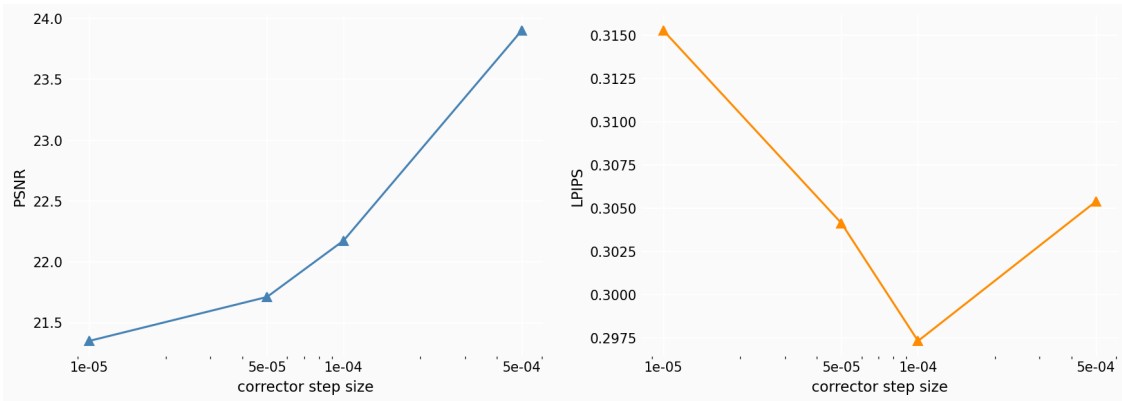

Figure 14: Tuning the stepsize of the corrector in AFDPS-ODE on the Gaussian deblurring task

Finally, we investigate the impact of the corrector stepsize on the performance of the AFDPS-ODE algorithm. Here we fix the number of particles, the number of discretized timesteps and the number of corrector steps to be 50, 500 and 5, respectively. Just as Figure 13 above, we plot both the PSNR and the LPIPS metric with respect to different stepsizes. From the plots in Figure 14, we have that larger stepsizes for the corrector step in general yield images of better quality. With additional computational resource, it would also be interesting to investigate how the number of corrector steps might impact the algorithm's performance in the future.

