# OpenReview forum: "Solving Inverse Problems via Diffusion-Based Priors: An Approximation-Free Ensemble Sampling Approach"
_TMLR — Accepted by TMLR_

### Review · Reviewer_Lf7J · 2025-11-06

**Summary Of Contributions:**

This work aims to solve a Bayesian inverse problem with diffusion models. It starts with a diffusion process of the prior distribution (i.e., the distribution of raw images) and derives the corresponding diffusion process of the posterior distribution. This posterior distribution is then sampled via partial SMC. Theoretical investigation is conducted to assess the convergence rate.  Empirical results are presented with reasonable performance. Compared with prior similar works, it extends to nonlinear cases and also provides a rigorous analysis.
1. Section 2.2. introduces the EDM framework with **general** F and G functions. But later in this section, it apparently refers to the specific case that $p_T$ converges to a Gaussian noise distribution. Also, it writes "$\widehat{\overleftarrow{p_0}}$ being exactly Gaussian", but in the proof of Theorem 4.1, its distribution is $p_0*N(0, T^2I_n)$. I think the writing here is not rigorous and may cause confusion to readers unfamiliar with the context.
2. Although the paper focuses on general inverse problems, all simulations are about linear inverse problems. Maybe it is possible to add a simulation on nonlinear models (even without a baseline)
3. Tightness of the upper bound in Thm 4.1. Even if $\epsilon_s^2=0$, the upper bound in Thm 4.1 doesn't diminish to 0. Just curious whether a tighter bound can be derived.
4. Can authors discuss the choice of $N$? Thm 4.2 requires $N\rightarrow\infty$ while simulations use $N=5$ or 10.
5. In Figure 2 (and appendix examples), are the reconstructed images the average across $N$ particles or just one random particle? Please specify.
6. Assumption 4.4 deserves more discussion or explanations in the main text. It is different to understand the purpose of this assumption (or quantity $I(x,t)$) without reading the proof of Theorem 4.2

**Additional Comments:**

none

**Audience:**

Yes

**Audience Explanation:**

I believe the theoretical analysis presented in this paper is a nice addition to the literature. To the best of my knowledge, this is the first of its kind.

**Claims And Evidence:**

Yes

**Claims Explanation:**

The paper provides a rigorous and sound theoretical justification and convincing numerical results.

**Requested Changes:**

Please refer to my comments. I request changes corresponding to 1.4.5.6. It would be nice to see an additional simulation based on my comment #2, if feasible

---

> ### Author Response · Authors · 2025-12-19
> **Rebuttal (Part 1)**
>
> We would like to thank the reviewer’s insightful comments and suggestions on our work. In the following paragraphs, we address the six questions listed in the review.
>
> (1) Thank you so much for pointing this out. We will make sure to revise the writing so that subsection 2.2 discusses the general EDM framework [1] parametrized by the two functions $s$ and $\sigma$ (where $F = \frac{\dot{s}}{s}$ and $G = s\sqrt{2\dot{\sigma}\sigma}$). Specifically, the phrase "$\widehat{\overleftarrow{p_0}}$ being exactly Gaussian" in subsection 2.2 will be changed to "$\widehat{\overleftarrow{p_0}}$ being an approximation of the distribution $p_T$". At the end of subsection 2.2, we will clarify that our focus for the remaining parts of the paper will be on the special cases when $p_T$ converges to some Gaussian distribution (For instance, $p_T$ is given by $p_0 \ast \mathcal{N}(0,\sigma(t)^2I)$ when $s(t)=1$, which is proved in Appendix B.1 of [1]), which are most widely used in practice.
>
> Regarding the theoretical results presented in Section 4, we would like to gently note that the second-to-last sentence of the first paragraph in Section 4 already clarifies that our analysis applies only to the special case when $s(t) = 1$ and $\sigma(t) = t$. We will ensure that all theorems in Section 4 are revised accordingly to clearly state that the analysis is restricted to this special case.
>
>
> (2) We thank the reviewer for bringing this to our attention. Currently, we are evaluating our method on two nonlinear inverse problems - Fourier phase retrieval (FPR) and coded diffraction patterns (CDP) - under similar experimental settings (dataset: FFHQ-256 grayscale images) as in [2]. We will update the rebuttal once we complete our experiments on these two tasks and revise the manuscript accordingly.
>
> (3) Thanks for raising this question. We would like to kindly note that the running time $T$ does depend on the score matching error $\epsilon_{s}$. By picking $T = \Theta(\sqrt{\epsilon_{s}^{-1}})$, we then obtain an asymptotic upper bound of magnitude $\Theta(\sqrt{\epsilon_{s}})$. Hence, the upper bound in Theorem 4.1 can be arbitrarily small as $\epsilon_s \rightarrow 0$. We do agree that it will be interesting to investigate whether a tighter bound (in terms of the dependency on $\epsilon_s$) can be achieved or not, but this is beyond the scope of this paper. We will revise the manuscript accordingly to include this as future work.

---

> ### Author Response · Authors · 2025-12-19
> **Rebuttal (Part 2)**
>
> (4) We appreciate the reviewer’s question regarding how the number of particles (denoted by $N$) affects the method’s performance. We are currently adding an extra numerical experiment (hyperparameter tuning) to assess our method's sensitivity with respect to the number of particles used in practice. The experimental setting is similar to that in Section 5.3 of [3]​, which considers a single inverse problem on the FFHQ-256 dataset. Again, once we complete experiments across a range of different $N$'s, we will incorporate the results into the rebuttal and revise the manuscript accordingly, including a discussion on how to balance the algorithm's performance against constraints on memory as $N$ increases. Currently some preliminary results show that the PSNR metric does get improved as $N$ increases, but the improvement becomes much less pronounced when $N$ reaches $15$.
>
> (5) Thanks for bringing up this question. We would like to clarify that in our ensemble, the particle with the largest weight is returned as the final estimator of the recovered image. This is closely analogous to the best-of-$N$ strategy (i.e., selecting the best-performing sample among candidates). Such selection rule is also stated in the second-to-last sentence of the first paragraph in Appendix D.2 and applies to all images shown in Figure 2 and the Appendix. We will ensure that the main text is revised to make this selection criterion explicit.
>
> (6) We would like to thank the reviewer for pointing this out. We will revise the manuscript to clearly explain, for the special case when $s(t)=1$, $\sigma(t) =t$, how the function $I(x,t)$ defined in Assumption 4.4 relates to the function $U - \widehat{H}^T\nabla\mu_{y}$ in (3.5), which governs the evolution of the particle weights. Specifically, a direct computation yields that:
>
> $$U - \widehat{H}^T\nabla_{x}\mu_{y} = \frac{2t}{2}(||\nabla\mu_{y}||_2^2 - \Delta\mu) -\frac{4t}{2}\phi^T \nabla\mu = tI(x,t).$$
> We will ensure that the above computation is included and properly discussed in the revised manuscript. Moreover, we will expand the discussion of Assumption 4.4 in the main text by relating it to Assumption 1 in [4]. Both assumptions serve to control the particle weights during evolution and thereby mitigate weight degeneracy.
>
> Overall, we thank the reviewer again for the thoughtful review, which has greatly helped us improve the quality of our paper. We will make sure to post another update once we finish revising the manuscript.
>
> ### References
>
> [1] Karras, Tero, et al. "Elucidating the design space of diffusion-based generative models." Advances in neural information processing systems 35 (2022): 26565-26577.
>
> [2] Wu, Zihui, et al. "Principled probabilistic imaging using diffusion models as plug-and-play priors." Advances in Neural Information Processing Systems 37 (2024): 118389-118427.
>
> [3] Dou, Zehao, and Yang Song. "Diffusion posterior sampling for linear inverse problem solving: A filtering perspective." The Twelfth International Conference on Learning Representations. 2024.
>
> [4] Domingo-Enrich, Carles, et al. "A mean-field analysis of two-player zero-sum games." Advances in neural information processing systems 33 (2020): 20215-20226.

---

### Review · Reviewer_5k5w · 2025-11-12

**Summary Of Contributions:**

The paper focuses on the problem of posterior inference for inverse problems with Diffusion model priors. The authors propose a particle/ensemble-based approach based on the Sequential Monte Carlo-framework (SMC). After deriving a partial differential equation (PDE) describing the time evolution of the posterior distribution, the paper proposes two methods: a stochastic differential equation and an ordinary differential equation+corrector method, which both are claimed to yield samples from the exact posterior without the need for approximations.

The paper provides two theoretical results: Theorem 4.1, which upper bounds the total variation-distance between the approximate and exact posterior, and Theorem 4.2, which states that the distribution of a particle converges to the solution of the PDE as the number of particles goes to infinity. Finally, the paper also provides a set of numerical experiments comparing the proposed methods to relevant baseline methods from the literature using the FFHQ-256 and ImageNet-256 datasets.

The paper addresses an important problem, which is of interest to a wide audience in science and engineering. The paper is generally well-written and well-structured, and the authors provide both theoretical and numerical evidence for their proposed method.
However, the significance of the theoretical results is not clearly described, and the numerical experiments could be much stronger from a statistical point of view.

**Additional Comments:**

- Abstract: Something is missing in the sentence: "Theoretically, we
prove that the error between the true posterior distribution can be bounded..."
- Introduction: ILVR abbreviation is undefined
- Section 2.2: Adding quotation marks will make the line "We adopt the Elucidating the design space of Diffusion Models (EDM) framework from ..." more readable
- In eq. (3.1), it is clear that \hat{Z}_y(t) always exists?
- In the text above eq. (3.3), the relation should be proportional to rather than equality between \hat{\arrow{p}}_t(x) and the r.h.s. since the right hand side is unnormalized as far as I can tell.
- Where are \beta_t and \gamma_t defined? I assume they represent the importance weights and target distribution for SMC, respectively, but it is not stated anywhere in the main text.

**Audience:**

Yes

**Audience Explanation:**

The paper addresses an important problem, which is of interest to a wide audience in science and engineering, including the TMLR audience.

**Broader Impact Concerns:**

None.

**Claims And Evidence:**

No

**Claims Explanation:**

My answer to the question: "Are the claims made in the submission supported by accurate, convincing and clear evidence?" would be "in some ways yes and some ways no". I'll elaborate in the following.

The proposed methods are well-described and sound, and they are supported by both theoretical and numerical results.

Theorem 4.2 is a nice sanity that the approximation "does the right thing" if given enough resources. The theorem many relies on for assumptions A4.1-A4.4. While assumptions A4.1-A4.3 are relatively easy to understand, assumption A4.4 is less clear, and there is no discussion of how strong this assumption is compared to a realistic set-up. Moreover, for non-linear problems, the author states that one can plug in standard sampling methods (MALA, AIS etc.) to sample from the initial distribution q_y(x, 0), and hence, the samples will only be approximately correct and not exact like for the linear Gaussian case. It is unclear if and how this affects the theoretical results.

For the numerical experiments, the authors consider classic inverse problems (deblurring, inpainting, superresolution etc.) and compare the proposed method against a reasonable set of baselines for the well-known datasets FFHQ-256 and ImageNet256. However, the chosen performance metrics are evaluated on a validation set of 100 images. First of all, it is unclear if the same validation set has been used for model selection or tuning. Why not use a proper test set? Second of all, why use such a small validation/test set? Finally, the authors do not report any kind of uncertainty measures or error bars for the numbers in Tables 1 & 2, which is crucial for such a small dataset, and hence, it is hard to assess the effect of the proposed method.

**Requested Changes:**

- Please make sure all symbols, notation, and abbreviation are properly defined (Abbreviation "ILVR" is not defined, \beta_t is not defined in the main text etc.)
- The details of how the diffusion methods are integrated in the SMC-framework are somewhat unclear from the main text. Please clarify this.
- The paper proposes two algorithmic variants: the SDE and the ODE variant. However, there is no discussion of the pros and cons.  Please add a discussion on this.
- While the paper does a really could job listing some of the related work, it is somewhat unclear how the proposed methods differ from other work that combines SMC with diffusion model priors for inverse problems. E.g. Kelvinius et al. 2025. Please elaborate on how your methods are different from the methods in the related work.
- It is really nice that the authors provide both theoretical and numerical evidence. However, I think the theoretical results would be more appreciated if the author included a small discussion of their significance as well as a discussion of the assumption A4.4 (as described above)
- The numerical experiments could be made much stronger by using an independent test set (or at least make it clear that you are using an independent test set), use more than 100 samples and report error bars for important metrics. This is necessary for me to recommend acceptance.
- In Section 3.1, the authors state that many general-purpose samplers can be used to generate samples from the initial distribution \hat{q}_y(x, 0). However, it is unclear which sampler the authors propose, and moreover, how are the theoretical results affected if e.g. the MALA sampler does not converge?
- The experiments focus only on the performance metrics, but I think it would be equally interesting to include experiments that shine light on the inner-working of the algorithms. For example, how do the effective sample sizes evolve during the experiments? How sensitive is the ODE+corrector method to the configuration of ULA?
- I can't find the value that the author used for the ESS threshold c anywhere.

---

> ### Author Response · Authors · 2026-01-05
> **Rebuttal (Part 1)**
>
> We sincerely thank the reviewer for the constructive review of our paper. Below we discuss the reviewer’s concerns one at a time.
>
> (1) Questions about the case of nonlinear inverse problems
>
> We appreciate the reviewer's comment on the case of nonlinear inverse problems. The reviewer is correct that, for the PDE dynamics presented in the manuscript, the initial distribution $\widehat{q}_y(x,0)$ can only be approximately sampled via plug-in samplers like MALA and AIS. (These are the two most commonly used samplers in practice. We will make sure to state clearly that they are used for sampling the initial distribution in section 3.1 of the revised manuscript.)
>
> We note that this can be incorporated into our analysis by adding an initialization error term that quantifies the discrepancy between the true initial distribution and its approximation produced by the plug-in sampler. In fact, such extra initialization error is related to the term $D_{KL}(\rho_{0} || \widetilde{\rho}_{0})$ in equation (C.1) of Lemma (C.2). Moreover, we would like to remark that such initialization error can also be eliminated by changing the time-dependent distribution path and its associated PDE dynamics.
>
> Specifically, let $\alpha: [0,T] \rightarrow [0,1]$ be a parametrized curve and write $\alpha(t) = \alpha_t$ (For instance, a common choice is the linear function $\alpha(t) = \frac{t}{T}$). Compared to the PDE dynamics described in subsection 3.1 and Figure 1, below we consider a more generic distribution path $q_{\alpha}(x,t) \propto \overleftarrow{p_t}(x)e^{-\alpha(t)\mu(x)}$ ($t \in [0,T]$) parametrized by the curve $\alpha$. Using notations similar to those used in the manuscript, we denote the time-dependent unnormalized density by $Q_{\alpha}(x,t) := \overleftarrow{p_t}(x)e^{-\alpha(t)\mu(x)}$. Below we will provide a sketch of the derivations that lead to an $\alpha$-dependent PDE dynamics governing the evolution of $q_{\alpha}(x,t)$ for $t \in [0,T]$ by following the steps outlined in Stage II and Figure 1 of subsection 3.1. More precisely, we again start with the Fokker Planck PDE governing the evolution of the prior distribution $\overleftarrow{p_t}$:
>
> $$\frac{\partial}{\partial t}\overleftarrow{p_t} = -\nabla_{x} \cdot (H(x,t)\overleftarrow{p_t}) + \frac{1}{2}V(t)^2\Delta_{x}\overleftarrow{p_t}.$$
>
> where $H(x,t):= -F(t)x + V(t)^2\phi_{\theta}(x,t)$ above denotes the same function as defined in subsection 3.1 (Stage II) of the paper. By substituting $\overleftarrow{p_t}(x) = Q_{\alpha}(x,t)e^{\alpha(t)\mu(x)}$ into the PDE above and simplifying the equation via chain rule, we further obtain that
>
> $$(\frac{\partial}{\partial t}Q_{\alpha}(x,t))e^{\alpha(t)\mu(x)} + Q_{\alpha}(x,t)e^{\alpha(t)\mu(x)}(\alpha'(t)\mu(x)) = \frac{\partial}{\partial t}(Q_{\alpha}(x,t)e^{\alpha(t)\mu(x)}) = -\nabla_{x} \cdot (H(x,t)(Q_{\alpha}(x,t)e^{\alpha(t)\mu(x)})) + \frac{1}{2}V(t)^2\Delta_{x}(Q_{\alpha}(x,t)e^{\alpha(t)\mu(x)}).$$
>
> By moving the second term on the LHS of the equation above to the RHS and following similar computational steps in the proof of Lemma B.1, we can further simplify the equation above to obtain the PDE governing the evolution of $Q_{\alpha}(x,t)$ as below:
>
> $$\frac{\partial}{\partial t}Q_{\alpha}  = -\nabla_{x} \cdot ((H(x,t) - V(t)^2\nabla_{x}\mu)Q_{\alpha}) + \frac{1}{2}V(t)^2\Delta_{x}Q_{\alpha} + (U(x,t) - H(x,t)^T\nabla_{x}\mu-\alpha'(t)\mu(x))Q_{\alpha}.$$
>
> where $U(x,t):= \frac{1}{2}V(t)^2(||\nabla_{x}\mu||^2 - \Delta_{x}\mu)$ above denotes the same function as defined in subsection 3.1 (Stage II) of the manuscript. By applying Lemma B.2 in the Appendix again to "normalize" the PDE dynamics above, we finally obtain the PDE dynamics governing the evolution of the $\alpha$-parametrized density $q_{\alpha}(x,t)$ as follows:
>
> $$\frac{\partial}{\partial t}q_{\alpha} = -\nabla_{x} \cdot ((H(x,t) - V(t)^2\nabla_{x}\mu)q_{\alpha}) + \frac{1}{2}V(t)^2\Delta_{x}q_{\alpha} + ((U(x,t) - H(x,t)^T\nabla_{x}\mu-\alpha'(t)\mu(x)) - (\int(U(x,t) - H(x,t)^T\nabla_{x}\mu-\alpha'(t)\mu(x))q_{\alpha}(x)dx))q_{\alpha}$$
>
> As we can see from the equation above, an extra linear term $(-\alpha'(t)\mu(x) + \int \alpha'(t)\mu(x)q_{\alpha}(x)dx))q_{\alpha}$ is induced by the curve $\alpha$. For the special case when $\alpha \equiv 1$ is constant, the equation above reduced to the PDE dynamics presented in the current manuscript. In contrast, any curve $\alpha$ satisfying $\alpha(0) = 0$ and $\alpha(T) = 1$ yields a distribution path that starts with $\overleftarrow{p_0}$ (Gaussian distribution) and ends at the target posterior distribution (propotional to $\overleftarrow{p_T}e^{\mu}$), which resolves the issue that the initial distribution may not be exactly sampled.

---

> ### Author Response · Authors · 2026-01-05
> **Rebuttal (Part 2)**
>
> Overall, we will update both the theoretical and empirical parts of the manuscript to incorporate the more general PDE dynamics presented above, with a particular focus on the special cases when $\alpha \equiv 1$ (PDE dynamics in the current manuscript, which is more suitable for Gaussian linear inverse problems as the initial distribution can be exactly sampled) or $\alpha(0) = 0, \alpha(T) = 1$ (new PDE dynamics to be added, which is more suitable for nonlinear inverse problems). For the empirical side on nonlinear inverse problems, we are currently running additional experiments on two nonlinear inverse problems (Fourier phase retrieval and coded diffraction patterns, i.e., FPR and CDP) using the FFHQ-256 grayscale dataset under the same experimental settings as in [1].
>
> (2) Question about existence of the normalization constant $\widehat{Z}_{y}(t)$ in equation (3.1)
>
> We thank the reviewer for bringing this to our attention. Below we provide a proof sketch of the statement under assumption 4.1, which stipulates that the log-likehood function $\mu_y$ is bounded below by some constant $C^{(1)}(y)$ for any observation $y$. Then from the definition of $\widehat{Z}_{y}(t)$ we have the following inequality for any $t$:
>
> $$\widehat{Z}_{y}(t) = \int \widehat{Q}_y(x,t)dx \leq \int \widehat{\overleftarrow{p_t}}(x) e^{-C^{(1)}(y)}dx = e^{-C^{(1)}(y)} < \infty.$$
>
> where the inequality above follows from Assumption 4.1. In the revised manuscript, we will incorporate the above proof into the first paragraph of Subsection 3.1 after introducing the definition of $\widehat{Z}_{y}(t)$.
>
>
> (3) Question about the identity $\widehat{\overleftarrow{p_t}}(x) = \widehat{Q}_y(x,t)e^{\mu}$ above equation (3.3)
>
> Thank you so much for raising this question. We would like to kindly clarify that the identity in the current manuscript is indeed correct. Specifically, the reviewer is correct that the term $\widehat{Q}_y(x,t) = \widehat{\overleftarrow{p_t}}(x)e^{-\mu}$ denotes the unnormalized density (For comparison, $\widehat{q}_y(x,t) = \frac{1}{\widehat{Z}_y(t)}\widehat{Q}_y(x,t)$ denotes the normalized density). Multiplying $e^{\mu}$ on both sides of the identity above then yields $\widehat{\overleftarrow{p_t}}(x) = \widehat{Q}_y(x,t)e^{\mu}$. If the reviewer has any follow-up question about this identity and the derivation, we are happy to answer it at any time.
>
>
> (4) Additional comments related to writing clarity and typos
>
> We sincerely thank the reviewer for reading the manuscript carefully and providing a comprehensive list of typos and writing issues, which has greatly helped improve the quality of our manuscript. Below we address these points one by one.
>
> (i) Regarding the abstract, we thank the reviewer for pointing out the missing text in the second-to-last sentence. We apologize for this omission and will correct it in the updated manuscript as follows:
>
> "Theoretically, we prove that the error between the true posterior and the empirical distribution of the generated samples can be bounded in terms of the..."
>
> (ii) For the symbols, notations, and abbreviations used throughout this paper, we will double check and ensure that all of them all properly defined. In particular, we will clarify that the ILVR abbreviation refers to "Iterative Latent Variable Refinement" in the introduction. Moreover, for the first sentence of subsection 2.2, we will follow the reviewer's advice by adding quotation marks around the EDM framework. The sentence will now read "We adopt the 'Elucidating the design space of Diffusion Models' (EDM) framework from...". Furthermore, regarding the definition of $\beta_t$ and $\gamma_t$, we agree with the reviewer that they denote the particle weight and joint probability distribution of $(x_t,\beta_t)$ (i.e., position and weight of the particle), respectively. In the revised manuscript, we will explicitly state the definition of $\beta_t$ and $\gamma_t$ at the beginning of subsection 3.2.

---

> ### Author Response · Authors · 2026-01-06
> **Rebuttal (Part 3)**
>
> (5) Intuitive explanation of Assumption 4.4 and significance of the main theoretical results
>
> We would like to thank the reviewer for pointing out that both the theoretical results and Assumption 4.4 need further discussion. Regarding Assumption 4.4, we will revise the manuscript to clearly explain, for the special case when $s(t)=1$ and $\sigma(t) =t$, how the function $I(x,t)$ defined in Assumption 4.4 relates to the function $U(x,t) - H(x,t)^T\nabla_{x}\mu_{y}(x)$ in equation (3.5), which governs the evolution of the particle weights. Specifically, a direct computation yields that
>
> $$U(x,t) - H(x,t)^T\nabla\mu(x) = \frac{2t}{2}(||\nabla\mu(x)||_2^2 - \Delta \mu(x)) - \frac{4t}{2}\phi(x,t)^T\nabla\mu(x) = tI(x,t).$$
>
> We will ensure that the above computation is included and properly discussed in the revised manuscript. Moreover, we will expand the discussion on how restrictive Assumption 4.4 is in the main text by relating it to Assumption 1 in [2]. Essentially speaking, both assumptions serve to control the particle weights during evolution and thereby mitigate weight degeneracy. In practice, however, we note that the particle weights are often controlled via the resampling step (Algorithm 1) and the ESS threshold. We remark that it would be interesting to investigate whether Assumption 4.4 can be relaxed in the theoretical analysis and link to the resampling step. This is a promising direction for future work but beyond the scope of the current paper. In the revised manuscript, we will make sure to explicitly note this point in the main text.
>
> Regarding the significance of our main theoretical results (Theorems 4.1 and 4.2), we note that our analysis provides a principled continuous-time framework for bounding the distributional discrepancy between the true posterior and the distribution induced by the generated samples. In particular, the bound decomposes into two components: Theorem 4.1 quantifies the contribution from score-matching error, while Theorem 4.2 captures the approximation error induced by using a finite number of particles. More broadly, these results also open the door to future work on consistency and stability with respect to time discretization (step size). To the best of our knowledge, pursuing this direction will likely require more advanced tools from the numerical analysis of (stochastic) weighted particle/SMC methods, as also noted at the end of Section 6. In the revised main text, we will incorporate the above discussion at the end of Section 4.
>
> (6) Questions related to the experimental settings (test size, uncertainty measures, etc.)
>
> Thanks for bringing up questions related to the experimental settings. We would like to kindly note that we use the same collection of 100 images from the FFHQ-256 and ImageNet-256 validation sets to match the experimental settings adopted in current state-of-the-art (SOTA) methods, such as the SGS-DM (Split Gibbs Sampler + Diffusion Model) method [1] and DAPS (Decoupled Annealing Posterior Sampling) method [3]. Specifically, the reviewer may refer to the first sentence in the "Dataset and inverse problems" part of subsection 4.2 in [1] or the second sentence in the "Datasets and metrics" part of subsection 4.1 in [3].
>
> Moreover, to the best of the authors' knowledge, most existing works that develop methods for solving imaging inverse problems based on priors parameterized by pretrained diffusion models report their results on the validation datasets (see, e.g., [1,3,4,5,6]). This is because these approaches are essentially inference-time samplers built on the same fixed pretrained model (i.e., they do not perform model selection or hyperparameter tuning on the pretrained model), which ensures a fair comparison. Therefore, we think an additional held-out test set is probably not strictly required for a fair comparison with existing SOTA methods under the standard evaluation protocol. Would it be possible for the reviewer to clarify the specific concern that motivates the need for an independent test set rather than a validation set?
>
> However, we agree with the reviewer that it is important to evaluate the proposed methods on a larger set of images from the validation set and report the uncertainty measures (e.g., error bars). We are currently conducting two additional sets of experiments and we will incorporate their results into the revised manuscript:
>
> (i) Compute and report uncertainty measures (e.g., standard deviation) for AFDPS-SDE and AFDPS-ODE in the current experiments on 100 images (Tables 1 and 2).
>
> (ii) Evaluate AFDPS-SDE and AFDPS-ODE on a larger collection of 1,000 images (same as that of [4,5,6]) from the validation set.

---

> ### Author Response · Authors · 2026-01-07
> **Rebuttal (Part 4)**
>
> (7) Comparison between our method and existing work on combining SMC with diffusion-based priors
>
> We are grateful to the reviewer for drawing our attention to this point. The key novelty of our method compared to existing works that combine SMC with diffusion-based priors can be summarized from the following two aspects:
>
> The first aspect is our derivation from a continuous-time perspective, which yields a PDE-based formulation of the algorithm. In contrast, existing approaches such as [6,7,8] are typically defined via proposal kernels based on the discrete-time formulation. Our PDE-based viewpoint may facilitate future work on designing more efficient numerical schemes and extending the framework to incorporating with more complicated diffusion processes for the prior evolution (like continuous-time markov chain or levy processes).
>
> The second aspect is about the choice of the likelihood function, which defines the distribution paths used in practice. Below, we expand on the distribution paths used in several representative examples [6,8,9] from existing work that combine SMC with diffusion-based priors for inverse problems, which have deployed two different kinds of likelihood functions. Specifically, [6,9] adopted the setting of particle filtering by generating a corresponding noisy observation $y_t$ based on the observed $y$ for any $x_t$ (The reviewer may also refer to Part 2 of the Rebuttal associated with Reviewer XoVc above to see how the noisy sequence is generated), which in turn leads to the following distribution path:
>
> $$l_t(x) \propto p_t(x) e^{-\mu_{y_t}(x)}$$
>
> The other work [7], on the other hand, applies Tweedie's formula to approximate the conditional score and yields a sequence of evolving posterior distribution of the form below:
>
> $$l_t(x) \propto p_t(x)p(y|x_t=x) = p_t(x)(\int p(y|x_0=z)p(x_0=z|x_t=x)dz) \propto p_t(x)(\int p(x_0=z|x_t=x)e^{-\frac{1}{2\sigma^2}||y-A(z)||^2}dz)$$
>
> From the formulas above we can see that the two kinds of distribution paths deployed in [6,7,8] are all different from the one $q_t(x) \propto p_t(x)e^{-\mu_{y}(x)}$ used in our work. We are currently revising the manuscript to include a more detailed version of the discussion above as an extra remark at the end of subsection 3.2, which highlights the difference between our method and existing SMC-based works.
>
> (8) Question about the integration of diffusion-based priors and the SMC framework
>
> Thanks for raising this question. We would like to point out that the SMC-framework follows from the PDE derivation presented in subsection 3.1. Specifically, our derivation yields the linear term $(\cdots)\widehat{q}_y$ in the PDE (3.4) by normalizing the PDE (3.3) (Detailed steps are provided in Lemma B.2 of Appendix B), which needs to simulated via the SMC/stochastic weighted particle method. In the revised manuscript, we will add a few sentences at the end of subsection 3.1 to explain how the linear term in (3.4) motivates the use of an SMC-style method, which also provides a smoother transition into subsection 3.2.
>
> (9) Comments on the inner-workings of the proposed algorithms
>
> We appreicate the reviewer's questions regarding the inner-workings of the proposed algorithms. Below we will address the two main questions one by one.
>
> (i) For a brief discussion on the pros and cons of the two variants (AFDPS-SDE and AFDPS-ODE), we first note that the “Summary” part of Appendix E already includes a short comparison between them in terms of reconstruction quality. However, we agree with the reviewer that a more detailed comparison should also be included in the main text, and we will add it to Subsection 3.2 (before Remark 3.1). Below we provide a brief sketch of such discussion on the comparison between the two methods:
>
> "Based on our experience, the ODE variant typically requires tuning of the hyperparameters associated with the corrector (e.g., step size and number of corrector steps), whereas the SDE variant does not. On the other hand, the ODE formulation is often more amenable to the design of higher-order numerical solvers with improved accuracy and faster sampling speed. More broadly, the relative advantages of SDE versus ODE formulations remain an active research topic and the readers may refer to [9] for related studies."
>
> In addition, the reviewer is correct that it would be valuable to conduct a case study on the sensitivity of the AFDPS-ODE method to the hyperparameters of the MALA/ULA-based corrector. We are currently running such an extra experiment and will incorporate the results into the revised manuscript.

---

> ### Author Response · Authors · 2026-01-07
> **Rebuttal (Part 5)**
>
> (ii) Regarding the ESS thresholds in Algorithm 1 (which is used in Algorithms 3 and 4), we would like to first clarify that in practice, the particle with the largest weight in the ensemble is returned as the final estimator of the recovered image. This is closely analogous to the best-of-$N$ strategy (i.e., selecting the best-performing sample among candidates). Therefore, the resampling step in Algorithm 1 is omitted in our practical implementations of Algorithms 3 and 4, which allows us to save computational cost by updating particle positions and weights in parallel. This is also noted in the last three sentences of the first paragraph in Appendix D.2. Hence, the current manuscript doesn't contain any specific value of the ESS threshold.
>
> However, we agree with the reviewer that it would be meaningful to conduct additional experiments to see how the ESS threshold should be chosen and ESS value associated with the ensemble of particles evolve throughout the sampling process, just as what have been plotted in Figures 9, 11, 16 and 17 in [10]. We are currently carrying out these additional experiments and will include the results in the revised manuscript once they become available.
>
> Overall, we sincerely thank the reviewer for the patience and all the insightful feedback. We hope that the clarifications and planned revisions listed above can address the reviewer’s concerns, and we would be grateful if the reviewer could consider reevaluating our work based on our response. We will let all the reviewers know once we finished finalizing the revised manuscript by adding all the required experiments. In case the reviewer has any follow-up questions, we are more than happy to address them.
>
> ### References
>
> [1] Wu, Zihui, et al. "Principled probabilistic imaging using diffusion models as plug-and-play priors." Advances in Neural Information Processing Systems 37 (2024): 118389-118427.
>
> [2] Domingo-Enrich, Carles, et al. "A mean-field analysis of two-player zero-sum games." Advances in neural information processing systems 33 (2020): 20215-20226.
>
> [3] Zhang, Bingliang, et al. "Improving diffusion inverse problem solving with decoupled noise annealing." Proceedings of the Computer Vision and Pattern Recognition Conference. 2025.
>
> [4] Xu, Xingyu, and Yuejie Chi. "Provably robust score-based diffusion posterior sampling for plug-and-play image reconstruction." Advances in Neural Information Processing Systems 37 (2024): 36148-36184.
>
> [5] Chung, Hyungjin, et al. "Diffusion Posterior Sampling for General Noisy Inverse Problems." The Eleventh International Conference on Learning Representations.
>
> [6] Dou, Zehao, and Yang Song. "Diffusion posterior sampling for linear inverse problem solving: A filtering perspective." The Twelfth International Conference on Learning Representations. 2024.
>
> [7] Wu, Luhuan, et al. "Practical and asymptotically exact conditional sampling in diffusion models." Advances in Neural Information Processing Systems 36 (2023): 31372-31403.
>
> [8] Cardoso, Gabriel, et al. "Monte Carlo guided diffusion for Bayesian linear inverse problems." The Twelfth International Conference on Learning Representations (2024).
>
> [9] Cao, Yu, et al. "Exploring the optimal choice for generative processes in diffusion models: Ordinary vs stochastic differential equations." Advances in Neural Information Processing Systems 36 (2023): 33420-33468.
>
> [10] Ren, Yinuo, et al. "Driftlite: Lightweight drift control for inference-time scaling of diffusion models." arXiv preprint arXiv:2509.21655 (2025).

---

### Review · Reviewer_XoVc · 2025-11-29

**Summary Of Contributions:**

The authors propose a particle filtering approach for posterior sampling with a diffusion-model prior. Their contributions include deriving the PDE of the posterior evolution given the diffusion-model-based prior evolution. They also provide theoretical error bounds and convergence guarantees. They provide numerical experiments showing improved PSNR and LPIPS over five baselines evaluated on four inverse problems and two image datasets.

Strengths:
* The proposed algorithm is sound and provides an alternative view of posterior sampling with diffusion-model priors.
* The proposed algorithm is free of any heuristics approximating the posterior. It appears that given enough particles, a good-enough time discretization, and a good-enough score approximation, one can get the true posterior.

Weaknesses:
* I find the experimental results really lacking. The theoretical analysis and empirical validation are currently disjoint. The experiments should include numerical validation of convergence to the true posterior, since being approximation-free is the main advantage of the approach.
* In the same vein as the above point, I don’t find PSNR/LPIPS between the posterior mean and the ground-truth image to be a suitable metric. The whole point of the paper is to do accurate posterior sampling. Ideally, there would be a metric showing distributional similarity between the estimated and true posteriors (maybe using the maximum mean discrepancy or kernel Inception distance metrics, for example). At the very least, the authors should include PSNR/LPIPS evaluated on posterior samples rather than the posterior mean.
* The experiments only include linear inverse problems (as far as I understand, the authors consider linear deblurring problems). It would be nice to show that the method works on nonlinear inverse problems, too.
* The baselines don’t exactly exhaust current state-of-the-art methods. For example, it would be nice to also compare to DAPS [1]
* There could be more intuitive explanations throughout the paper. For example, what are intuitive explanations of the (ILVR) and (DPS) approximations? More critically, what’s the intuitive explanation of the PDE derived in this paper (3.4)? I’m personally wondering why you get a different PDE with your approach than if you were to replace the prior score with the posterior score in the reverse SDE formulation of diffusion models.

[1] Bingliang Zhang, Wenda Chu, Julius Berner, Chenlin Meng, Anima Anandkumar, and Yang Song. “Improving diffusion inverse problem solving with decoupled noise annealing.” CVPR 2025.

**Additional Comments:**

Since I don’t have a background in numerical analysis, I couldn’t critically check the derivations and proofs.

**Audience:**

Yes

**Audience Explanation:**

This work is definitely relevant to people working on diffusion models for inverse problems or more generally in generative modeling, inverse problems, and imaging.

**Claims And Evidence:**

No

**Claims Explanation:**

Theoretically, the approach and justification seem solid (although I don’t have background in numerical analysis so I couldn’t check the proofs in detail). However, there is little to no empirical evidence provided that the method converges to the true posterior. If the authors instead want to emphasize superior reconstruction performance, then this claim should be made clearer and earlier on in the paper.

**Requested Changes:**

I would require the following changes to recommend acceptance:
* Some kind of empirical validation of convergence to the true posterior
* Evaluation of PSNR/LPIPS on posterior samples in addition the posterior mean
* Additional intuitive explanation (even if just a few sentences) of Equation (3.4)

Some minor critiques that are not critical but would improve the manuscript:
* The first sentence in the second paragraph (beginning with “To overcome these limitations”) doesn’t logically follow from the previous sentence. The previous paragraph brings up how MCMC can’t model high-dimensional posteriors. But then this sentence talks about priors. I don’t see how “deep generative models have been proposed for encoding prior distributions” solves the problem of high-dimensional posterior sampling.
* “…existing work include but not limit to ILVR…” --> “…existing work include but are not limited to ILVR…”
* Add intuitive explanations of the ILVR and DPS approximations.
* The phrase “more flexible PDE dynamics” (at the bottom of page 2) is confusing to me. What does it mean for PDE dynamics to be flexible? Also, more flexible compared to what?
* The explanation of Theorem 4.1 (“a trade-off controlled by the time horizon T between the prior mismatch and score approximation error”) is difficult to understand as it relates to the actual bound.
* In Figure 2, the middle-row images don’t appear to be blurred even though the caption says they are. It almost looks like a denoising setting to me.
* In the discussion and conclusion section, the authors claim that the framework could be extended to lots of other variants of DMs, including latent DMs and discrete DMs. These are some pretty broad claims. If this is the case, then the authors should provide a brief explanation of why they believe this to be the case.

---

> ### Author Response · Authors · 2025-12-21
> **Rebuttal (Part 1)**
>
> We would like to thank the reviewer for the thorough and insightful review. Below we respond to each of the reviewer’s comments in sequence.
>
> (1) Numerical validation of convergence towards the true posterior and evaluation of the PSNR/LPIPS metrics
>
> We are grateful to the reviewer for drawing our attention to this point. However, we respectfully disagree with the reviewer’s assessment that the experimental results are "really lacking". On the one hand, in line with existing posterior-sampling methods [1,3,4,5,6,7] for solving high-dimensional imaging inverse problems on real-world datasets (e.g., FFHQ256 and ImageNet256), our work primarily emphasizes superior reconstruction performance with respect to standard metrics like PSNR/LPIPS/SSIM, which is also most relevant for practical applications. We will follow the reviewer’s suggestion by revising the main text to make our emphasis on reconstruction performance explicit and to introduce it earlier in the manuscript.
>
> On the other hand, we would like to kindly note that for high-dimensional inverse problems on $256 \times 256$ images from real-world datasets such as FFHQ and ImageNet, the posterior distribution is generally not available in an analytically tractable form. Recently, there has already been some work evaluating our method on analytically tractable target distributions via distributional metrics like maximum mean discrepancy (MMD). For instance, one may refer to the Guidance-SMC (G-SMC) method listed in [2], which has tested our method on both Gaussian mixture models and two Boltzmann distributions (Double-Well-4 and Lennard-Jones-13) with respect to the MMD metric. To provide a more rigorous justification of our method in terms of accurate posterior sampling, we will follow the reviewer's advice by evaluating how well our method converges to the ground-truth posterior in a synthetic compressed sensing problem with a Gaussian prior. This setting, which admits an analytically tractable posterior, is highly similar to the experiments covered in section 4.1 of [7]. We will include the experimental results in the rebuttal once they are available and update the manuscript as well.
>
> Furthermore, we would like to clarify that our paper didn't use "PSNR/LPIPS between the posterior mean and the ground-truth image" in our evaluation. Specifically, in each generated ensemble, the particle with the largest weight is returned as the final estimator of the recovered image. This is closely analogous to the best-of-$N$ strategy (i.e., selecting the best-performing sample among candidates). Such selection rule is also stated in the second-to-last sentence of the first paragraph in Appendix D.2 and applies to all images shown in Figure 2 and the Appendix. To avoid any confusion, we will ensure that the main text is revised to make this selection criterion explicit.
>
>
> (2) Additional experiments on non-linear inverse problems and comparisons to state-of-the-art (SOTA) baselines
>
> We thank the reviewer for this comment. We are currently conducting additional experiments on two nonlinear inverse problems — Fourier phase retrieval (FPR) and coded diffraction patterns (CDP) — using the FFHQ-256 grayscale dataset under the same experimental settings as in [2]. Regarding additional SOTA baselines, we are currently evaluating DAPS [1] on the four inverse problems considered in the manuscript. We will report the results in the rebuttal upon completion and revise the paper accordingly.

---

> > ### Comment · Reviewer_XoVc · 2025-12-26
> >
> > Thank you to the authors for their response. I appreciate their efforts to include (1) an experiment with a ground-truth posterior based on compressed sensing with a Gaussian prior and (2) experiments with nonlinear inverse problems.
> >
> > I still have the question about an intuitive explanation of Equation (3.4) and why you get a different SDE than if you were to replace the prior score with the posterior score in the reverse SDE formulation of diffusion models.
> >
> > I'm happy to recommend acceptance if the proposed additions/revisions make it into the manuscript.

---

> ### Author Response · Authors · 2025-12-28
> **Rebuttal (Part 2)**
>
> (3) Intuitive explanations of the approximations adopted in existing work (DPS [3] and ILVR [8])
>
> Thanks for raising this question. We also noticed a few typos in our presentation of the ILVR and DPS approximations, and we will correct them in the revised manuscript. Below we first explain the main ideas behind the DPS [3] and ILVR [8] approximations to the posterior score function for the setting of linear inverse problems with Gaussian noise. We begin by introducing a few notations. Let $y=y_0 = Ax_0+n$ denote the forward model, where $x_0 \in \mathbb{R}^n, y=y_0 \in \mathbb{R}^m$ and $n \in \mathbb{R}^m$ denote the input, measurement, and Gaussian observational noise respectively, with $n \sim \mathcal{N}(0,\sigma^2I_m)$. Let $x_t = x_0 + \gamma(t)w$ denote the forward diffusion process with injected noise $w \sim \mathcal{N}(0,I_n)$. The corresponding forward noising noising process for the measurements is further denoted by $Y_t = y_0 + \gamma(t)\eta$, where $\eta = Aw$ is a transformed multivariate Gaussian distribution in $\mathbb{R}^m$. Below we use $y_t$ to denote a realization (sample) of the random variable $Y_t$.
>
> On the one hand, we remark that the ILVR approximation can essentially be understood as a preconditioned version of the posterior score approximation derived in the Score-SDE paper [10], which we will derive and explain first. A direct application of Bayes' formula gives us that
> $$p(y|x_t) = p(y_0|x_t) = \frac{p(y_0,x_t)}{p(x_t)} = \frac{p(x_t|y_0)p(y_0)}{p(x_t)} \Rightarrow \nabla_{x_t}\log p(y|x_t) = \nabla_{x_t}\log p(x_t|y_0) - \nabla_{x_t}\log p(x_t)$$
> Then we have that it suffices to derive an approximation of the term $\nabla_{x_t}\log p(x_t|y_0)$. Following the derivation in Appendix I.4 of [10], we have
> $$p(x_t|y_0) = \int p(x_t|Y_t,y_0)p(Y_t|y_0)dY_t \approx \int p(x_t|Y_t)p(Y_t|y_0)dY_t \approx p(x_t|y_t)$$
> Specifically, we note that the last approximation above results from the property that $y_t$ is a realization of $Y_t$ (i.e., sampled from $p(Y_t|y_0)$). Moreover, the first approximation above follows from the following facts:
>
> (i) When $t$ is small, $Y_t$ is almost the same as $y_0$, so we have $p(x_t | Y_t,y_0) \approx p(x_t|Y_t)$;
>
> (ii) When $t$ is large, we have that $x_t$ is close to pure Gaussian noise, which is away from $y_0$ and again implies $p(x_t | Y_t,y_0) \approx p(x_t|Y_t)$.
>
> Then we can plug in the approximation $p(x_t|y_0) \approx p(x_t|y_t)$ above and apply Bayes' formula again, which indicate that
> $$\nabla_{x_t}\log p(y|x_t) \approx \nabla_{x_t}\log p(x_t|y_t) - \nabla_{x_t}\log p(x_t) = \nabla_{x_t}\log \frac{p(x_t|y_t)}{p(x_t)} = \nabla_{x_t}\log p(y_t|x_t)$$
> Furthermore, from our definition of $Y_t$ above we have that
> $$Y_t - Ax_t = y_0 - Ax_0 = n \sim \mathcal{N}(0,\sigma^2I_m) \Rightarrow p(y_t|x_t) \propto e^{-\frac{1}{2\sigma^2}||y_t - Ax_t||^2}$$
>
> Then we can further deduce that
> $$\nabla_{x_t}\log p(y|x_t) \approx \nabla_{x_t}\log p(y_t|x_t) = -\frac{1}{2\sigma^2}\nabla_{x_t}||y_t-Ax_t||^2 = \frac{1}{\sigma^2}A^T(y_t-Ax_t)$$
> which is the approximation adopted in the Score-SDE [10] paper. The ILVR approximation can then be interpreted as a preconditioned version ($\frac{1}{\sigma^2}(A^TA)^{-1}A^T(y_t-Ax_t)$) of the Score-SDE approximation derived above, where $(A^TA)^{-1}$ is the preconditioner.
>
> On the other hand, the DPS approximation is mainly based on the factorization below:
>
> (Specifically, we note that the second equation below follows from Bayes' formula, the third equation below is derived based on the fact that $y$ is conditionally independent of $x_t$ given $x_0$, and the last equation below follows from approximating the expectation of a function via its evaluation at the expectation, i.e., $\mathbb{E}[f(Z)] \approx f(\mathbb{E}[Z])$):
>
> $$p(y|x_t) = \int p(y, x_0|x_t)dx_0  = \int p(y|x_0,x_t)p(x_0|x_t)dx_0 = \int p(y|x_0)p(x_0|x_t)dx_0 = \mathbb{E}_{z \sim p(x_0|x_t)}[p(y|z)] \approx  p(y | X_0(t))$$
>
> where $X_0(t) = \mathbb{E}_{z \sim p(x_0|x_t)}[z] = \mathbb{E}[x_0|x_t]$ above denotes the conditional expectation of $x_0$ given $x_t$. A direct application of Tweedie's formula yields that
>
> $$X_0(t) = \mathbb{E}[x_0|x_t] = x_t + \gamma(t)^2\nabla_{x_t}\log p_t(x_t)$$
>
> Then we can apply the operator $\nabla_{x_t}\log(\cdot)$ on both sides of the approximation $p(y|x_t) \approx  p(y | X_0(t))$ above to deduce that
>
> $$\nabla_{x_t}\log p(y|x_t) \approx \nabla_{x_t}\log(p(y|X_0(t))) = -\frac{1}{2\sigma^2}\nabla_{x_t}||y - AX_0(t)||^2 = \frac{1}{\sigma^2}(\nabla_{x_t}(AX_0(t)))^T(y-AX_0(t)) = \frac{1}{\sigma^2}(\nabla_{x_t}X_0(t))^TA^T(y-AX_0(t))$$
>
> Plugging in the expression of $X_0(t)$ above then gives us the DPS approximation of the posterior score function:
>
> $$\nabla_{x_t}\log p(y|x_t) \approx \frac{1}{\sigma^2}(\nabla_{x_t}X_0(t))^TA^T(y-AX_0(t)) = \frac{1}{\sigma^2}(I_n + \gamma(t)^2\nabla^2_{x_t}\log p_t(x_t))^TA^T(y-A\mathbb{E}[x_0|x_t])$$

---

> ### Author Response · Authors · 2025-12-28
> **Rebuttal (Part 3)**
>
> For a complete derivation of the ILVR and DPS approximations described above, as well as a list of all related approximations, we refer the readers to Figure 1 and Subsection 3.1 of [9]. Following the reviewer’s suggestion, we will (i) correct current typos (e.g., sign errors, missing $\sigma$ and $\gamma(t)$) and (ii) add a clear explanation of the motivation behind the approximations used in prior work, including Score-SDE [10], DPS [3] and ILVR [8]. However, due to space constraints on the main text, we will move the detailed derivations and discussion to a new appendix subsection, and we will explicitly point readers to both the newly added subsection and [9] in the main text of the revised manuscript.
>
> (4) Question regarding the transition between the first and second paragraphs of the Introduction
>
> We appreciate the reviewer for noting a potential issue in the logical flow of this transition. To address this, we will first add a more detailed explanation of the relationship between the prior and posterior. Specifically, we will revise the last sentence of the first paragraph as follows:
>
> "However, in many practical inverse problems, the prior distribution is already high-dimensional and may contain multiple well-separated modes. Coupled with an ill-posed forward model and noisy observations, such complex priors often induce posteriors that are likewise high-dimensional and strongly multimodal. Consequently, traditional Markov Chain Monte Carlo (MCMC) methods often struggle with sampling from these posterior distributions primarily due to metastability, i.e., the difficulty in transitioning between distinct high-probability modes that are separated by regions of low probability."
>
> Then, we will remove the phrase “for encoding prior distributions” at the beginning of the second paragraph, since the cited works based on normalizing flows/GANs do not merely use generative models to represent the prior. Instead, they incorporate problem structure with generative models to characterize the posterior in a more synergistic manner. In a conrete way, we will revise the first sentence of the second paragraph as follows:
>
> "To address these limitations, prior work has leveraged generative models like normalizing flows (NFs) and generative adversarial networks (GANs) to model and sample from those high-dimensional and multimodal posterior distributions."
>
> In the revised manuscript, we will modify these two sentences (the last sentence of the first paragraph and the first sentence of the second paragraph) to ensure a smooth logical transition in the introduction.
>
> (5) Question about the explanation of Theorem 4.1
>
> Thank you so much for pointing out the ambiguity in our explanation of Theorem 4.1. Below is a revised version of the sentence starting with "The result of Theorem 4.1 reveals a trade-off...". which we will incorporate into the revised manuscript:
>
> "The result of Theorem 4.1 reveals that under assumptions 4.1-4.3, the discrepancy between the posterior distributions can be upper-bounded by a term proportional to the discrepancy between the prior distributions. The upper bound further captures a tradeoff between the error of the forward process and the score matching error, which is controlled by the time horizon $T$."
>
> (6) Miscellaneous questions related to writing and typos
>
> We thank the reviewer for proofreading the manuscript carefully and providing a comprehensive list of typos, writing and clarity issues. Below we respond to the remaining points that were not addressed above.
>
> Specifically, regarding the grammatical issue with the phrase “but not limit to”, we agree that it should be replaced with “but are not limited to”. For the middle-row images in Figure 2, we agree that the term "blurred" is not quite accurate. We will replace it with "noisy (measurement)" in the revised manuscript.
>
> Furthermore, we thank the reviewer for noting that our statement, “the framework could be extended to many other variants of diffusion models, including latent DMs and discrete DMs,” is a bit too broad and needs further clarification. We apologize that these extensions are still ongoing work and therefore probably cannot be presented in full detail. Below we provide a brief sketch of the main ideas behind these extensions.

---

> ### Author Response · Authors · 2025-12-29
> **Rebuttal (Part 4)**
>
> Essentially speaking, we consider some high-dimensional prior distribution $\overleftarrow{p_T}$ parameterized by either a discrete diffusion model or a latent diffusion model, whose associated PDE dynamics $\mathcal{P}$ (distinct from those under the EDM framework of continuous diffusion models) transforms a tractable reference distribution $\overleftarrow{p_0}$ to $\overleftarrow{p_T}$. An analogous task of interest is to sample from the posterior/tilted distrbution $q_{T} \propto \overleftarrow{p_T}e^{\mu}$, where $\mu$ can be either a log-likelihood or a reward function. Under the framework developed in our paper, we can again define $q_t \propto \overleftarrow{p_t} e^{\mu}$ to be the evolving posterior for $t \in [0,T]$ and obtain a SMC-type method for sampling from $q_{T}$ by following the two steps below:
>
> (I) Derive the PDE dynamics $\mathcal{Q}$ governing the evolution of $(q_t)_{t \in [0,T]}$ based on $\mathcal{P}$, which governs the evolution of $p_t$ for $t \in [0,T]$;
>
> (II) Solve $\mathcal{Q}$ numerically via weighted particle method.
>
> In the revision, we will temper the claim and incorporate a brief discussion, which outlines the key intuition for adapting the framework to latent and discrete diffusion models, into the “Discussion and Conclusion” section of the revised manuscript.
>
> (7) Questions related to our PDE-based derivation (3.4)
>
> We appreciate the reviewer's question about the derived PDE (3.4), which is highly relevant to the main contribution of this paper. Below, we provide an intuitive explanation of how the PDE in (3.4) is derived (Interpretation of each term will also be included), using notations similar to that used in the manuscript. Specifically, we define $(\overleftarrow{p_t})_{t \in [0,T]}$ to be the time-evolving prior distribution parameterized by a pretrained diffusion model. Let $\mu$ and $Q_t := \overleftarrow{p_t}e^{\mu}$ denote the likelihood function and time-dependent unnormalized posterior distribution. We further use $Z_t:= \int \overleftarrow{p_t}(x)e^{\mu(x)}dx$ and $q_t:= \frac{Q_t}{Z_t}$ to represent the time-evolving normalizing constant and normalized posterior distribution, respectively. Just as what we outlined in subsection 3.1 and Figure 1 of the manuscript, our derivation of the PDE (3.4) can be divided into the following steps:
>
> (I) Write out the Fokker-Planck PDE that governs the evolution of $(\overleftarrow{p_t})_{t \in [0,T]}$, which is provided standard based on existing work like Score-SDE [10].
>
> (II) Substitute $\overleftarrow{p_t} = Q_te^{-\mu}$ into the PDE in (I) above and rearrange the terms, which yields a PDE governing the evolution of $(Q_t)_{t \in [0,T]}$.
>
> (III) Normalize the PDE in (II) above by plugging in $q_t = \frac{1}{Z_t}Q_t$, which yields the desired PDE (3.4) that governs the evolution of $(q_t)_{t \in [0,T]}$
>
> A detailed computation following the steps (I), (II) and (III) above then yields the PDE (3.4), which we summarized below:
>
> $$\frac{\partial}{\partial t}q_t = -\nabla_{x} \cdot ((H(x,t) - V(t)^2\nabla_{x}\mu)q_t) + \frac{1}{2}V(t)^2\Delta_{x}q_t + ((U(x,t) - H(x,t)^T\nabla_{x}\mu) - (\int(U(x,t) - H(x,t)^T\nabla_{x}\mu)q_t(x)dx))q_t$$
>
> We note that the PDE above consists of three main components: the first term corresponds to the drift, the second to the diffusion (Laplacian), and the third to the reweighting term. On the one hand, the term $H(x,t)$ above is exactly the original drift used in the Fokker Planck PDE that $p_t$ satisfies for $t \in [0,T]$. On the other hand, both the additional term $V(t)^2\nabla_{x}\mu$ in the drift and the function $U(x,t)$ in the reweighting term are dependent on $V(t)$ and $\mu$ only from our derivation, which originates from the term $e^{\mu}$ in $q$. In fact, a key novelty of our method is that the derived PDE (3.4) includes an explicit reweighting term, which needs to simulated via weighted particle method/SMC. This requires the sampling/inference algorithm to keep track of an ensemble of samples, each of which has an associated weight. Intuitively, particles with larger weights are more likely to lie in high-probability regions of the posterior and thus tend to correspond to higher-quality reconstructions. In fact, such procedure can also be interpreted through the lens of debiasing: maintaining an ensemble of weighted particles helps reduce simulation bias.

---

> ### Author Response · Authors · 2025-12-29
> **Rebuttal (Part 5)**
>
> The reweighting term also distinguishes our method (as well as the corresponding PDEs) from existing approaches that directly replace the prior score with an (approximate) posterior score. Essentially speaking, these approaches still track **only one particle at a time**. Hence, the Fokker Planck PDEs associated with the time-evolving distributions tracked by these approaches **contain only a drift term (which directly relates to the approximate posterior score) and a diffusion (laplacian) term**. As there is **no reweighting term involved**, these methods **all admit a single-particle SDE interpretation (drift and diffusion only)**. Derivations of both the Fokker Planck PDEs and the single-particle SDEs for these methods are exactly the same as standard diffusion models, where the only slight difference is to replace the prior score with the (approximate) posterior score. Mathematically, we can follow similar steps as in (3) of Rebuttal (Part 2) above to explicitly derive the time-evolving density tracked by methods relying on posterior scores as below:
>
> $$l_t(x) \propto p_t(x)p(y|x_t=x) = p_t(x)(\int p(y|x_0=z)p(x_0=z|x_t=x)dz) \propto p_t(x)(\int p(x_0=z|x_t=x)e^{-\frac{1}{2\sigma^2}||y-A(z)||^2}dz)$$
>
> Comparing $q_t$ with $l_t$ yields that the tilting term in the product (i.e., the term that $p_t$ is multiplied by) $e^{\mu} = e^{-\frac{1}{2\sigma^2}||y-A(x)||^2}$ and $(\int p(x_0=z|x_t=x)e^{-\frac{1}{2\sigma^2}||y-A(z)||^2}dz)$ are different for these two distribution paths, so they ultimately lead to different PDEs. Moreover, practical methods also need to seek different approximations of the term $(\int p(x_0=z|x_t=x)e^{-\frac{1}{2\sigma^2}||y-A(z)||^2}dz)$, just as we described above in (3) of Rebuttal (Part 2). This is in fact one of the main contributions of our work: we derive an alternative PDE dynamics, which still converge to the target posterior and can be simulated without approximating the posterior score (Hence our method is "approximation-free"), based on the pretrained prior score.
>
> In addition, we thank the reviewer for pointing out that the phrase “more flexible PDE dynamics” at the bottom of page 2 might be a bit confusing. We will revise the manuscript by removing “more flexible” and retaining the other descriptors (“rigorously derived” and “previously unexplored”) only. We will make sure to revise the subsection 3.1 in the manuscript to include both an intuitive explanation of each term in PDE (3.4) and an emphasis of our main contribution ("approximation-free") compared to existing work that relies on approximations of posterior scores.
>
> Overall, we thank the reviewer again for the patience and all the insightful feedback. We will incorporate all revisions into the updated manuscript, and we will update both the rebuttal and the paper once the additional experiments and final edits are completed.
>
> ### References
>
> [1] Zhang, Bingliang, et al. "Improving diffusion inverse problem solving with decoupled noise annealing." Proceedings of the Computer Vision and Pattern Recognition Conference. 2025.
>
> [2] Ren, Yinuo, et al. "Driftlite: Lightweight drift control for inference-time scaling of diffusion models." arXiv preprint arXiv:2509.21655 (2025).
>
> [3] Chung, Hyungjin, et al. "Diffusion Posterior Sampling for General Noisy Inverse Problems." The Eleventh International Conference on Learning Representations.
>
> [4] Dou, Zehao, and Yang Song. "Diffusion posterior sampling for linear inverse problem solving: A filtering perspective." The Twelfth International Conference on Learning Representations. 2024.
>
> [5] Song, Yang, et al. "Solving Inverse Problems in Medical Imaging with Score-Based Generative Models." International Conference on Learning Representations.
>
> [6] Xu, Xingyu, and Yuejie Chi. "Provably robust score-based diffusion posterior sampling for plug-and-play image reconstruction." Advances in Neural Information Processing Systems 37 (2024): 36148-36184.
>
> [7] Wu, Zihui, et al. "Principled probabilistic imaging using diffusion models as plug-and-play priors." Advances in Neural Information Processing Systems 37 (2024): 118389-118427.
>
> [8] Choi, Jooyoung, et al. "Ilvr: Conditioning method for denoising diffusion probabilistic models." arXiv preprint arXiv:2108.02938 (2021).
>
> [9] Daras, Giannis, et al. "A survey on diffusion models for inverse problems." arXiv preprint arXiv:2410.00083 (2024).
>
> [10] Song, Yang, et al. "Score-Based Generative Modeling through Stochastic Differential Equations." International Conference on Learning Representations.

---

### Author Response · Authors · 2026-01-23
**Updates on the revision of the manuscript (questions related to theory and methodology)**

Dear Action Editor and Reviewers,

Hope this message finds you all well! We sincerely thank you for your time, patience and all the insightful suggestions, which have greatly helped improve the quality of our paper. We are writing to update that we have finished revising the theoretical and methodological parts of the paper based by taking all the reviewers' requested changes into consideration (In the current manuscript, all the modifications are highlighted in red).

We are currently working on the extra experiments (validation on a synthetic example from compressed sensing, nonlinear inverse problems, hyperparameter tuning, inner-workings of the algorithms, etc.) requested by all the reviewers. We will make sure to update the revision accordingly once we finished collecting all the experimental results. However, it seems to us that the added experiments won't be able to be included in the main text due to the space limit (12 pages). If the reviewers think that it will be beneficial for us to present the extra experiments around the end (Section 5) of the main text, we will be more than happy to switch from a regular submission to a long submission (more than 12 pages for the main text)

In the meantime, please don't hesitate to let us know in case the reviewers have any follow-up question regarding the revision and the rebuttals we submitted earlier. We thank the reviewers and the action editor again for all your time and help!

Best regards,

Authors of Paper 6054 submitted to TMLR

---

### Decision · Action_Editor_CGmy · 2026-02-16

**Recommendation:** Accept with minor revision

**Additional Comments:**

The authors introduce a new Sequential Monte Carlo method to solve posterior sampling problems using diffusion-based prior, referred to as the Approximation-Free Diffusion Posterior Sampler (AFDPS). The main contribution is a principled use of a diffusion model with an explicit derivation of the PDE governing the evolution of the posterior distribution. They use Sequential Monte Carlo (SMC) methods to propose two sampling algorithms to approximate the posterior. They establish an explicit upper bound on the total variation distance between the approximate posterior and the true posterior (Theorem 1). Moreover, they prove that the empirical distribution of the particle system converges to the solution of the posterior PDE in terms of the mean Wasserstein distance (Theorem 2). They also support their methodology with an extensive numerical section with Linear Inverse Problems and several baselines.

Reviewers were convinced of the interest of the proposed methodology, in particular the derivation of the partial differential equation associated with the posterior distribution and the solid theoretical guarantees provided by the authors. They raise several concerns on the experiments (focus on linear inverse problems), on the clarity of the methodology and of its comparison with existing literature on the combination of SMC with diffusion-based models. They also asked for additional comments on the assumptions and on the derivation of the methodology.

The authors provided convincing answers and significantly improved the contribution in particular by detailing the methodology and the  PDE dynamics governing the posterior evolution which is a crucial part of the contribution. They also provided detailed comments on Assumption 4.4 and on Theorem 4.2.  This result highlights the interest of the continuous-time framework to analyze the upper bound between the true posterior and the approximate distribution, which interestingly completes the rich literature on posterior sampling with SMC and diffusion-based models. I also believe that the Intuitive Explanations of the Posterior Score Approximations provided in the appendix as an answer for the reviewers is insightful for a large audience.
Other minor updates such as the choice of the particle with the largestweight as the final estimator clarifies the implementation choices and the overall readability of the paper.

In order to improve their paper, the authors also proposed several additional experiments which would highlight the practical interest of the algorithm but also the quality of the posterior approximation with synthetic data (compressed sensing problem with a Gaussian prior). This experiment along with nonlinear inverse problems would be of great interest for most of the TMLR audience. To the best of my understanding, these simulation results are not currently included in the manuscript. Their addition in the final version, even only in the appendix, would be beneficial.

**Audience:**

Yes

**Audience Explanation:**

Solving posterior sampling problems with diffusion models and SMC methods is a very active area of research. In addition, the framework allowing to derive a PDE governing the evolution of the posterior distribution and the fact that the method is theoretically grounded is of interest for the TMLR audience.

**Claims And Evidence:**

Yes

**Claims Explanation:**

The authors motivate clearly the new method which is given with theoretical guarantees and various experiments. The evidence are clear in the linear inverse problem settings. Once the additional experiments will be added in the appendix all the claims will be convinving for the TMLR audience.